# A Theoretical Framework for Partially-Observed Reward States in RLHF

**Chinmaya Kausik**
University of Michigan
ckausik@umich.edu

**Mirco Mutti**
Technion
mirco.m@technion.ac.il

**Aldo Pacchiano**\*
Boston University
Broad Insitute of MIT & Harvard
pacchian@bu.edu

**Ambuj Tewari**\*
University of Michigan
tewaria@umich.edu

## Abstract

The growing deployment of reinforcement learning from human feedback (RLHF) calls for a deeper theoretical investigation of its underlying models. The prevalent models of RLHF do not account for neuroscience-backed, partially-observed "internal states" that can affect human feedback, nor do they accommodate intermediate feedback during an interaction. Both of these can be instrumental in speeding up learning and improving alignment. To address these limitations, we model RLHF as reinforcement learning with partially observed reward-states (PORRL). We accommodate two kinds of feedback – cardinal and dueling feedback. We first demonstrate that PORRL subsumes a wide class of RL problems, including traditional RL, RLHF, and reward machines. For cardinal feedback, we present two model-based methods (POR-UCRL, POR-UCBVI). We give both cardinal regret and sample complexity guarantees for the methods, showing that they improve over naive history-summarization. We then discuss the benefits of a model-free method like GOLF with naive history-summarization in settings with recursive internal states and dense intermediate feedback. For this purpose, we define a new history aware version of the Bellman-eluder dimension and give a new guarantee for GOLF in our setting, which can be exponentially sharper in illustrative examples. For dueling feedback, we show that a naive reduction to cardinal feedback fails to achieve sublinear dueling regret. We then present the first explicit reduction that converts guarantees for cardinal regret to dueling regret. In both feedback settings, we show that our models and guarantees generalize and extend existing ones.

## 1 Introduction

As automated systems become more ubiquitous, the need to understand how to align their objectives with the needs of humans that interact with them has become increasingly important (Ji et al., 2023). The development and study of reinforcement learning from human feedback (RLHF) has been an important way of formalizing these problems and design methods for alignment (Wirth et al., 2017). RLHF is concerned with the study of how to find a policy that maximizes an objective defined in terms of human labeled data in an RL domain (Christiano et al., 2017).

Many RLHF methods entail learning a reward function from human data, and then using the learned reward function as an input to a traditional reinforcement learning algorithm such as PPO (Schulman et al., 2017). These reward-based RLHF methods have been pivotal in the development of several technologies such as robotics (Brown et al., 2019; Shin et al., 2023), recommender systems (Xue et al., 2022), and the training of large language models (LLMs) (Ouyang et al., 2022).

It is important to emphasize that reward-based RLHF is not limited to preferential feedback. In fact, there exist two dominant kinds of feedback, namely *cardinal* and *dueling* feedback. Cardinal feedback requires the human labeler to provide a single label over an entire trajectory of interaction between the agent and the environment. Dueling feedback requires the human to specify a preference between two trajectories. In practice, cardinal feedback has been used for LLM alignment algorithms

---

\*Joint senior authorship.

like KTO (Ethayarajh et al., 2024), while dueling feedback has been used in algorithms like DPO (Rafailov et al., 2023) and PPO-RLHF (Ouyang et al., 2022). Past theoretical work (Chatterji et al., 2021; Wang et al., 2023b; Saha et al., 2023) has designed algorithms for both cardinal and dueling feedback under various metrics – standard/cardinal regret, sample complexity or dueling regret.

We observe that current models of reward-based RLHF assume a very specific model of non-Markovian rewards. Modeling rewards as non-Markovian is natural, since human responses to stimuli are known to be affected by partially-observed and evolving "internal states" (Flavell et al., 2022). For example, when a human reads a piece of text (possibly generated by an LLM), their assessment may oscillate between opposing sentiments in different parts of the text. Unfortunately, current models do not explicitly incorporate such "internal states" that affect rewards, and are limited to a specific linear model of rewards. While one can incorporate internal states using naive history-summarization, i.e. by treating the entire trajectory $\tau[h]$ so far as the state, we show below that better general algorithms can be designed with improved guarantees.

Additionally, current models assume that feedback is received only once at the end of an episode or pair of episodes. In many applications such as robot motion (Lee et al., 2021) and mathematical reasoning (Uesato et al., 2022), correctly incorporating intermediate or "snippet-level" feedback can speed up learning as well as improve alignment. With this in mind, we ask the following questions:

*How do we generalize the RLHF setting to incorporate internal states and intermediate feedback? What algorithms and guarantees can improve over naive history-summarization here?*

**Contributions:**

- **Introducing PORRL:** In Section 2, we introduce PORRL, which generalizes current RLHF models to incorporate "internal states" and intermediate feedback.
- **Improving over naive history-summarization (model-based algorithms):** In Section 3.1, we design model-based optimistic algorithms that, POR-UCRL and POR-UCBVI, achieving a regret of $\widetilde{\mathcal{O}}((poly(H,S,A) + p\sqrt{d_E d_C})\sqrt{T})$ and a sample complexity of $\widetilde{\mathcal{O}}((poly(H,S,A)/\varepsilon^2 + p^2 d_E d_C/\varepsilon^2)$ under minimal assumptions.[1] The $poly(H,S,A)$ term would be $(SA)^{\Omega(H)}$ under naive history-summarization. We show that our guarantees subsume and improve over past results.
- **Leveraging recursive structure on internal states (model-free algorithms):** In Section 3.2, we study the model-free algorithm GOLF, applied using history-summarization. We define a new "history-aware" notion of dimension, $d_{\mathrm{HABE}}$ and show that GOLF has regret $\widetilde{\mathcal{O}}(pH\sqrt{d_{\mathrm{HABE}} d_C T})$. We show using an example that when internal states have a recursive structure, our guarantee can be exponentially smaller than existing guarantees and guarantees for our model-based methods.
- **Reduction from Dueling to Cardinal PORRL:** In section 4, we show that a naive blackbox reduction from dueling to cardinal PORRL always fails. We design a whitebox reduction from dueling PORRL to a large class of optimistic algorithms for cardinal PORRL. To the best of our knowledge, this is the first explicit reduction from cardinal to dueling regret guarantees for MDPs.
- **Practical Implications:** While the aim of our work is largely theoretical, we extract practical insights from our results throughout the text. These are summarized in section 5.

## 1.1 RELATED WORK

**RLHF.** RL with human preferences has a long history (Akrour et al., 2012; Busa-Fekete & Hüllermeier, 2014; Sadigh et al., 2017). It has been successfully used in disparate domains such as robotics, games, and LLMs. The problem of learning from cardinal feedback has been theoretically studied in (Efroni et al., 2021; Chatterji et al., 2021). Theoretical guarantees for utility-based preferential (dueling) feedback can be found in (Novoseller et al., 2020; Saha et al., 2023; Chen et al., 2022b; Zhan et al., 2023). The non-Markovian nature of the optimal policy under these RLHF models contributes greatly to why the problem is harder than traditional RL.

**Internal states and intermediate feedback.** There is evidence in neuroscience research indicating that human responses to stimuli are affected by "internal states" — partially hidden variables that profoundly shape perception, cognition, and action" (see Flavell et al., 2022). Despite not explicitly

---

[1] $d_E$ is a relevant eluder dimension and $d_C$ is a relevant covering dimension. We are working under general function-approximation for the *reward* model. It is straightforward to also add general function-approximation for the *transition* model, abstracting out the $S, A$ dependence. See remark 4 and Appendix F.

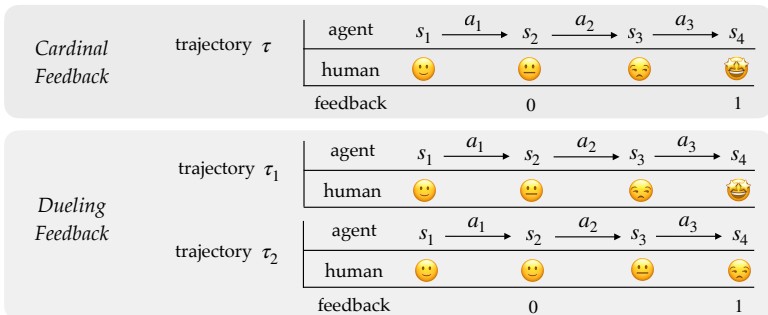

Figure 1: Illustrating how a human's internal states (represented by emojis) affect their feedback to an agent or LLM. Top: Cardinal or good/bad feedback. Bottom: Dueling or preferential feedback. In line with Definition 1, $u_h \in \mathcal{U}$ are represented by the emojis, $p = 2$ and $\mathcal{H}_p = \{2, 4\}$ in both cases.

recognizing the phenomenon of human internal states, several works in RLHF incorporate richer forms of feedback. For example, Wu et al. (2023) consider human labeling over sub-sections of the text. In work on process supervision (Uesato et al., 2022; Lightman et al., 2023), humans give feedback on intermediate steps. Our work aims to lay out the groundwork for a theoretical treatment of internal human states and intermediate feedback in RLHF, using partially observed reward-states.

**Partial observability in RL.** Although learning in POMDPs (Åström, 1965) is statistically intractable in general (Krishnamurthy et al., 2016; Jin et al., 2020), a flurry of recent works have studied POMDPs under various assumptions (Du et al., 2019; Liu et al., 2022a;b; Golowich et al., 2022; Zhan et al., 2022; Cai et al., 2022; Chen et al., 2022a; 2023; Wang et al., 2023a; Zhong et al., 2023). Our model is distinct from POMDPs since it does not require the latent state evolution to be Markovian, but assumes Markovian transitions for observed states. See Section 2.3 for a discussion.

## 2 DEFINING RL WITH PARTIALLY-OBSERVED REWARD STATES (PORRL)

In this paper, we consider an episodic reinforcement learning setting in which a learner interacts with an MDP having a state space $\mathcal{S}$, an action space $\mathcal{A}$, transitions dynamics $\mathbb{P}$, and episode length $H$. At each time-step $h \in [H]$ of an episode, the learner observes the state $s_h$ and takes an action $a_h$, generating a *trajectory* $\tau = (s_1, a_1, \cdots, s_H, a_H) \in \Gamma$, where $\Gamma$ denotes the space of trajectories.[2] In a typical RLHF setting, the learner observes a human feedback $o_H \in \mathcal{O}$ at the end of the episode, which is associated to but potentially different from a reward $r : \Gamma \to \mathbb{R}$ encoding the task. We now describe how internal states and intermediate feedback shall be incorporated in the latter RLHF framework through a guiding example, and we use this to formally introduce the PORMDP model.

### 2.1 PORMDPs

Let us consider the example of a human interacting with a language model, as in Figure 1. Here, an action is a token, the state is the text so far, and the reward is some score representing the human's satisfaction, which induces stochastic feedback. The internal states could be the human's emotional reaction to the text (e.g., happy, frustrated, or amused), or numbers in $[0, 1]$ encoding a confidence level that the text is progressing towards a coherent response. While an agent goes through a sequence of states and actions, the system (i.e., the human) progresses through internal states, which inevitably affect, together with agent's actions and the state of the process, the human's satisfaction.

Formally, this can be modeled by introducing internal states $u \in \mathcal{U}$ and defining the set of *underlying histories* $\Gamma_{h-1}^u$ that incorporate internal states by $\Gamma_{h-1}^u := \{\tau^u[h-1] = \{(s_l, u_l, a_l)\}_{l=1}^{h-1} \mid s_l \in \mathcal{S}, a_l \in \mathcal{A}, u_l \in \mathcal{U}\}$. We model the dynamics of internal states by saying that there exists an internal state generator $w_h : \Gamma_{h-1}^u \times \mathcal{S} \times \mathcal{A} \to \Delta(\mathcal{U})$ so that the human's internal state $u_h$ is sampled from the distribution defined by $w_h(\tau^u[h-1], s_h, a_h)$. The human's satisfaction at time $h$ should then be a function of the current state and action, but also the current internal state, given by $r_h(s_h, u_h, a_h)$.

The agent does not observe the reward $r_h$ directly, but a feedback $o_h$ depending on $r_h$. Typically, $o_h$ will be $\{0, 1\}$ feedback reflecting whether the human says that they are satisfied or not. In general, this

---

[2] We will further denote $\tau[h] = (s_1, a_1, \ldots, s_h, a_h)$ the sub-trajectory of $\tau$ of length $h$ and $\Gamma_h$ the corresponding space of sub-trajectories of length $h$.

could be stochastic. For instance, this could be $Ber(\sigma_h(r_h))$ for some function $\sigma_h$. So, $o_h \sim e_h(r_h)$ for some distribution $e_h(r_h)$. This leads to the general definition below, where we have introduced new objects $\mathcal{U}, \mathcal{H}_p, w, e$ not seen in traditional RL:

**Definition 1.** A PORMDP $\mathbb{M}$ with *cardinal feedback* is a tuple $(\mathcal{S}, \mathcal{A}, \mathcal{U}, \mathcal{O}, \mathbb{P}, \mathcal{H}_p, r, w, e)$, where:

- $\mathcal{S}, \mathcal{A}$ are fully observable states and actions, $\mathcal{U}$ are *unobserved internal reward-states*, $\mathcal{O}$ is a space of feedback, $\mathbb{P}(\cdot \mid s, a)$ is a Markovian transition matrix, $s_1 \in \mathcal{S}$ is an initial state.[3]

- $\mathcal{H}_p \subset [H]$ is a set of timesteps where reward and feedback is obtained with size $|\mathcal{H}_p| = p$.

- $r := \{r_h\}_{h \in \mathcal{H}_p}$ so that $r_h : \mathcal{S} \times \mathcal{U} \times \mathcal{A} \to \mathbb{R}$ are reward functions at time $h$.

- $w := \{w_h\}_{h \in \mathcal{H}_p}$ so that $w_h : \Gamma_{h-1}^u \times \mathcal{S} \times \mathcal{A} \to \Delta(\mathcal{U})$ are *internal state generators* that map underlying histories of $(s, a, u)$ tuples to distributions over $\mathcal{U}$.

- $e := \{e_h\}_{h \in \mathcal{H}_p}$ are *feedback functions* so that the feedback $o_h \sim e_h(r_h)$ is sampled from an $\eta_h$-subgaussian distribution $e_h$ with mean $\sigma_h(r_h)$ for some activation function $\sigma_h : \mathbb{R} \to \mathbb{R}$.[4]

In some relevant RLHF applications, the human is presented with two trajectories and they provide feedback based on the pair. In most cases, this involves indicating a 0-1 preference between trajectories. To accommodate this setting, we extend the framework to dueling feedback.

**Definition 2.** A PORMDP $\mathbb{M}$ with *dueling feedback* is a tuple $(\mathcal{S}, \mathcal{A}, \mathcal{U}, \mathcal{O}, \mathbb{P}, \mathcal{H}_p, r, w, e)$, where everything is identical to Definition 1, except that every episode now involves running two trajectories $\tau_1, \tau_2$ that produce rewards $r_{h,1}, r_{h,2} \; \forall h \in \mathcal{H}_p$, and feedback is distributed as $o_h \sim e_h(r_{h,1} - r_{h,2})$.

We note that PORMDPs subsume and model a wide class of RL settings, including RLHF. A brief list of settings that PORMDPs subsume is as follows: (i) traditional MDPs, by setting $\mathcal{U} = \{\star\}$; (ii) existing linear models of RLHF, setting $\mathcal{U} = \{\phi(\tau)^\top \mathbf{w}\}$ for a known feature map $\phi$ and unknown $\mathbf{w}$ (Chatterji et al., 2021; Efroni et al., 2021; Saha et al., 2023; Wang et al., 2023b); (iii) learning reward models with stochastic feedback by setting $\mathcal{U}$ to be the set of reward states (Icarte et al., 2019; 2018; 2022; Icarte, 2022). By using $\mathcal{U}$ to model implicit intentions, PORMDPs can also model learning from the following feedback: (iv) process supervision (Lightman et al., 2023; Uesato et al., 2022), (v) fine-grained feedback (Wu et al., 2023) and (vi) snippet-level feedback (Lee et al., 2021). Further, one can show that in all these settings, we can define the $\mathcal{U}$ generators $w_h$ to be deterministic.

One illustrative hard example of PORRL is that of a *combination lock*,[5] which we will also use later in the paper. Consider an $H$-digit numerical lock with a set $\mathcal{A}$ of options at each digit. Let the true combination be $a_1^\star, \ldots a_H^\star$. An agent tries to unlock it by listening for "clicks" while rotating the dial at each digit $h$. Naturally, we only hear clicks at digit $h$ if the entire combination so far is correct. We thus model this as a PORMDP with non-Markovian rewards, $\mathcal{S} = \{\star\}, \mathcal{U} = \{\bigcup_h \mathcal{A}^h\}$ and the appropriate dynamics. Arguing that the click might sometimes be too faint, we consider stochastic rewards. Specifically, we model this as $r_h(s_h, u_h, a_h) = Ber(q\mathbb{1}_{a_1^\star, \ldots a_h^\star}(u_h))$ for some uncertainty parameter $q$. Notice that the internal states have a recursive structure here, and they evolve in a Markovian way. This is a toy model for the problem of learning to take desirable sequences of actions using intermediate feedback. It can be viewed as a simplified version of many such tasks – navigating mazes, writing structured essays with guidance, writing a proof with feedback on correctness.

## 2.2 Reinforcement Learning in PORMDPs with Cardinal and Dueling Feedback

Due to the complex nature of observability in our problem, we will use this subsection to carefully set up a meaningful set of RL problems, in which an agent interacts with a PORMDP to optimize a policy. At each step $h$, the agent observes a history $\tau[h-1] \in \Gamma_{h-1}$ and takes an action $a_h \sim \pi(\tau[h-1], s_h)$. The agent does not observe the reward $r_h$, but receives an observation $o_h \sim e_h(r_h)$.

**Defining the learning objective.** Since rewards are partially observed and dependent on the entire history, there is a subtlety in defining value functions. We first choose and fix some subclass $\Pi$ of history-dependent policies and we define the total expected reward of a policy $\pi \in \Pi$ as

$$V_w(\mathbb{M}, \pi) := \mathbb{E}_{\tau^u \sim \mathbb{P}^{w, \pi}} \left[ \sum_{h \in \mathcal{H}_p} r_h(s_h, u_h, a_h) \right]$$

---

[3]Recall that choosing a formal state $s_1$ to serve as a placeholder initial state is not restrictive.

[4]This subsumes and generalizes the example of Bernoulli feedback in RLHF.

[5]This is a variant of a common example used to generate lower bounds in POMDPs (Krishnamurthy et al., 2016; Jin et al., 2020). In contrast, we will use it to illustrate the power of our upper bounds.

$V_w(\mathbb{M}, \pi)$ is taking an expectation over the dynamics of *underlying* trajectories $\tau^u = \{(s_h, u_h, a_h)\}_{h=1}^H \sim \mathbb{P}^{w,\pi}$. Since the states $u$ are never revealed, these dynamics can never be learnt, making $V_w$ hard to directly deal with. In this light, we introduce stochastic functions $g_h : \Gamma_h \to \Delta(\mathcal{U})$ that marginalize the internal state generator $w_h$ over the sequence $u_1, \ldots u_{h-1}$. That is, given an $(s, a)$ history $\tau[h]$, we can define[6] $g_h(\tau[h]) \sim u_h \mid \tau[h]$. Now define

$$V_g(\mathbb{M}, \pi) := \mathbb{E}_{\tau \sim \mathbb{P}^\pi} \left[ \sum_{h \in \mathcal{H}_p} \mathbb{E}_{u_h \sim g_h(\tau[h])} \big[ r_h(s_h, u_h, a_h) \big] \right]$$

$V_g(\mathbb{M}, \pi)$ is a much more tractable object, where the outer expectation is taken over the dynamics of the *observed* trajectories $\tau$. The following result establishes that as one would hope, $V_w = V_g$.

**Lemma 1** (Replacing $w$ with $g$)**.** *For any history-dependent policy $\pi$ that selects an action $a_h \sim \pi(\tau[h-1], s_h)$, $V_w(\mathbb{M}, \pi) = V_g(\mathbb{M}, \pi)$ holds for any $\mathbb{M}$.*

For the purposes of value functions, $\mathbb{M}$ is fully specified by $(\mathcal{S}, \mathcal{A}, \mathcal{U}, \mathcal{O}, \mathbb{P}, \mathcal{H}_p, r, g, e)$. Henceforth we replace $w$ with $g$ and denote $V_g(\mathbb{M}, \pi)$ by $V(\mathbb{M}, \pi)$. The optimal policy is $\pi_\star := \arg\max_{\pi \in \Pi} V(\mathbb{M}, \pi)$.

**Cardinal PORRL.** Consider an algorithm producing a sequence of policies $\pi_1, \ldots, \pi_T \in \Pi$, where $\pi_t$ is chosen only using trajectories $\{\tau_i\}_{i=1}^{t-1}$ generated by $\{\pi_i\}_{i=1}^{t-1}$. We measure the performance of such an algorithm by its *cardinal regret* under model $\mathbb{M}_\star$:

$$\text{Regret}(T) = \sum_{t=1}^T V(\mathbb{M}_\star, \pi_\star) - V(\mathbb{M}_\star, \pi_t)$$

One can also ask for the sample complexity of learning a good policy. Given a randomized algorithm that completes $N$ episodes of interaction and outputs $\pi_N$, the *sample complexity* $N(\varepsilon, \delta)$ of the algorithm is the minimum $N$ so that $V(\mathbb{M}_\star, \pi_\star) - V(\mathbb{M}_\star, \pi_N) \leqslant \varepsilon$ with probability at least $1 - \delta$ over the randomness of the feedback and the algorithm. It makes sense to study cardinal regret and sample complexity in two RLHF settings:

- Using a learnt reward model: In most deployments of offline RLHF, an offline dataset of dueling feedback from humans is typically used to create a cardinal feedback oracle (a reward model), which is then used to train the policy using RL. In fact, Anonymous (2024) do exactly this under our model. The sample complexity of the algorithm is important in this setting.
- Improving a deployed model with batched feedback: One can learn from batches of interaction with humans and hope to improve the model/policy adaptively over multiple batches. This is compatible with deploying LLMs or recommender systems to users, collecting a batch of good/bad feedback, and then fine-tuning the model offline using this batch. This approach is also discussed in (Swamy et al., 2024; Dong et al., 2024). Regret is a better metric than sample complexity here, since we want users to be satisfied (exploiting) while improving the model (exploring). Instead of good/bad feedback, we can also ask for dueling feedback against a fixed $\pi_0$ and treat it as cardinal feedback.[7]

**Dueling PORRL.** In dueling PORRL, we play a *duel* by running two policies $(\pi_1, \pi_2) \in \Pi \times \Pi$ in parallel to obtain trajectories $(\tau_1, \tau_2)$ and receive feedback $\{o_h\}_{h \in \mathcal{H}_p}$. Again, note that the rewards of the policies are not observed. While the definitions of $V(\mathbb{M}, \pi)$ and $\pi_\star$ are the same as before, we define a new measure of regret accordingly. If we play $T$ duels $(\pi_{1,1}, \pi_{2,1}), \ldots, (\pi_{1,T}, \pi_{2,T})$ according to an algorithm, we aim to minimize the *dueling regret* given by

$$\text{Regret}_D(T) = \sum_{t=1}^T V(\mathbb{M}_\star, \pi_\star) - \frac{V(\mathbb{M}_\star, \pi_{1,t}) + V(\mathbb{M}_\star, \pi_{2,t})}{2}$$

It makes sense to consider this metric when improving a deployed model with batched dueling feedback. We can do the same batching as the batched feedback example above, but instead compare our model/policy $\pi_t$ to a fixed base policy $\pi_0$ and ask for dueling feedback. The induced feedback can be treated as cardinal feedback. This is similar to the ideas in Wang et al. (2023b), who consider this setting and give cardinal regret/sample complexity guarantees. However, when deploying a model, we typically want humans to be satisfied with *both* the options they are given. Cardinal regret only accounts for one of the options being good. Dueling regret demands that *both* policies used are good.

---

[6]More technically, define $g_h(\tau[h])$ to be the *regular conditional distribution* of the random variable $w_h((\tau[h-1], u_1, \ldots u_{h-1}), s_h, a_h)$, conditioned on $\tau[h]$.

[7]If the activation function is Lipschitz and monotone, then we can get cardinal regret guarantees for this problem by using the difference function class.

**Remark 1.** PORRL subsumes the settings of (Saha et al., 2023; Chatterji et al., 2021), which in turn subsume the feedback models of RLHF (Wang et al., 2023b). Crucially, (Wang et al., 2023b; Chatterji et al., 2021) measure performance using only sample complexity or cardinal regret, while (Saha et al., 2023) only study dueling regret. We have discussed above why both metrics are important.

## 2.3 A GENERAL YET TRACTABLE CASE

The nature of the feedback in PORMDPs, which depends on a reward that is function of the entire history, signals that PORRL may be intractable in general. We now instantiate the model into a statistically tractable sub-class that still subsumes most existing work on RLHF and all the examples provided at the end of Section 2.1. Specifically, we assume that the internal reward-state functions $g_h$ are deterministic and the feedback is emitted according to a Bernoulli distribution depending on the reward. We will work under this assumption in the remainder of the paper.

**Assumption 1.** We work in a realizable setting. That is, the unknown transition kernel $\mathbb{P}$ lies in a known class $\mathcal{P}$, and the unknown reward function $r_h : \mathcal{S} \times \mathcal{U} \times \mathcal{A} \to \mathbb{R}$ lies in a known class $\mathcal{R}_h$ with $|r_h| \leq B$ for all $h$. Assume that $g_h$ is deterministic[8] (but unknown) and belongs to a known class of "decoder functions" $\mathcal{G}_h$. Let $\mathcal{O} = \{0, 1\}$ and let $e_h$ only depend on the rewards. For dueling feedback, let $e_h(r_{h,1} - r_{h,2})$ be $\eta_h$-subgaussian with unknown mean $\sigma_h(r_{1,h} - r_{2,h})$. Also assume that $\sigma_h$ and $\sigma_h^{-1}$ are Lipschitz with unknown Lipschitz constants $\kappa_{1,h}$ and $\kappa_{2,h}$ respectively. Call the resulting class of PORMDPs $\mathcal{M}$.

We also define a function class induced by $\mathcal{R}_h$ and $\mathcal{G}_h$.

**Definition 3.** Let us then consider the decoder-induced function classes $\mathcal{F}_h$ given by

$$\mathcal{F}_h := \left\{ f_h : \Gamma_h \to \mathbb{R} \;\middle|\; \exists g_h \in \mathcal{G}_h, r_h \in \mathcal{R}_h \quad \text{s.t.} \quad f_h(\tau[h]) = r_h(s_h, g_h(\tau[h-1]), a_h), \; \forall \tau \right\}$$

Also define $\mathcal{F} := \prod_{h \in \mathcal{H}_p} \mathcal{F}_h$ so that $f = \{f_h\}_{h \in \mathcal{H}_p} \in \mathcal{F}$. A model $\mathbb{M}$ is then fully determined by $(\mathbb{P}, f)$, so we denote $V(\mathbb{P}, f, \pi) := V(\mathbb{M}, \pi)$. Note that $V(\mathbb{P}, f, \pi) = \mathbb{E}_{\tau \in \mathbb{P}^\pi} \left[ \sum_{h \in \mathcal{H}_p} f_h(\tau[h]) \right]$.

**Remark 2.** We note that all examples from Section 2.1 satisfy Assumption 1.

Giving statistically efficient algorithms for this framework comes with numerous challenges:

- **Traditional RL incurs linear regret:** We show in Lemma 3 that any method returning a possibly time-dependent but memoryless policy can incur linear regret.
- **POMDP results do not apply:** PORMDPs cannot be viewed as a subcase of POMDPs with latent states $\mathcal{S} \times \mathcal{U}$ since $s, u, a \to s', u'$ is not Markovian.[9] Even if we considered the subclass of PORMDPs where $s, u, a \to s', u'$ is Markovian, which would be a subclass of reward machines, this is a specific kind of overcomplete POMDP. Literature on overcomplete POMDPs is much more scarce than their undercomplete counterpart. The only paper that gives guarantees for overcomplete POMDPs to our knowledge is (Liu et al., 2022a), where they assume that the reward function is *fully* observable and only depends on *observed* states. This cannot apply to our setting, since our rewards have to be *partially* observable, and fundamentally depend on latent states too. Also, this is not a minor difference, since the number of latent states can be $(SA)^{\Omega(H)}$.
- **Naive history-summarization is inefficient:** It is overkill to use naive history-summarization – where one treats the history $\tau[h]$ as the state $s_h$ and executes traditional RL. This is because while policies are non-Markovian, state transitions are Markovian. It is unclear if we can leverage this structure without running into explicit exponential dependence on $H$. Moreover, most work on MDPs works with known rewards, but not knowing the rewards is a truly non-trivial problem here, since exploring the reward at each latent state could take $(SA)^{\Omega(H)}$ steps.
- **Ensuring satisfactory utilization of additional structure:** Examples like the combination lock signal that there are intuitive ways to leverage a recursive structure on the internal states. In the combination lock, one should wait for the "click" at each digit before moving onto the next digit, giving us a polynomial dependence on $A, H$ in sample complexity. It is unclear if *general* algorithms for PORRL can implicitly leverage such structure to achieve polynomial guarantees.

---

[8]We make this assumption for simplicity of exposition, it is not necessary. As long as $f_h$ is $\eta_h$ subgaussian conditioned on $\tau[h]$, all our theory follows verbatim irrespective of whether $g_h$ is deterministic or stochastic.

[9]Since observed state transitions are Markovian, PORMDPs are also not more general than POMDPs.

## 3 Optimistic Algorithms for Cardinal PORRL

### 3.1 Improving over Naive History-Summarization with Model-Based Methods

We present two optimistic methods that leverage Markovian transitions in PORMDPs – POR-UCRL and POR-UCBVI. The methods explicitly learn both the unknown reward model and the unknown transition model, while still accounting for the Markovian nature of transitions. We describe them below and provide formal versions in Appendix D, E.

- **POR-UCRL:** At each timestep $t$, we maintain a least squares estimate $\hat{f}^{t+1}$ of $f$ and an MLE estimate $\hat{\mathbb{P}}_t$ and define confidence sets $\mathcal{C}_h^t(\delta)$ that consider all $f_h$ with a small mean squared error against $\hat{f}_h^{t+1}$, such that $\mathcal{C}_{\mathcal{F}}^t(\delta) = \prod_{h=1}^{H} \mathcal{C}_h^t(\delta)$. The probability transition confidence sets $\mathcal{C}_{\mathcal{P}}^t(\delta)$ are the same as UCRL (Jaksch et al., 2010). At timestep $t$, following confidence-set optimism, we play an optimistic policy $\tilde{\pi}_t$ that maximizes its highest value $V(\mathbb{M}, \pi)$ over all models $\mathbb{M} \in \mathcal{C}_{\mathcal{F}}^t \times \mathcal{C}_{\mathcal{P}}^t$.
- **POR-UCBVI:** It is trickier to adapt ideas from UCBVI (Azar et al., 2017). Yet again, we maintain a least squares estimate $\hat{f}^t$ and an MLE estimate $\hat{\mathbb{P}}_t$. Instead of confidence sets, we define trajectory-dependent bonuses for $\mathcal{F}$ as $b_{\mathcal{F}}^t(\tau, \delta) := \sum_{h \in \mathcal{H}_p} \max_{f_h, f_h' \in \mathcal{C}_h^t(\delta)} f_h(\tau[h]) - f_h'(\tau[h])$. We use these to define policy-level bonuses for $\mathcal{F}$ as $b_{\mathcal{F}}^t(\mathbb{P}, \pi, \delta) := \mathbb{E}_{\tau \sim \mathbb{P}^\pi} [b_{\mathcal{F}}^t(\tau, \delta)]$. Then, the standard UCBVI bonuses provide policy-level bonuses for $\mathcal{P}$. At timestep $t$, following bonus-based optimism, we play an optimistic policy $\tilde{\pi}_t$ that maximizes its bonus-boosted value under $\hat{f}^t, \hat{\mathbb{P}}^t$. POR-UCBVI bonuses are in fact computable for many $\mathcal{U}$ and $\mathcal{F}$, such as those in remark 3.

We show that POR-UCRL enjoys the guarantee below.

**Theorem 1** (POR-UCRL Regret). *Under Assumption 1, the regret* $\mathrm{Regret}(T)$ *of POR-UCRL is bounded by the following with probability at least* $1 - \delta$

$$\widetilde{\mathcal{O}}\left( \left( pS\sqrt{HA} + \sum_{h \in \mathcal{H}_p} \sqrt{d_{E,h} d_{C,h}} \right) \sqrt{T} \right)$$

*where* $d_{E,h} = \dim_E\left(\mathcal{F}_h, \frac{B}{T}\right)$ *and* $d_{C,h} = \log(\mathcal{N}(\mathcal{F}_h, 1/T, \|\cdot\|_\infty))$.

Here, the first term comes from uncertainty in $\mathbb{P}$. Under naive history-summarization, the first term would be exponential in $H$ since the modified state space of trajectories would have size $\Omega((SA)^H)$. Similar regret guarantees are given for POR-UCBVI in Theorem 8. Both guarantees are proved by viewing each algorithm as a specific instance of a generic optimistic algorithm for PORRL (see Appendix C, D, E). By a simple regret-to-PAC conversion, we also show that POR-UCRL has sample complexity of $\widetilde{\mathcal{O}}\left( \frac{p^2 HS^2 A}{\varepsilon^2} + \frac{p^2 d_E d_C}{\varepsilon^2} \right)$, where $d_E := \max_{h \in \mathcal{H}_p} d_{E,h}$, and $d_C := \max_{h \in \mathcal{H}_p} d_{C,h}$. POR-UCBVI has sample complexity $\widetilde{\mathcal{O}}\left( \frac{p^2 HSA \max(H,S)}{\varepsilon^2} + \frac{p^2 d_E \max(d_C, H) \log(1/\delta)}{\varepsilon^2} \right)$.

**Challenges:** There are three main technical challenges in proving these guarantees. First, we have to handle non-Markovian reward functions with Markovian transitions. Second, in POR-UCBVI, we have the added challenge of ensuring that the bonus is uniformly optimistic over all history-dependent policies. This is typically a doubly exponential set $(A^{(SA)^H})$, so a union bound does not help us. Third, we are working with general function approximation for reward functions using $\mathcal{F}$.

**Remark 3** (Comparison to past results). Notice that with $\mathcal{U} = \phi(\tau)^\top \mathbf{w}$ with $\mathbf{w} \in \mathbb{R}^d$ and $\mathcal{H}_p = \{H\}$, we are in the setting of (Chatterji et al., 2021). Here, $d_{E,H} = d_{C,H} = d$, so POR-UCRL and POR-UCBVI both improve over their regret guarantees. With respect to sample complexity guarantees, we compare to (Wang et al., 2023b). While they use dueling feedback, our methods use cardinal feedback. In their setting, $\mathcal{U}$ is the set of all histories and $\mathcal{H}_p = \{H\}$. Their best guarantee is from P-OMLE, which makes $\widetilde{\mathcal{O}}\left( \frac{H^2 S^2 A}{\varepsilon^2} + \frac{H^2 d_{E,H} d_{C,H}}{\varepsilon^2} \right)$ dueling oracle queries for tabular $\mathcal{P}$. Both POR-UCRL and POR-UCBVI have a smaller complexity for cardinal feedback queries.

**Remark 4** (General function approximation for $\mathcal{P}$). For clearer exposition, we have assumed that $\mathcal{P}$ is a tabular class with finite $\mathcal{S}, \mathcal{A}$ in the results stated above. This is because handling general function approximation for $\mathcal{F}$ is the non-trivial part of this work. We provide straightforward extensions to general function approximation for $\mathcal{P}$ with continuous $\mathcal{S}, \mathcal{A}$ in Appendix F, using existing work.

### 3.2 Leveraging Recursive Structures Using Model-Free Methods

We have established that the model-based methods POR-UCRL and POR-UCBVI improve over naive history-summarization and have a $poly(S, A, H)$ guarantee in terms of transition function estimation.

However, we recall the last challenge mentioned in Section 2.3 – can they adapt to examples like the combination lock, where there is a recursive structure on the internal states? Disappointingly, we will see in Proposition 1 that the answer is no – they are *exponentially* worse than the ideal solution. Intuitively, learning the reward and transition models separately is needlessly expensive here. At the more technical level, since POR-UCRL and POR-UCBVI decouple the learning of reward functions across timesteps, they are unable to incorporate a recursive structure on the reward functions.

In this light, we consider model-free methods. Unlike model-based methods that have to account for Markovian transitions, we can simply use naive history-summarization here and treat $\tau[h]$ as the state for Q-functions $Q_h$. However, under history-summarization, there is a subtlety involved in choosing the class $\mathcal{Q}$ of Q-functions given a known class $\mathcal{M}$ of models. Using product classes $\mathcal{Q}_1 \times \cdots \times \mathcal{Q}_H$ is wasteful, since often exponentially many tuples in a product class cannot be realized by any model $\mathcal{M}$.[10] Instead, one should consider the class of only the tuples $(Q_1, \ldots, Q_H)$ that can be *realized* by a model $\mathcal{M}$. In practice, this translates to the problem of good representation learning – one should use a shared network for all Q-functions instead of using a different network for each timestep. This is reflected in the experimental choices of Anonymous (2024).

Model-free methods rely on the Bellman error, which relates consecutive Q-functions and couples their learning. It is thus natural to expect model-free methods like GOLF (Jin et al., 2021) to adapt to a recursive structure on internal states and perform better than model-based methods. However, existing guarantees do not reflect this. It turns out from Proposition 1 below that the Bellman-eluder (BE) dimension of the combination lock problem is $A^H$, even with the minimal Q-function class.

The issue is that the proof of GOLF bounds the $h$-step Bellman errors in a decoupled manner, which is why it still fails to incorporate a recursive structure on internal states. Intuitively, one wants to *wait* for Bellman errors at timesteps $1, \ldots, h-1$ to become small before bounding the Bellman error at $h$. In this light, given a parameter $\alpha$, we define the function class

$$\mathcal{Q}(\alpha, h) := \left\{ Q \in \mathcal{Q} \mid |\mathbb{E}_{\mu_l(Q)}[Q_l - \mathcal{T}_l Q_{l+1}]| \leqslant \alpha, \ \forall 1 \leqslant l \leqslant h \right\}$$

that considers all tuples $(Q_1, \ldots, Q_H)$ where the Bellman errors until step $h$ are already low. We can use this class to define the $\alpha$-history aware Bellman-eluder dimension (HABE) of $\mathcal{Q}$ as follows. Recall that $\pi_Q$ is the policy that acts greedily according to $Q = (Q_1, \ldots Q_H)$.

**Definition 4.** Consider the Bellman errors $\Phi_h := \left\{ Q_h - \mathcal{T}_h Q_{h+1} \mid Q \in \mathcal{Q}(\alpha, h-1) \right\}$. Denote $\mu_h(Q)$ the distribution induced on $\tau[h-1], a_h$ by $\pi_Q$ and let $\mathcal{D}_{h,\mathcal{Q}} := \{\mu_h(Q) \mid Q \in \mathcal{Q}\}$. Let $\dim_{DE}$ the distributional eluder dimension and define $\dim_{\text{HABE}}(\mathcal{Q}, \alpha, \varepsilon) := \max_h \dim_{DE}(\Phi_h, \mathcal{D}_{h,\mathcal{Q}(\alpha,h-1)}, \varepsilon)$.

Intuitively, $\alpha$-HABE dimension measures how hard it is to reduce the Bellman error at timestep $h$ if the errors at *previous timesteps* $1, \ldots, h-1$ are already small. This captures the hardness of adapting to the recursive structure on internal states one/a few timesteps at a time. We discuss in Appendix G.1 how the $\alpha$-HABE dimension compares to the Bellman-eluder dimension in general. We now give a new guarantee for GOLF using the $\alpha$-HABE dimension.

**Theorem 2** (Modified GOLF Regret). *Let Assumption 1 hold, let $\mathcal{Q}$ be Bellman complete, let $d_{\text{HABE}} = \dim_{\text{HABE}}(\mathcal{Q}, \alpha, \min(\alpha, \sqrt{1/T}))$ and let $d_{C,\mathcal{Q}} := \log(\mathcal{N}(\mathcal{Q} \cup \mathcal{G}, 1/T, \|\cdot\|_\infty))$. Choose hyperparameter $\beta = c(\log(HT) + d_{C,\mathcal{Q}})$ for some universal constant $c$ and the auxiliary function class $\mathcal{G}$ used in GOLF, and define . Then, GOLF satisfies $\text{Regret}(T) = \mathcal{O}\left(pH\sqrt{d_{\text{HABE}}d_{C,\mathcal{Q}}T}\right)$.*

Using a regret-to-PAC conversion, we also show in Corollary 7 that the sample complexity of GOLF is $\widetilde{\mathcal{O}}\left(\frac{p^2 H^2 d_{\text{HABE}} d_{C,\mathcal{Q}}}{\varepsilon^2}\right)$. As foreshadowed above, we now show in Proposition 1 that these guarantees can be polynomial even when the the usual guarantees for GOLF as well as guarantees for our model-based algorithms are exponential. Note that this improvement is achieved only given dense intermediate feedback. Under sparse intermediate feedback, one cannot adapt to internal states "a few timesteps at a time," and we in fact have $\Omega(\sqrt{A^H T})$ regret under *any* algorithm. However, dense feedback case is quite realistic for many applications, such as automated mathematical reasoning.

**Proposition 1** (Dimensions for the Combination Lock). *Consider the combination lock problem with model class $\mathcal{M} = \mathcal{P} \times \mathcal{F}$ and induced Q-function class $\mathcal{Q}$.*

---

[10]The reader can use the example of the combination lock to convince themselves of this.

- *Under dense intermediate feedback with $\mathcal{H}_p = [H]$, $\dim_{\mathrm{HABE}}(\mathcal{Q}, \alpha) = A$ for all $\alpha < q$, while its BE dimension is at least $A^H - 2$. The eluder dimension for reward functions $\dim_E(\mathcal{F}_h, \frac{B}{T})$ is at least $A^h$ for any $h \leqslant H$.*
- *For sparse intermediate feedback with $\mathcal{H}_p = \{H\}$ and any $\alpha > 0$, the $\alpha$-HABE dimension, the BE dimension and the eluder dimension of $\mathcal{F}_H$ are all at least $A^H - 2$. Moreover, any algorithm in this setting will have regret $\Omega(\sqrt{A^H T})$.*

We discuss in Appendix G.1 that in general, we do not have an inequality in either direction between the $\alpha$-HABE dimension and the BE dimension. However, the $\alpha$-HABE dimension is typically smaller.

## 4 Dueling to Optimism Reduction

The dueling and cardinal feedback models are intimately related. It is thus tempting to use algorithms for cardinal PORRL to solve dueling PORRL. However, we detail why the "obvious" reduction from dueling feedback to cardinal feedback fails. This both demonstrates the hardness of the problem and motivates our reduction.

### 4.1 The Naive Reduction Always Fails

Consider a modified PORMDP $\overline{\mathbb{M}}$ with $\overline{\mathcal{S}} := \mathcal{S} \times \mathcal{S}$, $\overline{\mathcal{A}} := \mathcal{A} \times \mathcal{A}$, $\overline{\mathbb{P}} := \mathbb{P} \otimes \mathbb{P}$, where we run the pair of policies $\overline{\pi} := (\pi_1, \pi_2)$ and obtain observations based on the decoder-induced function $\overline{f}_h(\tau_1[h], \tau_2[h]) := \underline{f_h(\tau_1[h])} - f_h(\tau_2[h])$. Consider the space of all such PORMDPs induced by $\mathcal{M}$, and denote it by $\overline{\mathcal{M}}$. Since cardinal feedback in $\overline{\mathbb{M}}$ exactly corresponds to dueling feedback in $\mathbb{M}$, it is tempting to restrict to searching over $\Pi \times \Pi$ and run any algorithm for cardinal PORRL on this modified PORMDP $\overline{\mathbb{M}}$ to achieve low dueling regret.

This fails because the feedback model and regret metric are fundamentally non-aligned in dueling feedback, unlike in cardinal feedback. While the agent receives dueling feedback over the duel for $(\pi_{1,t}, \pi_{2,t})$, dueling regret is instead concerned with duels for $(\pi_\star, \pi_{1,t})$ and $(\pi_\star, \pi_{2,t})$. Running an algorithm for cardinal PORRL on the modified MDP will maximize the dueling *feedback* itself. This is achieved by playing one good and one really bad policy, unlike the two good policies needed for low dueling regret. We formalize this in Lemma 4, showing that the naive reduction leads to linear dueling regret for *any* PORMDP and *any* cardinal PORRL algorithm with sublinear regret.

### 4.2 Reducing Dueling to Optimistic Cardinal PORRL

The naive reduction fails because maximizing dueling feedback can lead to bad policies being played. In this subsection, we present a white-box reduction where we ensure that we only play potentially good policies for *both* $\pi_{1,t}$ and $\pi_{2,t}$. We detail here how we can obtain an algorithm for the dueling feedback problem from *any* optimistic algorithm for cardinal PORRL. We will focus on the case of confidence sets here for smoother exposition, the much harder case of bonuses is treated in Appendix H.2. A *generic optimistic algorithm using confidence sets* maintains confidence sets $\mathcal{C}_\mathcal{M}(\mathcal{D}_t, \delta)$ using the collected dataset $\mathcal{D}_t$ of trajectories and feedback. We define it formally in Appendix C.1. For the reduction to work, we require that the confidence sets are well-designed, as demanded by Assumption 2. This assumption is satisfied for confidence sets used by POR-UCRL.

**Assumption 2** (Controlling Value Error due to Confidence Sets). $\mathbb{M}_\star \in \mathcal{C}_\mathcal{M}(\mathcal{D}_t, \delta)$ for arbitrary sequences $(\mathbb{P}_t, f^t) \in \mathcal{C}_\mathcal{M}(\mathcal{D}_t, \delta)$, both $\left| \sum_{t=1}^T V(\mathbb{P}_t, f^t, \pi_t) - V(\mathbb{P}_\star, f^t, \pi_t) \right| = \widetilde{\mathcal{O}}(C_P(\mathcal{M}, T, \delta))$ and $\left| \sum_{t=1}^T V(\mathbb{P}_\star, f^t, \pi_t) - V(\mathbb{P}_\star, f_\star, \pi_t) \right| = \widetilde{\mathcal{O}}(C_F(\mathcal{M}, T, \delta))$ hold with probability $1 - \delta/2$ each.

The key insight is to use confidence sets from cardinal PORRL to search for $\pi_{1,t}$ and $\pi_{2,t}$ only among policies that *both* have a chance of being optimal. Then one plays the *most uncertain* duel among all possible choices for $\pi_{1,t}$ and $\pi_{2,t}$. This generalizes and abstracts out ideas in (Pacchiano et al., 2021), which presents a specific algorithm to achieve low dueling regret in their model. We present the reduction to optimism over confidence sets in Algorithm 1, the version for bonuses is in Appendix H.2. Define $V_D(\overline{\mathbb{M}}, \pi, \pi') = V(\mathbb{M}, \pi) - V(\mathbb{M}, \pi')$. We compute the confidence sets $\mathcal{C}_{\overline{\mathcal{P}}}(\mathcal{D}, \delta)$ as the image of $\mathcal{C}_\mathcal{P}(\mathcal{D}, \delta)$ under $\mathbb{P} \mapsto \overline{\mathbb{P}}$. We compute $\mathcal{C}_{\overline{\mathcal{F}}}(\mathcal{D}, \delta)$ by treating $\{o_h\}_{h \in \mathcal{H}_p}$ as cardinal feedback in $\overline{\mathbb{M}}$. As an example, for POR-UCRL, we perform a least squares fit for $\overline{f}$ and use Lemma 7 to define our confidence sets again. We then get the following regret guarantee.

---

**Algorithm 1** Reduction from Dueling to Cardinal Confidence-Set Optimism

---

1: **Input** Known reward function $\{r_h\}_{h=1}^H$, method to compute $\mathcal{C}_{\overline{\mathcal{M}}}(\mathcal{D}, \delta) \leftrightarrow \mathcal{C}_{\overline{\mathcal{P}}}(\mathcal{D}, \delta) \times \mathcal{C}_{\overline{\mathcal{F}}}(\mathcal{D}, \delta)$

2: **Initialize** dataset $\mathcal{D}_1 \leftarrow \{\}, \mathcal{C}_{\overline{\mathcal{M}}}(\mathcal{D}_1, \delta) := \overline{\mathcal{P}} \times \overline{\mathcal{F}}$

3: **for** $t = 1, ..., T$ **do**

4:      **Compute** $\Pi_t = \left\{ \pi \in \Pi \middle| \exists \overline{\mathbb{M}} \in \mathcal{C}_{\overline{\mathcal{M}}}(\mathcal{D}_t, \delta) \text{ s.t. } V(\overline{\mathbb{M}}, \pi, \pi_1) \geqslant 0 \; \forall \pi_1 \in \Pi \right\}$    {Candidates $\pi_\star$}

5:      **Play** $(\pi_{1,t}, \pi_{2,t}) \in \underset{\pi, \pi' \in \Pi_t}{\arg\max} \; \underset{\overline{\mathbb{M}}, \overline{\mathbb{M}}' \in \mathcal{C}_{\overline{\mathcal{M}}}(\mathcal{D}_t, \delta)}{\max} V_D(\overline{\mathbb{M}}, \pi, \pi') - V_D(\overline{\mathbb{M}}', \pi, \pi')$ {Most uncertain duel}

6:      **Observe** trajectories $\tau_{i,t} = \left\{ (s_{i,h}^t, a_{i,h}^t) \right\}_{h=1}^H$ along with feedback $\{o_h\}_{h \in \mathcal{H}_p}$

7:      **Update** $\mathcal{D}_t$ to $\mathcal{D}_{t+1}$ using the data and compute $\mathcal{C}_{\overline{\mathcal{P}}}(\mathcal{D}_{t+1}, \delta), \mathcal{C}_{\overline{\mathcal{F}}}(\mathcal{D}_{t+1}, \delta)$

8: **end for**

---

**Theorem 3** (Reduction from Dueling to Confidence-Set-Based Optimism). *If the confidence sets* $\mathcal{C}_{\mathcal{M}}(\mathcal{D}_t, \delta)$ *satisfy Assumption 2, then the dueling regret* $\mathrm{Regret}_D(T)$ *of Algorithm 1 is given by*

$$\mathrm{Regret}_D(T) = \widetilde{\mathcal{O}}(C_P(\mathcal{M}, T, \delta) + C_F(\overline{\mathcal{M}}, T, \delta))$$

Note that complexity parameter $C_F$ depends on $\overline{\mathcal{M}}$. It is a priori unclear how the complexity of $\overline{\mathcal{M}}$ relates to that of $\mathcal{M}$. Fortunately, Lemma 2 below settles this, and we can then use our results for POR-UCRL to get Corollary 1 below. See Appendix F.3 for a straightforward extension to general function approximation for $\mathcal{P}$, abstracting out the $S, A$ dependence.

**Lemma 2** (Relating $\mathcal{F}$ and $\overline{\mathcal{F}}$). *For any function class* $\mathcal{F}$, $\dim_E(\overline{\mathcal{F}}, \varepsilon) \leqslant 9 \dim_E(\mathcal{F}, \varepsilon/2)$.

**Corollary 1** (Dueling Regret using POR-UCRL Confidence Sets). *The confidence sets from POR-UCRL satisfy Assumption 2 and using them in Algorithm 1 leads to the following regret bound* $\mathrm{Regret}_D(T) = \widetilde{\mathcal{O}} \left( \left( pS\sqrt{HA} + \sum_{h \in \mathcal{H}_p} \sqrt{d_{E,h} d_{C,h}} \right) \sqrt{T} \right)$.

## 5 CONCLUSIONS AND FUTURE WORK

In this work, we have introduced PORMDPs and their analysis as a way to better model internal states of humans and intermediate feedback in RLHF. We have introduced two statistically efficient algorithms for handling partially observed reward-states and have shown that they improve over naive history summarization. We have noted that these methods subsume as well as improve over a lot of past work in RLHF. We have studied how one can further leverage a recursive structure over internal states using model-free methods. For this purpose, we have defined a new notion of dimension, the $\alpha$-HABE dimension, that captures the hardness of utilizing the recursive structure. Finally, we have also provided a novel reduction from dueling regret to optimistic algorithms for cardinal regret.

Besides our theoretical contributions, we would like to note the practical implications of our work.

- When the feedback is suspected to have a recurrent structure, we conclude in sections 3.1 and 3.2 that it can be exponentially more statistically efficient to use practical model-free methods like learning history-dependent Q-functions or using actor-critic methods. In the absence of such a structure, a model-based approach learning $f_\star$ and $\mathbb{P}_\star$ explicitly will also suffice.
- We note in section 3.2 that in practice, using a single network for the Q-function or critic across timesteps $h$ is important for the "exponential improvement" mentioned above, as opposed to using a different network for the Q-function or critic for each timestep.
- We are hoping that our work inspires new practical algorithms for PORRL. Theoretical advances in optimistic algorithms are known to inspire practical (e.g. perturbative) versions of the algorithms.
- The dueling to optimism reduction in section 4 can inspire future practical algorithms that achieve low dueling regret, which we have established in section 2.2 as an important metric for online (and online iterative) RLHF applications, like in Dong et al. (2024); Xiong et al. (2023).

We hope that our ideas lay the groundwork for further understanding of both statistical and algorithmic aspects of learning good policies when interacting with "stateful" feedback, such as that of humans. Our algorithms and proofs are presented in high generality and modularity in the appendix, and we hope that they can be used to provide novel algorithms and bounds in the future.

**Acknowledgement.** CK would like to acknowledge the support of the Rackham International Student Fellowship (the Indian Alumni Fellowship) for this work.

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

## A    Lemmas and Discussion

### A.1    Relation between $V_w$ and $V_g$

**Lemma 1** (Replacing $w$ with $g$). *For any history-dependent policy $\pi$ that selects an action $a_h \sim \pi(\tau[h-1], s_h)$, $V_w(\mathbb{M}, \pi) = V_g(\mathbb{M}, \pi)$ holds for any $\mathbb{M}$.*

*Proof.* By a slight abuse of notation, the following chain of equalities holds. Here, $(i)$ holds since $r_h(s_h, u_h, a_h)$ is a function of $s_h, u_h, a_h$. Equation $(ii)$ holds since we have already conditioned on $\tau[h]$, which includes $s_h, a_h$. Equation $(iii)$ holds by the definition of $g_h(\tau[h])$ as the conditional distribution of $u_h$ given $\tau[h]$.

$$
V_w(\mathbb{M}, \pi) = \mathbb{E}_{\tau^u \sim \mathbb{P}^{w,\pi}} \left[ \sum_{h \in \mathcal{H}_p} r_h(s_h, u_h, a_h) \right]
$$

$$
= \sum_{h \in \mathcal{H}_p} \mathbb{E}_{\tau^u \sim \mathbb{P}^{w,\pi}} \left[ r_h(s_h, u_h, a_h) \right]
$$

$$
= \sum_{h \in \mathcal{H}_p} \mathbb{E}_{\tau^u[h] \sim \mathbb{P}^{w,\pi}} \left[ r_h(s_h, u_h, a_h) \right]
$$

$$
= \sum_{h \in \mathcal{H}_p} \mathbb{E}_{\tau[h] \sim \mathbb{P}^{\pi}} \left[ \mathbb{E}_{\tau^u[h] \sim \mathbb{P}^{w,\pi}} \left[ r_h(s_h, u_h, a_h) \right] \Big| \tau[h] \right]
$$

$$
\overset{(i)}{=} \sum_{h \in \mathcal{H}_p} \mathbb{E}_{\tau[h] \sim \mathbb{P}^{\pi}} \left[ \mathbb{E}_{u_h, s_h, a_h \sim \mathbb{P}^{w,\pi}} \left[ r_h(s_h, u_h, a_h) \right] \Big| \tau[h] \right]
$$

$$
\overset{(ii)}{=} \sum_{h \in \mathcal{H}_p} \mathbb{E}_{\tau[h] \sim \mathbb{P}^{\pi}} \left[ \mathbb{E}_{u_h \sim \mathbb{P}^{w,\pi}} \left[ r_h(s_h, u_h, a_h) \right] \Big| \tau[h] \right]
$$

$$
\overset{(iii)}{=} \sum_{h \in \mathcal{H}_p} \mathbb{E}_{\tau[h] \sim \mathbb{P}^{\pi}} \left[ \mathbb{E}_{u_h \sim g_h(\tau[h])} \left[ r_h(s_h, u_h, a_h) \right] \right]
$$

$$
= \sum_{h \in \mathcal{H}_p} \mathbb{E}_{\tau \sim \mathbb{P}^{\pi}} \left[ \mathbb{E}_{u_h \sim g_h(\tau[h])} \left[ r_h(s_h, u_h, a_h) \right] \right]
$$

$$
= \mathbb{E}_{\tau \sim \mathbb{P}^{\pi}} \left[ \sum_{h \in \mathcal{H}_p} \mathbb{E}_{u_h \sim g_h(\tau[h])} \left[ r_h(s_h, u_h, a_h) \right] \right]
$$

$$
= V_g(\mathbb{M}, \pi)
$$

$\square$

### A.2    Ignoring Internal Reward-States is Bad for Alignment

We define traditional RL methods as those that output possibly time-dependent Markovian policies. In this section, we provide a toy example showing that there is a PORMDP with good sublinear regret guarantees where any time-dependent Markovian policy has value bounded away from the maximum value. This means that traditional RL methods will always incur linear regret. We hope that this illustrates that RL methods that ignore internal reward-states can be bad for alignment.

**Lemma 3** (Markovian policies are not enough). *There is a PORMDP where POR-UCRL and POR-UCBVI achieve $poly(H, S, A)\sqrt{T}$ regret, but any Markovian policy is at least $\frac{1}{4}$-suboptimal and so any method that outputs Markovian (possibly time-dependent) policies will lead to linear regret.*

*Proof.* Consider a PORMDP $\mathbb{M}$ in the setting of Chatterji et al. (2021) (see point (ii) below Definition 2) and set $\mathcal{S} = \{s_1, s_2\}$, $\mathcal{A} = \{a_1, a_2\}$ and $\mathbf{w} = 1 \in \mathbb{R}$. Let the transition matrix be $\mathbb{P}(s' \mid s, a) = \frac{1}{2}$ for all $s, a, s'$. Let the starting state always be $s_1$.

Consider the set $\mathcal{T}$ of all trajectories that have $a_2$ until $s_2$ appears, and then only have $a_1$. Choose $\phi(\tau) = \mathbb{1}(\tau \in \mathcal{T})$. The best non-Markovian policy $\pi_\star$ can follow this rule and achieve $\phi(\tau) = 1$ for all $\tau \sim \mathbb{P}^{\pi_\star}$. Thus, $\max_{\pi \in \Pi} V(\mathbb{M}, \pi) = 1$, where $\Pi$ is given by all history-dependent policies.

On the other hand, consider a Markovian but potentially time-dependent policy $\pi$. If $\pi(a_2) = 0$, then its value is zero. If $\pi(a_2) > 0$, then conditioned on the event that $s_2$ appears first at timestep 1, the expected total reward is at most $\pi_1(a_2)(1 - \pi_2(a_2))$. Conditioned on the event that $s_2$ appears first at timestep 2, the expected total reward is at most $\pi_1(a_2)\pi_2(a_2)$. Conditioned on seeing $s_2$ at or after $h = 3$, the expected total reward is certainly at most 1. Using these crude inequalities, we can bound the expected reward of a Markovian policy $\pi$ by

$$\frac{\pi_1(a_2)(1 - \pi_2(a_2))}{2} + \frac{\pi_1(a_2)\pi_2(a_2)}{4} + \sum_{h=3}^{H} \frac{1}{2^h} \leqslant \frac{\pi_1(a_1)(2 - \pi_2(a_2))}{4} + \frac{1}{4} \leqslant \frac{1}{2} + \frac{1}{4} = \frac{3}{4}$$

This means that the value of any time-dependent Markovian policy is at most $\frac{3}{4}$ and so any time-dependent Markovian policy is at least $\frac{1}{4}$-suboptimal and incurs $\frac{T}{4}$ regret. □

Recall that we defined traditional RL algorithms as those that output (possibly time-dependent) Markovian policies. Clearly, any traditional RL algorithm in this sense will have at least $\frac{T}{4}$ regret, which is linear regret.

### A.3 THE NAIVE REDUCTION FROM DUELING TO CARDINAL PORRL FAILS

**Lemma 4** (Naive Reduction Lower Bound). *Using any algorithm for cardinal PORRL with sublinear cardinal regret on $\overline{\mathbb{M}}$ with policy class $\Pi' := \Pi \times \Pi$ to get a sequence $(\pi_{1,1}, \pi_{2,1}), \ldots, (\pi_{1,T}, \pi_{2,T})$ leads to linear dueling regret for $\mathbb{M}$ whenever all policies $\pi$ do not have the same value $V(\mathbb{M}, \pi)$.*

*Proof.* Define $\pi_\star := \arg\max_{\pi \in \Pi} V(\mathbb{M}, \pi)$ and let $\pi_{\min} := \arg\min_{\pi \in \Pi} V(\mathbb{M}, \pi)$. Then note that

$$\max_{\pi, \pi' \in \Pi} V_D(\mathbb{M}, \pi, \pi') = \max_{\pi, \pi' \in \Pi} V(\mathbb{M}, \pi) - V(\mathbb{M}, \pi')$$
$$= \max_{\pi \in \Pi} V(\mathbb{M}, \pi) + \max_{\pi' \in \Pi} \left[ -V(\mathbb{M}, \pi') \right]$$
$$= \max_{\pi \in \Pi} V(\mathbb{M}, \pi) - \min_{\pi' \in \Pi} V(\mathbb{M}, \pi')$$
$$= V(\mathbb{M}, \pi_\star) - V(\mathbb{M}, \pi_{min})$$

Under the naive reduction described in Section 4, a cardinal PORRL algorithm is used to maximize *dueling feedback*. If the algorithm has sublinear cardinal regret, then it will produce duels $(\pi_{1,t}, \pi_{2,t}), t = 1 \to T$, satisfying

$$\sum_{t=1}^{T} \max_{\pi, \pi' \in \Pi} V_D(\mathbb{M}, \pi, \pi') - V_D(\mathbb{M}, \pi_{1,t}, \pi_{2,t}) = o(T)$$

From above, this means that

$$\sum_{t=1}^{T} \left[ V(\mathbb{M}, \pi_\star) - V(\mathbb{M}, \pi_{1,t}) \right] + \left[ V(\mathbb{M}, \pi_{2,t}) - V(\mathbb{M}, \pi_{min}) \right] = o(T)$$

Now note that by definition of $\pi_\star$ and $\pi_{min}$, both terms are positive. This is the key point. We thus have

$$\sum_{t=1}^{T} V(\mathbb{M}, \pi_\star) - V(\mathbb{M}, \pi_{1,t}) = o(T)$$
$$\sum_{t=1}^{T} V(\mathbb{M}, \pi_{2,t}) - V(\mathbb{M}, \pi_{min}) = o(T)$$

This means that for dueling regret $\text{Regret}_D(T)$, we have the following.

$$\text{Regret}_D(T) = \sum_{t=1}^{T} \left[ V(\mathbb{M}, \pi_\star) - V(\mathbb{M}, \pi_{1,t}) \right] + \left[ V(\mathbb{M}, \pi_\star) - V(\mathbb{M}, \pi_{2,t}) \right]$$
$$= \sum_{t=1}^{T} \left[ V(\mathbb{M}, \pi_\star) - V(\mathbb{M}, \pi_{1,t}) \right] + \left[ V(\mathbb{M}, \pi_\star) - V(\mathbb{M}, \pi_{min}) \right]$$

$$+ \sum_{t=1}^{T} \left[ V(\mathbb{M}, \pi_{min}) - V(\mathbb{M}, \pi_{2,t}) \right]$$
$$= o(T) + T \left[ V(\mathbb{M}, \pi_{\star}) - V(\mathbb{M}, \pi_{min}) \right]$$
$$= \Theta(T)$$

Where the last line holds since all policies $\pi$ do not have the same value $V(\mathbb{M}, \pi)$, and so $V(\mathbb{M}, \pi_{\star}) - V(\mathbb{M}, \pi_{min}) > 0$. $\qquad \square$

# B  REGRET-TO-PAC CONVERSION

When learning in MDPs, we can turn any guarantee on the regret into a corresponding PAC guarantee, the so-called "regret-to-PAC conversion" (Jin et al., 2018; Ménard et al., 2021; Wagenmaker et al., 2022; Tirinzoni et al., 2023). Similarly, we want to convert guarantees on the cardinal and dueling regret (see Section 2) into corresponding PAC guarantees, which are more adherent to an offline setting. We provide distinct results for the cardinal and dueling regret below.

**Lemma 5** (Cardinal regret to PAC). *For $T \in \mathbb{N}$ and $\delta \in [0, 1]$, let* ALG *be an algorithm for cardinal PORRL producing a sequence of policies $(\pi_t)_{t \in [T]}$ with cardinal regret bounded with probability at least $1 - \delta$ as*

$$\sum_{t=1}^{T} V(\mathbb{M}, \pi_\star) - V(\mathbb{M}, \pi_t) \leqslant R(T, \delta) \in \mathbb{R}.$$

*Then, a policy $\widehat{\pi}_T \sim \pi_1, \ldots, \pi_T$ sampled uniformly satisfies with probability at least $1 - 2\delta$*

$$V(\mathbb{M}, \pi_\star) - V(\mathbb{M}, \widehat{\pi}_T) \leqslant \frac{R(T, \delta)}{T} + 8Bp\sqrt{\frac{\log(1/\delta)}{T}}.$$

*Proof.* We consider the sequence of random variables $Y_t = V(\mathbb{M}, \pi_\star) - V(\mathbb{M}, \pi_t) \ \forall t \in [T]$. Through the Hoeffding's inequality on $Y_t$ and $|r_h| \leqslant B$ we have

$$V(\mathbb{M}, \pi_\star) - V(\mathbb{M}, \widehat{\pi}_T) = \mathbb{E}[V(\mathbb{M}, \pi_\star) - V(\mathbb{M}, \pi_t)]$$

$$\leqslant \frac{1}{T} \sum_{t=1}^{T} \left( V(\mathbb{M}, \pi_\star) - V(\mathbb{M}, \pi_t) \right) + 8Bp\sqrt{\frac{\log(1/\delta)}{T}}$$

with probability at least $1 - \delta$. Then, combining the latter inequality with the upper bound on the regret and a union bound, we get

$$V(\mathbb{M}, \pi_\star) - V(\mathbb{M}, \widehat{\pi}_T) \leqslant \frac{R(T, \delta)}{T} + 8Bp\sqrt{\frac{\log(1/\delta)}{T}}$$

with probability at least $1 - 2\delta$. $\qquad \square$

The latter result implies a PAC guarantee of the form $\mathbb{P}(V(\mathbb{M}, \pi_\star) - V(\mathbb{M}, \widehat{\pi}_T) \geqslant \varepsilon) \leqslant \delta$ for some $\varepsilon > 0$ and $\delta \in [0, 1]$ with a number of episodes of order $\widetilde{O}(1/\varepsilon^2)$. An analogous result can be stated for the dueling setting.

**Lemma 6** (Dueling regret to PAC). *For $T \in \mathbb{N}$ and $\delta \in [0, 1]$, let* ALG *be an algorithm for dueling PORRL producing a sequence of policy pairs $(\pi_{1,t}, \pi_{2,t})_{t \in [T]}$ with dueling regret bounded with probability at least $1 - \delta$ as*

$$\sum_{t=1}^{T} V(\mathbb{M}, \pi_\star) - \frac{V(\mathbb{M}, \pi_{1,t}) + V(\mathbb{M}, \pi_{2,t})}{2} \leqslant R_D(T, \delta) \in \mathbb{R}.$$

*Then, a policy $\widehat{\pi}_T \sim \pi_1, \ldots, \pi_T$ sampled uniformly satisfies with probability at least $1 - 4\delta$*

$$V(\mathbb{M}, \pi_\star) - V(\mathbb{M}, \widehat{\pi}_T) \leqslant \frac{R_D(T, \delta)}{T} + 16Bp\sqrt{\frac{\log(1/\delta)}{T}}.$$

*Proof.* The proof proceeds as in the previous lemma by applying Hoeffding separately on the sequences $Y_{1,t} = V(\mathbb{M}, \pi_\star) - V(\mathbb{M}, \pi_{1,t})$ and $Y_{2,t} = V(\mathbb{M}, \pi_\star) - V(\mathbb{M}, \pi_{2,t})$, then applying a union bound. $\qquad \square$

## C    PROOFS FOR GENERAL OPTIMISTIC ALGORITHMS FOR CARDINAL PORRL

### C.1    GENERIC MODEL-BASED OPTIMISM USING CONFIDENCE-SETS

We present a template to get regret bounds for a *generic model-based optimistic algorithm using confidence sets*, which we will later instantiate into POR-UCRL and also use in our reduction from the dueling PORRL to optimistic algorithms for cardinal PORRL.

A generic algorithm using confidence sets is determined by confidence sets $\mathcal{C}_{\mathcal{M}}(\mathcal{D}, \delta)$ based on a dataset $\mathcal{D}$. Maintaining a running dataset $\mathcal{D}_t$, at each step $t$, we run $\pi_t$ given by

$$\pi_t, \widetilde{\mathbb{M}}_t := \underset{\pi \in \Pi, \mathbb{M} \in \mathcal{C}_{\mathcal{M}}(\mathcal{D}_t, \delta)}{\arg\max} V(\mathbb{P}, f, \pi)$$

We obtain a trajectory $\tau_t \sim \mathbb{P}_\star^{\pi_t}$ and append it to $\mathcal{D}_t$ to get $\mathcal{D}_{t+1}$, recompute confidence sets $\mathcal{C}_{\mathcal{M}}(\mathcal{D}_{t+1}, \delta)$, and continue. This algorithm is formally presented in Appendix C.1 below.

---

**Algorithm 2** Generic Confidence-Set Optimism

---

1: **Input** Known family of reward functions $\{\mathcal{R}_h\}_{h=1}^H$, known model class $\mathcal{M}$ induced by known probability transition kernel class $\mathcal{P}$ and known decoder-induced function class $\mathcal{F}$, confidence level $\delta$.
2: **Initialize** dataset $\mathcal{D}_1 \leftarrow \{\}$ and $\mathcal{C}_{\mathcal{M}}(\mathcal{D}_1, \delta) \leftarrow \mathcal{M}$.
3: **for** $t = 1, ..., T$ **do**
4:     **Compute** the optimistic history dependent policy,

$$\pi_t, \widetilde{\mathbb{M}}_t = \underset{\pi, \ \mathbb{M} \in \mathcal{C}_{\mathcal{M}}(\mathcal{D}_t, \delta)}{\arg\max} V(\mathbb{M}, \pi)$$

5:     **Observe** trajectory $\tau_t = \{(s_h^t, a_h^t)\}_{h=1}^H$ and feedback $\{o_h\}_{h \in \mathcal{H}_p}$.
6:     **Update** $\mathcal{D}_{t+1} \leftarrow \mathcal{D}_t \cup \{\tau_t\}$ and compute new confidence set $\mathcal{C}_{\mathcal{M}}(\mathcal{D}_{t+1}, \delta)$.
7: **end for**

---

We now make the following assumption about our confidence sets. It essentially controls the effect of shrinking confidence sets for $\mathcal{P}$ and $\mathcal{F}$ on the value. Showing this assumption is the core of proving regret bounds for any instantiation of this generic algorithm. We will see later that it is satisfied by the confidence sets for POR-UCRL.

**Assumption 3** (Controlling Value Error due to Confidence Sets, Refined Version). For a transition kernel $\mathbb{P}_\star$ and function $f_\star$, consider any sequence of policies $\pi_t$ and datasets $\mathcal{D}_t$ that contain $\{\tau_i\}_{i=1}^t$ generated under $(\mathbb{P}_\star^{\pi_t}, f^\star)$. We require that $\mathbb{M}_\star \in \mathcal{C}_{\mathcal{M}}(\mathcal{D}_t, \delta)$ for all $t$ with probability $1 - \delta/16$. We require that there exist problem dependent functions $C_P(\mathcal{M}, T, \delta)$ and $C_F(\mathcal{M}, T, \delta)$ so that for arbitrary sequences $(\mathbb{P}_t, f^t) \in \mathcal{C}_{\mathcal{M}}(\mathcal{D}_t, \delta)$, the following hold with probability $1 - \delta/2$ each.

$$\left| \sum_{t=1}^T V(\mathbb{P}_t, f^t, \pi_t) - V(\mathbb{P}_\star, f^t, \pi_t) \right| = \widetilde{\mathcal{O}}(C_P(\mathcal{M}, T, \delta))$$

$$\left| \sum_{t=1}^T V(\mathbb{P}_\star, f^t, \pi_t) - V(\mathbb{P}_\star, f_\star, \pi_t) \right| = \widetilde{\mathcal{O}}(C_F(\mathcal{M}, T, \delta))$$

**Theorem 4** (Regret for Confidence-Set Optimism). *Under Assumption 3, any generic optimistic algorithm using confidence sets $\mathcal{C}_{\mathcal{M}}(\mathcal{D}, \delta)$ satisfies the regret bound*

$$\text{Regret}(T) = \widetilde{\mathcal{O}}\left(C_P(\mathcal{M}, T, \delta) + C_F(\mathcal{M}, T, \delta)\right)$$

*Proof.* Let $\widetilde{\mathbb{M}}_t$ be given by $\widetilde{\mathbb{P}}_t$ and $\widetilde{f}^t$. Note the following inequalities, where $(i)$ holds with probability 1 by the optimistic definition of $\pi_t$.

$$\text{Regret}(T) = \sum_{t=1}^T V(\mathbb{P}_\star, f^\star, \pi_\star) - V(\mathbb{P}_\star, f^\star, \pi_t)$$

$$\overset{(i)}{\leqslant} \sum_{t=1}^{T} V(\widetilde{\mathbb{P}}_t, \widetilde{f}^t, \pi_t) - V(\mathbb{P}_\star, f^\star, \pi_t)$$

$$\leqslant \sum_{t=1}^{T} \underbrace{V(\widetilde{\mathbb{P}}_t, \widetilde{f}^t, \pi_t) - V(\mathbb{P}_\star, \widetilde{f}^t, \pi_t)}_{(I)} + \underbrace{V(\mathbb{P}_\star, \widetilde{f}^t, \pi_t) - V(\mathbb{P}_\star, f^\star, \pi_t)}_{(II)}$$

We now apply Assumption 3 to bound $(I)$ and $(II)$. We can use the assumption since $\mathcal{D}_t$ contains trajectories $\{\tau_i\}_{i=1}^{t}$ generated by $\mathbb{P}_\star^{\pi_t}$, $\widetilde{f}^t \in \mathcal{C}_\mathcal{F}(\mathcal{D}_t, \delta) \subset \mathcal{F}$ and $\widetilde{\mathbb{P}}^t \in \mathcal{C}_\mathcal{F}(\mathcal{D}_t, \delta)$. This immediately gives us that with probability $1 - \delta$

$$\mathrm{Regret}(T) = \widetilde{\mathcal{O}}(C_F(\mathcal{M}, T, \delta) + C_P(\mathcal{M}, T, \delta))$$

as desired. $\qquad\square$

## C.2 GENERIC MODEL-BASED OPTIMISM USING BONUSES

We present a template to get regret bounds for a *generic model-based optimistic algorithm using bonuses*, which we will later instantiate into POR-UCBVI and also use in our reduction from the dueling PORRL to optimistic algorithms for cardinal PORRL.

A generic optimistic algorithm using bonuses relies on bonuses $b_{\mathcal{P}}^{\mathcal{D}}(\mathbb{P}, \pi, \delta), b_{\mathcal{F}}^{\mathcal{D}}(\mathbb{P}, \pi, \delta)$ that depend on a policy $\pi$, transition kernel $\mathbb{P}$ and dataset $\mathcal{D}$. It also relies on estimates $\hat{\mathbb{P}}_{\mathcal{D}}$ and $\hat{f}_{\mathcal{D}}$ that depend on $\mathcal{D}$. Maintaining a running dataset $\mathcal{D}_t$, at each step $t$, we run $\pi_t := \arg\max_{\pi \in \Pi} \tilde{V}(\hat{\mathbb{P}}_{\mathcal{D}_t}, \hat{f}_{\mathcal{D}_t}, \pi)$, where $\tilde{V}(\hat{\mathbb{P}}_{\mathcal{D}_t}, \hat{f}_{\mathcal{D}_t}, \pi)$ is given by:

$$V(\hat{\mathbb{P}}_{\mathcal{D}_t}, \hat{f}_{\mathcal{D}_t}, \pi) + b_{\mathcal{F}}^{\mathcal{D}_t}(\hat{\mathbb{P}}_{\mathcal{D}_t}, \pi, \delta) + z(Bp)b_{\mathcal{P}}^{\mathcal{D}_t}(\hat{\mathbb{P}}_{\mathcal{D}_t}, \pi, \delta)$$

where $z$ is defined below. We obtain a trajectory $\tau_t \sim \mathbb{P}_{\star}^{\pi_t}$ and append it to $\mathcal{D}_t$ to get $\mathcal{D}_{t+1}$, compute new bonuses and estimates, and continue. This algorithm is formally presented in Appendix C.2.

---

**Algorithm 3** Generic Bonus-Based Optimism

---

1: **Input** Known family of reward functions $\{\mathcal{R}_h\}_{h=1}^H$, method $\texttt{Est}(\mathcal{D})$ to estimate $\hat{\mathbb{P}}_{\mathcal{D}}$ and $\hat{f}_{\mathcal{D}}$ from dataset $\mathcal{D}$, bonus functions $b_{\mathcal{F}}^{\mathcal{D}}(\mathbb{P}, \pi, \delta)$ and $b_{\mathcal{P}}^{\mathcal{D}}(\mathbb{P}, \pi, \delta)$, confidence level $\delta$
2: **Initialize** $\mathcal{D}_1 \leftarrow \{\}$, initialize $\hat{f}^{\mathcal{D}_1}, \hat{\mathbb{P}}_{\mathcal{D}_1}$ arbitrarily.
3: **for** $t = 1, ..., T$ **do**
4:     **Compute** optimistic history dependent policy,

$$\pi_t = \arg\max_{\pi} V(\hat{\mathbb{P}}_{\mathcal{D}_t}, \hat{f}_{\mathcal{D}_t}, \pi) + b_{\mathcal{F}}^{\mathcal{D}_t}(\hat{\mathbb{P}}_{\mathcal{D}_t}, \pi, \delta) + z(Bp)(b_{\mathcal{P}}^{\mathcal{D}_t}(\hat{\mathbb{P}}_{\mathcal{D}_t}, \pi, \delta))$$

5:     **Observe** trajectory $\tau_t = \{(s_h^t, a_h^t)\}_{h=1}^H$ and feedback $\{o_h\}_{h \in \mathcal{H}_p}$.
6:     **Compute** new estimates $\hat{f}^{\mathcal{D}_{t+1}}, \hat{\mathbb{P}}^{\mathcal{D}_{t+1}} \leftarrow \texttt{Est}(\mathcal{D}_{t+1})$ and compute new bonus functions $b_{\mathcal{F}}^{\mathcal{D}_{t+1}}(\hat{f}^{\mathcal{D}_{t+1}}, \cdot, \delta), b_{\mathcal{P}}^{\mathcal{D}_{t+1}}(\hat{\mathbb{P}}^{\mathcal{D}_{t+1}}, \cdot, \delta)$.
7: **end for**

---

We now make the following assumption about our bonuses. Showing this assumption is the core of proving regret bounds for any instantiation of this generic algorithm. We will see later that it is satisfied by the bonuses for POR-UCBVI.

**Assumption 4** (Controlling Value Error via Bonuses). For a transition kernel $\mathbb{P}_{\star}$ and function $f_{\star}$, consider any sequence of policies $\pi_t$ and datasets $\mathcal{D}_t$ that contain $\{\tau_i\}_{i=1}^t$ generated under $(\mathbb{P}_{\star}^{\pi_t}, f^{\star})$. We require that for sequences $\hat{\mathbb{P}}_{\mathcal{D}_t}$ and $\hat{f}_{\mathcal{D}_t}$ and any sequence $f^t \in \mathcal{F}$, the following hold.

- *Bounding effect of error in $\mathcal{F}$:* With probability $1 - \delta/32$, for any $\mathbb{P}$ and uniformly over all policies $\pi$, $|V(\mathbb{P}, \hat{f}_{\mathcal{D}_t}, \pi) - V(\mathbb{P}, f^{\star}, \pi)| \leqslant b_{\mathcal{F}}^{\mathcal{D}_t}(\mathbb{P}, \pi, \delta)$ and there is a function $C_F(\mathcal{M}, T, \delta)$ so that $\sum_{t=1}^T b_{\mathcal{F}}^{\mathcal{D}_t}(\mathbb{P}_{\star}, \pi_t, \delta) = \tilde{\mathcal{O}}(C_F(\mathcal{M}, T, \delta))$ with probability $1 - \delta/32$

- *Bounding effect of error in $\mathcal{P}$:* For any function $\mu : \Gamma_H \to \mathbb{R}$ bounded by $D$, there is a function $z(D) \geqslant D$ so that the following holds uniformly over all policies $\pi$ with probability $1 - \delta/32$.

$$\mathbb{E}_{\tau \sim (\hat{\mathbb{P}}_{\mathcal{D}_t})^{\pi}} \mu(\tau) - \mathbb{E}_{\tau \sim \mathbb{P}_{\star}^{\pi}} \mu(\tau) \leqslant z(D)b_{\mathcal{P}}^{\mathcal{D}_t}(\mathbb{P}_{\star}, \pi, \delta)$$

  The statement also holds if we switch $\mathbb{P}_{\star}$ and $\hat{\mathbb{P}}_{\mathcal{D}_t}$. Additionally, the statement holds for a suitable $D$ if we replace $\mathbb{E}_{\tau \sim \mathbb{P}^{\pi}} \mu(\tau)$ with $b_{\mathcal{P}}(\mathbb{P}, \pi, \delta)$ or $b_{\mathcal{F}}(\mathbb{P}, \pi, \delta)$.[11] Finally, there is a function $C_P(\mathcal{M}, T, \delta)$ so that $\sum_{t=1}^T b_{\mathcal{P}}^{\mathcal{D}_t}(\mathbb{P}_{\star}, \pi_t, \delta) = \tilde{\mathcal{O}}(C_P(\mathcal{M}, T, \delta))$ with probability $1 - \delta/32$.

**Theorem 5** (Regret for Bonus-Based Optimism). *Under Assumption 4, with $z_1(D) = z(D) + z(2D) + z(2z(D))$, any generic optimistic algorithm using bonuses satisfies*

$$\text{Regret}(T) = \tilde{\mathcal{O}}\left(C_F(\mathcal{M}, T, \delta) + z_1(Bp)C_P(\mathcal{M}, T, \delta)\right)$$

---

[11]This would instantly hold with $D = Bp$ if $b_{\mathcal{F}}(\mathbb{P}, \pi, \delta) := \mathbb{E}_{\tau \sim \mathbb{P}^{\pi}} b_{\mathcal{F}}(\tau, \delta)$ for some trajectory level bonus $b_{\mathcal{F}}(\tau, \delta)$, and similarly for $\mathcal{P}$.

*Proof.* Note that we can use Assumption 4 since $\mathcal{D}_t$ contains trajectories $\{\tau_i\}_{i=1}^t$ generated by $\mathbb{P}_\star^{\pi_t}$, $\hat{f}_{\mathcal{D}_t} \in \mathcal{F}$ is computed using $\mathcal{D}_t$ and $\hat{\mathbb{P}}_{\mathcal{D}_t}$ is computed using $\mathcal{D}_t$. Also note that WLOG, $b_{\mathcal{P}}^{\mathcal{D}_t}(\mathbb{P}, \pi, \delta) \leqslant 2$ always holds since we can otherwise clip it at 2 and our assumption will still hold. Similarly, WLOG $b_{\mathcal{F}}^{\mathcal{D}_t}(\mathbb{P}, \pi, \delta) \leqslant 2z(Bp)$, otherwise we can clip it at 1 and our assumption will still hold. Now note the following inequalities.

$$\text{Regret}(T) = \sum_{t=1}^T V(\mathbb{P}_\star, f^\star, \pi_\star) - V(\mathbb{P}_\star, f^\star, \pi_t)$$

$$\overset{(i)}{\leqslant} \sum_{t=1}^T V(\hat{\mathbb{P}}_{\mathcal{D}_t}, f^\star, \pi_\star) + z(Bp)(b_{\mathcal{P}}^{\mathcal{D}_t}(\hat{\mathbb{P}}_{\mathcal{D}_t}, \pi_\star, \delta)) - V(\mathbb{P}_\star, f^\star, \pi_t)$$

$$\overset{(ii)}{\leqslant} \sum_{t=1}^T V(\hat{\mathbb{P}}_{\mathcal{D}_t}, \hat{f}_{\mathcal{D}_t}, \pi_\star) + b_{\mathcal{F}}^{\mathcal{D}_t}(\hat{\mathbb{P}}_{\mathcal{D}_t}, \pi_\star, \delta) + z(Bp)(b_{\mathcal{P}}^{\mathcal{D}_t}(\hat{\mathbb{P}}_{\mathcal{D}_t}, \pi_\star, \delta)) - V(\mathbb{P}_\star, f^\star, \pi_t)$$

$$\overset{(iii)}{\leqslant} \sum_{t=1}^T V(\hat{\mathbb{P}}_{\mathcal{D}_t}, \hat{f}_{\mathcal{D}_t}, \pi_t) + b_{\mathcal{F}}^{\mathcal{D}_t}(\hat{\mathbb{P}}_{\mathcal{D}_t}, \pi_t, \delta) + z(Bp)(b_{\mathcal{P}}^{\mathcal{D}_t}(\hat{\mathbb{P}}_{\mathcal{D}_t}, \pi_t, \delta)) - V(\mathbb{P}_\star, f^\star, \pi_t)$$

$$= \sum_{t=1}^T V(\hat{\mathbb{P}}_{\mathcal{D}_t}, \hat{f}_{\mathcal{D}_t}, \pi_\star) - V(\mathbb{P}_\star, f^\star, \pi_t) + b_{\mathcal{F}}^{\mathcal{D}_t}(\hat{\mathbb{P}}_{\mathcal{D}_t}, \pi_t, \delta) + z(Bp)(b_{\mathcal{P}}^{\mathcal{D}_t}(\hat{\mathbb{P}}_{\mathcal{D}_t}, \pi_t, \delta))$$

$$= \sum_{t=1}^T V(\hat{\mathbb{P}}_{\mathcal{D}_t}, \hat{f}_{\mathcal{D}_t}, \pi_\star) - V(\hat{\mathbb{P}}_{\mathcal{D}_t}, f^\star, \pi_\star) + V(\hat{\mathbb{P}}_{\mathcal{D}_t}, f^\star, \pi_\star) - V(\mathbb{P}_\star, f^\star, \pi_t)$$

$$+ \sum_{t=1}^T b_{\mathcal{F}}^{\mathcal{D}_t}(\hat{\mathbb{P}}_{\mathcal{D}_t}, \pi_t, \delta) + z(Bp)(b_{\mathcal{P}}^{\mathcal{D}_t}(\hat{\mathbb{P}}_{\mathcal{D}_t}, \pi_t, \delta))$$

Here, inequality $(i)$ holds with probability $1 - \delta/16$ by the second point in Assumption 4. Inequality $(ii)$ holds with probability $1 - \delta/16$ by the first point in Assumption 4. Inequality $(iii)$ holds with probability 1 by the optimistic definition of $\pi_t$. Continuing, we have

$$\text{Regret}(T) \overset{(iv)}{\leqslant} 2 \sum_{t=1}^T b_{\mathcal{F}}^{\mathcal{D}_t}(\hat{\mathbb{P}}_{\mathcal{D}_t}, \pi_t, \delta) + z(Bp)(b_{\mathcal{P}}^{\mathcal{D}_t}(\hat{\mathbb{P}}_{\mathcal{D}_t}, \pi_t, \delta))$$

$$\overset{(v)}{\leqslant} 2 \sum_{t=1}^T b_{\mathcal{F}}^{\mathcal{D}_t}(\mathbb{P}_\star, \pi_t, \delta) + 2z(Bp)(b_{\mathcal{P}}^{\mathcal{D}_t}(\mathbb{P}_\star, \pi_t, \delta)) + z(Bp)(b_{\mathcal{P}}^{\mathcal{D}_t}(\mathbb{P}_\star, \pi_t, \delta))$$

$$+ 2z(z(Bp))(b_{\mathcal{P}}^{\mathcal{D}_t}(\mathbb{P}_\star, \pi_t, \delta))$$

$$= \mathcal{O}\left(\sum_{t=1}^T b_{\mathcal{F}}^{\mathcal{D}_t}(\mathbb{P}_\star, \pi_t, \delta) + z_1(Bp)(b_{\mathcal{P}}^{\mathcal{D}_t}(\mathbb{P}_\star, \pi_t, \delta))\right)$$

Here, inequality $(iv)$ holds with probability $1 - \delta/8$ by a union bound over the first and the second point in Assumption 4. Finally, inequality $(v)$ holds with probability $1 - \delta/4$ by a union bound over four applications of the second point of Assumption 4. Finally, we use a union bound over both points of Assumption 4 to conclude that with probability $1 - \delta/8$

$$\mathcal{O}\left(\sum_{t=1}^T b_{\mathcal{F}}^{\mathcal{D}_t}(\mathbb{P}_\star, \pi_t, \delta) + z_1(Bp)(b_{\mathcal{P}}^{\mathcal{D}_t}(\mathbb{P}_\star, \pi_t, \delta))\right) = \widetilde{\mathcal{O}}\left(C_F(\mathbb{M}, T, \delta) + z_1(Bp)C_P(\mathbb{M}, T, \delta)\right)$$

By taking a union bound over the events of all inequalities above, we have that with probability $1 - \delta$

$$\text{Regret}(T) = \widetilde{\mathcal{O}}\left(C_F(\mathbb{M}, T, \delta) + z_1(Bp)C_P(\mathbb{M}, T, \delta)\right)$$

as desired. $\qquad\square$

## C.3 GENERIC MODEL-FREE OPTIMISM

---

**Algorithm 4** Generic Model-Free Optimism

---

1: **Input** Known Bellman-complete class of Q-functions $\mathcal{Q}$, confidence level $\delta$.
2: **Initialize** dataset $\mathcal{D}_1 \leftarrow \{\}$ and $\mathcal{C}_{\mathcal{Q}}(\mathcal{D}_1, \delta) \leftarrow \mathcal{Q}$.
3: **for** $t = 1, ..., T$ **do**
4: $\quad \tau[0] \leftarrow ()$
5: $\quad$ **for** $h = 1, \ldots H$ **do**
6: $\quad\quad$ **Play** $a_h^t, Q_h^t \leftarrow \arg\max_{a, Q \in \mathcal{C}_{\mathcal{Q}}(\mathcal{D}_t, \delta)} Q_h(\tau[h], a)$ and observe feedback $o_h^t$
7: $\quad$ **end for**
8: $\quad$ **Update** $\mathcal{D}_{t+1} \leftarrow \mathcal{D}_t \cup \{\tau, (o_1^t, \ldots o_H^t\}$
9: $\quad$ **Compute** $\mathcal{C}_{\mathcal{Q}}(\mathcal{D}_{t+1}, \delta)$
10: **end for**

---

Note that the method for choosing actions $a_h^t$ at time $t$ induces a history dependent policy $\pi_t$, whose suboptimality is what we use to define regret. Regret is still given by

$$\text{Regret}(T) = \sum_{t=1}^{T} V(\mathbb{M}_\star, \pi_\star) - V(\mathbb{M}_\star, \pi_t)$$

We now make the following assumption about our confidence sets. Showing this assumption is the core of proving regret bounds for any instantiation of this generic algorithm. We know that this is satisfied by GOLF using the BE-dimension. We will show that in our case, it is also satisfied by a more refined notion known as the $\alpha$-HABE dimension (the $\alpha$-history aware Bellman eluder dimension).

**Assumption 5.** For a Q-function $Q^\star$ induced by model $\mathbb{M}_\star$, consider any sequence of policies $\pi_t$ and datasets $\mathcal{D}_t$ that contain $\{\tau_i\}_{i=1}^t$ generated under $\mathbb{M}_\star$. We require that $Q^\star \in \mathcal{C}_{\mathcal{Q}}(\mathcal{D}_t, \delta)$ for all $t$ with probability $1 - \delta/16$. We require that there exists a problem dependent function $C_Q(\mathcal{Q}, T, \delta)$, so that for arbitrary sequences $Q^t \in \mathcal{C}_{\mathcal{Q}}(\mathcal{D}_t, \delta)$, the following holds for all $h$ with probability $1 - \delta/2$.

$$\sum_{j=1}^{t} |\mathbb{E}_{\mu_h(Q^t)}[Q_h^t - \mathcal{T}_h Q_h^{j+1}]| \leqslant C_Q(\mathcal{Q}, T, \delta)$$

**Theorem 6** (Regret for Generic Model-Free Optimism). *If the confidence sets $\mathcal{C}_{\mathcal{Q}}(\mathcal{D}, \delta)$ used in Algorithm 4 satisfy Assumption 5, then the regret of Algorithm 4 is bounded by*

$$\text{Regret}(T) = O\left(H C_Q(\mathcal{Q}, T, \delta)\right)$$

*Proof.* Note that $V(\mathbb{M}_\star, \pi_\star) = \max_a Q_1^\star(s_1, a) \leqslant \max_a Q_1^t(s_1, a)$ for all $t$, giving us the following result by the policy loss decomposition in Jiang et al. (2016).

$$
\begin{aligned}
\text{Regret}(T) &= \sum_{t=1}^{T} V(\mathbb{M}_\star, \pi_\star) - V(\mathbb{M}_\star, \pi_t) \\
&\leqslant \sum_{t=1}^{T} \max_a Q_1^t(s_1, a) - V(\mathbb{M}_\star, \pi_t) \\
&= \sum_{t=1}^{T} \sum_{h=1}^{H} \mathbb{E}_{\mu_h(Q^t)}[Q_h^t - \mathcal{T}_h Q_h^{j+1}] \\
&= \sum_{h=1}^{H} \sum_{t=1}^{T} \mathbb{E}_{\mu_h(Q^t)}[Q_h^t - \mathcal{T}_h Q_h^{j+1}] \\
&= O(H C_Q(\mathcal{Q}, T, \delta))
\end{aligned}
$$

where the last line holds by Assumption 5. □

# D    DETAILS AND PROOFS FOR CARDINAL POR-UCRL

We now instantiate Algorithm 2 using standard confidence sets to get POR-UCRL. We show that they satisfy Assumption 3 and get regret bounds for the algorithm. Note that our algorithm is **crucially different** from naively summarizing the history to define a modified state space, since we are separating the use of history summarization for getting confidence sets $f$ from using only the current state while learning the Markovian transitions $\mathbb{P}$. In this case, it is a priori unclear if we can use ideas from optimism to prove guarantees with a favorable (non-exponential) dependence on the complexity of transitions.

Recall that given a dataset of the first $t$ trajectory samples $\{\tau_i\}_{i=1}^t$ and an index $h \in [H]$, we consider the following least squares objective to estimate $f$:

$$\widehat{f}_h^{t+1} = \arg\min_{f_h \in \mathcal{F}_h} \sum_{i=1}^t \left(\sigma_h(f_h(\tau_i[h])) - o_h^i\right)^2$$

Simple least squares guarantees imply the lemma below.

**Lemma 7** (Concentration for $\sigma \circ f_h$). *Define*

$$\mathrm{MSE}_{h,t}(f_h, f_h') := \sum_{i=1}^t \left(\sigma_h(f_h(\tau_i[h])) - \sigma_h(f_h'(\tau_i[h]))\right)^2$$

*Also define $\bar{\beta}_{h,t}(\delta) = \eta_h^2 \log\left(\frac{N\left(\mathcal{F}_h, \frac{B}{T}, \|\cdot\|_\infty\right)}{\delta}\right) + \alpha_{h,t}$ with $\alpha_{h,t} := \frac{tB + t\eta_h \log\left(\frac{t}{\delta}\right)}{T}$. Then $f_h^\star$ simultaneously satisfies $\mathrm{MSE}_{h,t}(f_h^\star, \widehat{f}_h^{(t+1)}) \leqslant \bar{\beta}_{h,t}\left(\frac{\delta}{2t^2 H}\right)$ for all $h, t$ with probability $1 - \delta/32$.*

*Proof.* We apply Lemma 6 in Chan et al. (2021) and the last statement in its proof to each $h$ separately with the function class in the lemma set to $\{\sigma_h \circ f_h | f_h \in \mathcal{F}_h\}$, $P = 1$, $\mathbf{x}_{t,p} = \mathbf{x}_{t,1} = \tau_t[h]$ and misspecification $\varepsilon = 0$ (decoupled from the Eluder dimension's $\varepsilon$). We also note that $o_h^t$ are $\eta_h$-subgaussian samples with mean $\sigma_h(f_h(\tau[h]))$. This gives us that each of event indexed by $h, t$ below holds with probability at least $1 - \frac{\delta}{2t^2 H}$.

$$\sum_{i=1}^t \left(\sigma_h(f_h(\tau_i[h])) - \sigma_h(f_h'(\tau_i[h]))\right)^2 \leqslant \bar{\beta}_{h,t}\left(\frac{\delta}{2t^2 H}\right)$$

So, the events all simultaneously hold with probability at least $1 - \delta$ by a union bound. $\qquad\square$

Recall the definition of our confidence sets below.

**Confidence Sets for POR-UCRL.**    We instantiate the generic optimistic algorithm using confidence sets by defining $\mathcal{C}_\mathcal{M}(\mathcal{D}_t, \delta) := \mathcal{C}_\mathcal{P}^t(\delta) \times \mathcal{C}_\mathcal{F}^t(\delta)$ as our confidence sets below. We name the resulting algorithm POR-UCRL. We use the data from trajectories $\{\tau_i\}_{i=1}^t$ to build the confidence sets $\mathcal{C}_\mathcal{F}^{t+1}(\delta) = \prod_h \mathcal{C}_h^{t+1}(\delta)$ with $\mathcal{C}_h^{t+1}(\delta)$ defined below, where $\beta_{h,t}(\delta) := \bar{\beta}_{h,t}\left(\frac{\delta}{2t^2 H}\right)$.

$$\mathcal{C}_h^{t+1}(\delta) := \left\{f_h \in \mathcal{F}_h \middle| \mathrm{MSE}_{h,t}(f_h^\star, \widehat{f}_h^{(t+1)}) \leqslant \beta_{h,t}(\delta)\right\}$$

We also use the MLE estimate for $\mathbb{P}$ after $t$ episodes to define $\hat{\mathbb{P}}^t(\cdot \mid s, a) := \frac{N_t(s,a,s')}{N_t(s,a)}$. Now for $\zeta(n, \delta) = 2\sqrt{\frac{S \log(2) + \log(n(n+1)SA/\delta)}{2n}}$, define $\mathcal{C}_\mathbb{P}^t(\delta)$ as below:

$$\left\{\mathbb{P} \middle| \|\mathbb{P}(\cdot \mid s, a) - \hat{\mathbb{P}}_t(\cdot \mid s, a)\|_1 \leqslant \zeta(N_t(s,a), \delta) \forall s, a\right\}$$

**Confidence Sets for POR-UCRL in case $\mathbb{P}_\star$ is known.**    For known-model UCRL, the confidence sets $\mathcal{C}_\mathcal{F}^t(\delta)$ are still as above, but $\mathcal{C}_\mathcal{P}^t(\delta) := \{\mathbb{P}_\star\}$

For completeness, we repeat the algorithm POR-UCRL here, which is an instantiation of Algorithm 2, the generic optimistic algorithm using confidence sets.

---

**Algorithm 5** POR-UCRL

---

1: **Input:** Known family of reward functions $\{\mathcal{R}_h\}_{h=1}^H$, known probability transition kernel class $\mathcal{P}$ and known decoder-induced function class $\mathcal{F}$, confidence level $\delta$.

2: **Initialize** dataset $\mathcal{D}_1 \leftarrow \{\}$ and $\mathcal{C}_{\mathcal{F}}(\mathcal{D}_1, \delta) \leftarrow \prod_{h=1}^H \mathcal{F}_h, \mathcal{C}_{\mathcal{P}}(\mathcal{D}_1, \delta) \leftarrow \mathcal{P}$.

3: **for** $t = 1, ..., T$ **do**

4:     **Compute** the optimistic history dependent policy,

$$\pi_t, \widetilde{f}_t, \widetilde{\mathcal{P}}_t = \underset{\pi, \ \mathcal{F} \in \mathcal{C}_{\mathcal{F}}(\mathcal{D}_t, \delta), \mathbb{P} \in \mathcal{P}(\mathcal{D}_t, \delta)}{\arg\max} V(\mathbb{P}, f, \pi)$$

5:     **Collect** trajectory $\tau_t = \{(s_h^t, a_h^t)\}_{h=1}^H$ and feedback $\{o_h\}_{h \in \mathcal{H}_p}$ by sampling from $\mathbb{P}_{\star}^{\pi_t}$ with true decoder-induced function $f_{\star}$.

6:     **Update** $\mathcal{D}_{t+1} \leftarrow \mathcal{D}_t \cup \{\tau_t\}, \hat{\mathbb{P}}_{t+1}, \hat{f}_h^{t+1}$ for all $h$

7:     **Compute** new confidence sets $\mathcal{C}_{\mathcal{F}}(\mathcal{D}_{t+1}, \delta) \leftarrow \prod_{h=1}^H \mathcal{C}_h^{t+1}(\delta)$ and $\mathcal{C}_{\mathcal{P}}(\mathcal{D}_{t+1}, \delta)$ where

$$\mathcal{C}_h^{t+1}(\delta) \leftarrow \left\{ f_h \in \mathcal{F}_h \middle| \mathrm{MSE}_{h,t}(f_h^{\star}, \hat{f}_h^{(t+1)}) \leqslant \beta_{h,t}(\delta) \right\}$$

$$\mathcal{C}_{\mathcal{P}}(\mathcal{D}_{t+1}, \delta) \leftarrow \left\{ \mathbb{P} \middle| \|\mathbb{P}(\cdot \mid s, a) - \hat{\mathbb{P}}_{t+1}(\cdot \mid s, a)\|_1 \leqslant \zeta(N_{t+1}(s,a), \delta) \forall s, a \right\}$$

8: **end for**

---

We will now show our regret bound.

**Theorem 1** (POR-UCRL Regret). *Under Assumption 1, the regret* $\mathrm{Regret}(T)$ *of POR-UCRL is bounded by the following with probability at least* $1 - \delta$

$$\widetilde{\mathcal{O}}\left( \left( pS\sqrt{HA} + \sum_{h \in \mathcal{H}_p} \sqrt{d_{E,h} d_{C,h}} \right) \sqrt{T} \right)$$

*where* $d_{E,h} = \dim_E\left(\mathcal{F}_h, \frac{B}{T}\right)$ *and* $d_{C,h} = \log(\mathcal{N}(\mathcal{F}_h, 1/T, \|\cdot\|_{\infty}))$.

### D.1 SHOWING THAT ASSUMPTION 3 IS SATISFIED

#### D.1.1 BOUNDING REWARD MODEL DEVIATIONS

**Lemma 8** (Bounding Reward Model Deviations). *Consider decoder-induced functions* $\{f_h\}_{h \in \mathcal{H}_p}$ *satisfying* $|f_h| \leqslant B$ *that induce value functions* $V(\mathbb{P}, f, \pi)$. *For any sequence of policies* $\pi_t$, *if the confidence* $\mathcal{C}_{\mathcal{F}}^t(\delta)$ *is generated using data* $\tau_i \sim \mathbb{P}_{\star}^{\pi_i}, i = 1 \to t$ *and* $\widetilde{f}^t \in \mathcal{C}_{\mathcal{F}}^t(\delta)$ *is an arbitrary sequence of functions, then we have the following with probability* $1 - \delta/4$.

$$\left| \sum_{t=1}^T V(\mathbb{P}_{\star}, \widetilde{f}^t, \pi_t) - V(\mathbb{P}_{\star}, f^{\star}, \pi_t) \right|$$

*is bounded by*

$$\mathcal{O}\left( Bp\sqrt{T \log(T/\delta)} + \sum_{h \in \mathcal{H}_p} B\kappa_{2,h} d_{E,h} + \sum_{h \in \mathcal{H}_p} \kappa_{2,h} \sqrt{d_{E,h} \beta_{h,T}(\delta) T} \right)$$

*Proof.*

$$
\begin{aligned}
\sum_{t=1}^T V(\mathbb{P}_{\star}, \widetilde{f}^t, \pi_t) - V(\mathbb{P}_{\star}, f^{\star}, \pi_t) &= \sum_{t=1}^T \mathbb{E}_{\tau \sim \mathbb{P}_{\star}^{\pi_t}}\left[ \sum_{h=1}^H \widetilde{f}_h^t(\tau[h]) \right] - \mathbb{E}_{\tau \sim \mathbb{P}_{\star}^{\pi_t}}\left[ \sum_{h=1}^H f_h^{\star}(\tau[h]) \right] \\
&= \sum_{t=1}^T \mathbb{E}_{\tau \sim \mathbb{P}_{\star}^{\pi_t}}\left[ \sum_{h \in \mathcal{H}_p} \widetilde{f}_h^t(\tau[h]) \right] - \mathbb{E}_{\tau \sim \mathbb{P}_{\star}^{\pi_t}}\left[ \sum_{h \in \mathcal{H}_p} f_h^{\star}(\tau[h]) \right] \\
&= \sum_{t=1}^T \mathbb{E}_{\tau \sim \mathbb{P}_{\star}^{\pi_t}}\left[ \sum_{h \in \mathcal{H}_p} \widetilde{f}_h^t(\tau[h]) - f_h^{\star}(\tau[h]) \right]
\end{aligned}
$$

$$= \sum_{t=1}^{T} \left[ \sum_{h \in \mathcal{H}_p} \widetilde{f}_h^t(\tau_t[h]) - f_h^\star(\tau_t[h]) + X_{1,t} + X_{2,t} \right]$$

$$\overset{(ii)}{\leq} \sum_{t=1}^{T} \left[ \sum_{h \in \mathcal{H}_p} \widetilde{f}_h^t(\tau_t[h]) - f_h^\star(\tau_t[h]) \right] + \mathcal{O}\left( Bp\sqrt{T\log(T/\delta)} \right)$$

where

$$X_{1,t} := \mathbb{E}_{\tau \sim \mathbb{P}^\pi} \left[ \sum_{h \in \mathcal{H}_p} \widetilde{f}_h^t(\tau[h]) \right] - \left[ \sum_{h \in \mathcal{H}_p} \widetilde{f}_h^t(\tau_t[h]) \right]$$

$$X_{2,t} := \left[ \sum_{h \in \mathcal{H}_p} f_h^\star(\tau_t[h]) \right] - \mathbb{E}_{\tau \sim \mathbb{P}^\pi} \left[ \sum_{h \in \mathcal{H}_p} f_h^\star(\tau[h]) \right]$$

Inequality $(i)$ follows by the definition of $\pi_t$ and $\widetilde{f}_h^t$ – that is, by optimism. Inequality $(ii)$ holds with probability at least $1 - \delta$ since $X_{1,t}$ and $X_{2,t}$ are both martingales with respect to the filtration $\mathcal{G}_t$ given by the data of trajectories $\{\tau_s\}_{s=1}^{t-1}$. Also, $|X_{1,t}|, |X_{2,t}| \leq Bp$. We can thus apply the Azuma-Hoeffding inequality twice to obtain inequality $(ii)$.

Continuing, note the following.

$$\sum_{t=1}^{T} V(\mathbb{P}_\star, \widetilde{f}^t, \pi_t) - V(\mathbb{P}_\star, f^\star, \pi_t)$$

$$\leq \left[ \sum_{h=1}^{H} f_h^t(\tau_t[h]) - f_h^\star(\tau_t[h]) \right] + Bp\sqrt{T\log(T/\delta)}$$

$$\leq \left[ \sum_{h=1}^{H} \kappa_{2,h}\sigma_h(f_h^t(\tau_t[h])) - \kappa_{2,h}\sigma_h(f_h^\star(\tau_t[h])) \right] + Bp\sqrt{T\log(T/\delta)}$$

$$\leq \kappa_{2,h} \sum_{t=1}^{T} \sum_{h \in \mathcal{H}_p} \underbrace{\max_{f_h, f_h' \in \mathcal{W}_h^t(\delta)} \sigma_h(f_h(\tau_t[h])) - \sigma_h(f_h'(\tau_t[h]))}_{=: \bar{\gamma}_{h,t}(\tau_t[h], \delta)} + Bp\sqrt{T\log(T/\delta)}$$

$$= \kappa_{2,h} \sum_{h \in \mathcal{H}_p} \left[ \sum_{t=1}^{T} \bar{\gamma}_{h,t}(\tau_t[h], \delta) \right] + Bp\sqrt{T\log(T/\delta)}$$

The sum of these maximum uncertainty evaluations can be upper bounded using the Eluder dimension. The inequality below holds by applying Lemma 3 in (Chan et al., 2021) for each $h$ separately, with the function class in the lemma set to $\{\sigma_h \circ f_h | f_h \in \mathcal{F}_h\}$, $P = 1$, $\mathbf{x}_{t,p} = \mathbf{x}_{t,1} = \tau_t[h]$ and misspecification $\varepsilon = 0$ (decoupled from the Eluder dimension's $\varepsilon$). We also recall that $o_h^t$ are $\eta_h$-subgaussian samples with mean $\sigma_h(f_h(\tau_t[h]))$. We obtain

$$\sum_{t=1}^{T} \bar{\gamma}_{h,t}(\tau_t[h], \delta) \leq \mathcal{O}\left( Bd_{E,h} + \sqrt{d_{E,h}\beta_{h,T}(\delta)T} \right)$$

Where $d_{E,h} = \dim_E\left(\mathcal{F}_h, \frac{B}{T}\right)$ is the Eluder dimension of $\mathcal{F}_h$ and $\beta_{h,T}(\delta) = \bar{\beta}_{h,t}\left(\frac{\delta}{2t^2H}\right)$. Therefore, we have our result.

$$\sum_{t=1}^{T} V(\mathbb{P}_\star, \widetilde{f}^t, \pi_t) - V(\mathbb{P}_\star, f^\star, \pi_t)$$

is bounded by

$$\mathcal{O}\left( \sum_{h \in \mathcal{H}_p} B\kappa_{2,h}d_{E,h} + \sum_{h \in \mathcal{H}_p} \kappa_{2,h}\sqrt{d_{E,h}\beta_{h,T}(\delta)T} + Bp\sqrt{T\log(T/\delta)} \right)$$

Note that this entire argument can be repeated with $f_\star$ and $\tilde{f}^t$ switched, by the symmetry of the definition of $\bar{\gamma}_{h,t}(\tau_t[h], \delta)$ and the fact that the negative of a martingale is also a martingale. □

### D.1.2 BOUNDING PROBABILITY MODEL DEVIATIONS

**Lemma 9** (Bounding Probability Model Deviations). *Consider an arbitrary sequence of functions $f^t \in \mathcal{F}$ satisfying $|f_h| \leq B$ that induce value functions $V(\mathbb{P}, f, \pi)$. For any sequence of policies $\pi_t$, if the confidence $\mathcal{C}_{\mathcal{P}}^t(\delta)$ is generated using data that includes $\tau_i \sim \mathbb{P}_\star^{\pi_i}, i = 1 \to t$ and $\tilde{\mathbb{P}}_t \in \mathcal{C}_{\mathcal{P}}^t(\delta)$ is an arbitrary sequence of transition structures, then we have the following with probability $1 - \delta/4$.*

$$\left| \sum_{t=1}^{T} V(\tilde{\mathbb{P}}_t, f^t, \pi_t) - V(\mathbb{P}_\star, f^t, \pi_t) \right| \leq \mathcal{O}\left( c_\delta B p \sqrt{SAHT} + c_\delta B p H S A + B p \sqrt{pT \log(T/\delta)} \right)$$

*where $c_\delta := 8\sqrt{S \log(2) + \log(HTSA/\delta)}$.*

*Proof.* We first show the following.

**Lemma 10.**

$$\sum_{t=1}^{T} V(\tilde{\mathbb{P}}_t, f^t, \pi_t) - V(\mathbb{P}_\star, f^t, \pi_t) \leq \sum_{t=1}^{T} \sum_{h=1}^{H} 2p\zeta(N_t(s_h^t, a_h^t), \delta) + \mathcal{O}\left( B p \sqrt{pT \log(T/\delta)} \right)$$

Recall that we denote the Bellman operator by $\mathcal{T}^\pi$ where $\mathcal{T}^\pi f = \mathbb{E}_{a \sim \pi} \mathbb{P} f$. Momentarily define the following for $\tau = (\tau_{l-1}, s_l, \tau')$, where $\tau_{l-1}$ is an arbitrary trajectory of length $l - 1 \leq H$, and $s_l$ is an arbitrary state.

$$V_{l,\mathbb{P}}^t(\tau_{l-1}, s_l) := \mathbb{E}_{\tau' \sim \mathbb{P}^{\pi_t}} \left[ \sum_{h=l}^{H} \tilde{f}_h^t(\tau[h]) \right]$$

$$= \mathbb{E}_{\tau' \sim \mathbb{P}^{\pi_t}} \left[ \sum_{h \in \mathcal{H}_p, h \geq l} \tilde{f}_h^t(\tau[h]) \right]$$

So in the definition above, the first $l - 1$ observations in $\tau$ come from $\tau_{l-1}$ while the rest are generated by the input $\mathbb{P}$ starting with state $s_l$. Note that $V(\mathbb{P}, f^t, \pi_t) = V_{1,\mathbb{P}}^t(\varnothing, s_1)$. Also note that by the Bellman equation, we have the following.

$$V_{l,\mathbb{P}}^t(\tau_t[l-1], s_l) = \mathbb{E}_{a \sim \pi_t} \left[ f_l^t(\tau_t[l]) \right] + \mathbb{E}_{a \sim \pi_t} \left[ \mathbb{E}_{s' \sim \mathbb{P}(\cdot | s_l, a)} V_{l+1,\mathbb{P}}^t((\tau_t[l-1], s, a), s')) \right]$$

$$= \mathbb{E}_{a \sim \pi_t} \left[ f_l^t(\tau_t[l]) \right] + \mathbb{E}_{a \sim \pi_t} \left[ \mathbb{P}(\cdot | s_l, a)^\top V_{l+1,\mathbb{P}}^t((\tau_t[l-1], s_l, a), \cdot) \right]$$

Now use $\tau_t$ to set $\tau_{l-1} := \tau_t[l-1]$ and define the following.

$$\Delta_l^t(s_l) := V_{l,\mathbb{P}_\star}^t(\tau_t[l-1], s_l) - V_{l,\tilde{\mathbb{P}}_t}(\tau_t[l-1], s_l)$$

Note that

$$\Delta_1^t(s_1) = V(\tilde{\mathbb{P}}_t, f^t, \pi_t) - V(\mathbb{P}_\star, f^t, \pi_t) \tag{1}$$

The computation above then gives us the following.

$$\Delta_l^t(s_l^t) = \mathbb{E}_{a \sim \pi_t} \left[ \tilde{\mathbb{P}}_t(\cdot | s_l^t, a)^\top V_{l+1,\tilde{\mathbb{P}}_t}^t((\tau_t[l-1], s_l^t, a), \cdot)) \right]$$

$$\quad - \mathbb{E}_{a \sim \pi_t} \left[ \mathbb{P}_\star(\cdot | s_l^t, a)^\top V_{l+1,\mathbb{P}_\star}^t((\tau_t[l-1], s_l^t, a), \cdot)) \right]$$

$$= \tilde{\mathbb{P}}_t(\cdot | s_l^t, a_l^t)^\top V_{l+1,\tilde{\mathbb{P}}_t}^t(\tau_t[l], \cdot) - \mathbb{P}_\star(\cdot | s_l^t, a_l^t)^\top V_{l+1,\mathbb{P}_\star}^t(\tau_t[l], \cdot) + Y_{l,t} + Z_{l,t}$$

where $Y_{l,t}$ and $Z_{l,t}$ are stochastic processes defined below.

$$Y_{l,t} := \mathbb{P}_\star(\cdot | s_l^t, a)^\top V_{l+1,\mathbb{P}_\star}^t(\tau_t[l], \cdot) - \mathbb{E}_{a \sim \pi_t} \left[ \mathbb{P}_\star(\cdot | s_l^t, a)^\top V_{l+1,\mathbb{P}_\star}^t((\tau_t[l-1], s_l^t, a), \cdot) \right]$$

$$Z_{l,t} := \mathbb{E}_{a \sim \pi_t} \left[ \tilde{\mathbb{P}}_t(\cdot | s_l^t, a)^\top V_{l+1,\tilde{\mathbb{P}}_t}^t((\tau_t[l-1], s_l^t, a), \cdot) \right] - \tilde{\mathbb{P}}_t(\cdot | s_l^t, a)^\top V_{l+1,\tilde{\mathbb{P}}_t}^t(\tau_t[l], \cdot)$$

Consider the filtration $\mathcal{G}_{l,t}$ induced by the data of $\{\tau_s\}_{s=1}^{t-1} \cup \tau_t[l-1] \cup \{s_l^t\}$. Since $a_l^t \sim \pi_t$ and $(\tau_t[l-1], s_l^t, a_l^t) = \tau_t[l]$, we get that $\mathbb{E}[Y_{l,t}|\mathcal{G}_{l,t}] = \mathbb{E}[Z_{l,t}|\mathcal{G}_{l,t}] = 0$. So, one can see that both processes are martingales over $\mathcal{G}_{l,t}$. Also note that $|Y_{l,t}|, |Z_{l,t}| \leqslant p$. We thus have that

$$
\begin{aligned}
\Delta_l^t(s_l^t) &= \left[\widetilde{\mathbb{P}}_t(\cdot \mid s_l^t, a_l^t) - \mathbb{P}_\star(\cdot \mid s_l^t, a_l^t)\right] V_{l+1,\widetilde{\mathbb{P}}_t}^t(\tau_t[l], \cdot) \\
&\quad + \mathbb{P}_\star(\cdot \mid s_l^t, a_l^t)^\top \left[V_{l+1,\widetilde{\mathbb{P}}_t}^t(\tau_t[l], \cdot) - V_{l+1,\mathbb{P}_\star}^t(\tau_t[l], \cdot)\right] + Y_{l,t} + Z_{l,t} \\
&= \left[\widetilde{\mathbb{P}}_t(\cdot \mid s_l^t, a_l^t) - \mathbb{P}_\star(\cdot \mid s_l^t, a_l^t)\right] V_{l+1,\widetilde{\mathbb{P}}_t}^t(\tau_t[l], \cdot) + \mathbb{P}_\star(\cdot \mid s_l^t, a_l^t)^\top \Delta_{l+1}^t(s') + Y_{l,t} + Z_{l,t} \\
&= \left[\widetilde{\mathbb{P}}_t(\cdot \mid s_l^t, a_l^t) - \mathbb{P}_\star(\cdot \mid s_l^t, a_l^t)\right] V_{l+1,\widetilde{\mathbb{P}}_t}^t(\tau_t[l], \cdot) + \mathbb{E}_{s' \sim \mathbb{P}_\star(\cdot|s_l^t, a_l^t)}\left[\Delta_{l+1}^t(s')\right] + Y_{l,t} + Z_{l,t} \\
&= \left[\widetilde{\mathbb{P}}_t(\cdot \mid s_l^t, a_l^t) - \mathbb{P}_\star(\cdot \mid s_l^t, a_l^t)\right] V_{l+1,\widetilde{\mathbb{P}}_t}^t(\tau_t[l], \cdot) + \Delta_{l+1}^t(s_{l+1}^t) + U_{l,t} + Y_{l,t} + Z_{l,t}
\end{aligned}
$$

where

$$
U_{l,t} := \mathbb{E}_{s' \sim \mathbb{P}_\star(\cdot|s_l^t, a_l^t)}\left[\Delta_{l+1}^t(s')\right] - \Delta_{l+1}^t(s_{l+1}^t)
$$

Consider the filtration $\bar{\mathcal{G}}_{l,t}$ defined by the data of $\{\tau_s\}_{s=1}^{t-1} \cup \tau_t[l]$. Clearly, $U_{l,t}$ is a martingale over $\bar{\mathcal{G}}_{l,t}$. Also note that $|U_{l,t}| \leqslant p$ To conclude, we have that

$$
\Delta_l^t(s_l^t) - \Delta_{l+1}^t(s_{l+1}^t) = \left[\widetilde{\mathbb{P}}_t(\cdot \mid s_l^t, a_l^t) - \mathbb{P}_\star(\cdot \mid s_l^t, a_l^t)\right] V_{l+1,\widetilde{\mathbb{P}}_t}^t(\tau_t[l], \cdot) + U_{l,t} + Y_{l,t} + Z_{l,t}
$$

Using a telescoping sum over $l$ for a fixed $t$ and equation 1, we get that for any $t$, the following holds.

$$
\begin{aligned}
&V(\widetilde{\mathbb{P}}_t, f^t, \pi_t) - V(\mathbb{P}_\star, f^t, \pi_t) \\
&= \Delta_1^t(s_1) \\
&= \sum_{l=1}^H \left[\widetilde{\mathbb{P}}_t(\cdot \mid s_l^t, a_l^t) - \mathbb{P}_\star(\cdot \mid s_l^t, a_l^t)\right] V_{l+1,\widetilde{\mathbb{P}}_t}^t(\tau_t[l], \cdot) + U_{l,t} + Y_{l,t} + Z_{l,t}
\end{aligned}
$$

$$
\begin{aligned}
&V(\widetilde{\mathbb{P}}_t, f^t, \pi_t) - V(\mathbb{P}_\star, f^t, \pi_t) \\
&\leqslant \sum_{l=1}^H Bp \left\|\widetilde{\mathbb{P}}_t(\cdot \mid s_l^t, a_l^t) - \mathbb{P}_\star(\cdot \mid s_l^t, a_l^t)\right\|_1 + U_{l,t} + Y_{l,t} + Z_{l,t} \\
&\leqslant \sum_{l=1}^H Bp \left\|\widetilde{\mathbb{P}}_t(\cdot \mid s_l^t, a_l^t) - \hat{\mathbb{P}}_t(\cdot \mid s_l^t, a_l^t)\right\|_1 + Bp \left\|\mathbb{P}_\star(\cdot \mid s_l^t, a_l^t) - \hat{\mathbb{P}}_t(\cdot \mid s_l^t, a_l^t)\right\|_1 + U_{l,t} + Y_{l,t} + Z_{l,t}
\end{aligned}
$$
(2)

Until equation 2, all statements have held with probability 1 and did not use any facts about $\widetilde{\mathbb{P}}_t$. The last inequality also holds with probability 1 and uses the design of the confidence sets. Now, note the following well known concentration lemma. See, for example, Szepesvári (2023).

**Lemma 11.** *For* $\zeta(n, \delta) = 8\sqrt{\frac{S\log(2) + \log(n(n+1)SA/\delta)}{2n}}$ *and*

$$
\mathcal{C}_\mathcal{P}^t(\delta) = \left\{\mathbb{P} \,\middle|\, \|\mathbb{P}(\cdot \mid s, a) - \hat{\mathbb{P}}_t(\cdot \mid s, a)\|_1 \leqslant \zeta(N_t(s, a), \delta) \forall s, a\right\}
$$

*the true model* $\mathbb{P}^\star \in \mathcal{C}_\mathcal{P}^t(\delta)$ *for all* $t \geqslant 1$ *with probability at least* $1 - \delta/32$.

Applying the lemma twice and applying a union bound imply that the following holds with probability $1 - \delta/8$.

$$
\begin{aligned}
&\sum_{t=1}^T V(\widetilde{\mathbb{P}}_t, f^t, \pi_t) - V(\mathbb{P}_\star, f^t, \pi_t) \\
&\overset{(i)}{\leqslant} \sum_{l=1}^H 2Bp\zeta(N_t(s_l^t, a_l^t), \delta) + U_{l,t} + Y_{l,t} + Z_{l,t}
\end{aligned}
$$

$$= \sum_{t=1}^{T} \sum_{h=1}^{H} 2Bp\zeta(N_t(s_h^t, a_h^t), \delta) + \left[ \sum_{t=1}^{T} \sum_{h \in \mathcal{H}_p} U_{h,t} + Y_{h,t} + Z_{h,t} \right]$$

$$\overset{(ii)}{\leqslant} \sum_{t=1}^{T} \sum_{h=1}^{H} 2Bp\zeta(N_t(s_h^t, a_h^t), \delta) + \mathcal{O}\left( Bp\sqrt{pT \log(T/\delta)} \right)$$

Note that inequality $(i)$ is subtle since we could have used more data than that from $\tau_i, i = 1 \to t$ to construct $\mathcal{C}_{\mathcal{P}}^t$. The inequality still holds since $\zeta(n, \delta)$ is decreasing in $n$. Also, inequality $(ii)$ holds by the Azuma-Hoeffding inequality.

Now note that the whole argument above can be repeated with $\mathbb{P}_\star$ and $\widetilde{\mathbb{P}}_t$ switched, since the negative of a martingale is also a martingale. So, we have that with probability $1 - \delta/4$

$$\left| \sum_{t=1}^{T} V(\widetilde{\mathbb{P}}_t, f^t, \pi_t) - V(\mathbb{P}_\star, f^t, \pi_t) \right| \leqslant \sum_{t=1}^{T} \sum_{h=1}^{H} 2Bp\zeta(N_t(s_h^t, a_h^t), \delta) + \mathcal{O}\left( Bp\sqrt{pT \log(T/\delta)} \right)$$

Finally, we need the following easy lemma, proved in Szepesvári (2023).

**Lemma 12.** *Let* $c_\delta := 8\sqrt{S \log(2) + \log(HTSA/\delta)}$. *Then the following holds almost surely.*

$$\sum_{t=1}^{T} \sum_{h=1}^{H} 2Bp\zeta(N_t(s_h^t, a_h^t), \delta) \leqslant c_\delta Bp\sqrt{SAHT} + c_\delta BpHSA$$

This establishes our claim. $\qquad\square$

## D.2 Putting It All Together

**Theorem 1** (POR-UCRL Regret). *Under Assumption 1, the regret* $\text{Regret}(T)$ *of POR-UCRL is bounded by the following with probability at least* $1 - \delta$

$$\widetilde{\mathcal{O}}\left( \left( pS\sqrt{HA} + \sum_{h \in \mathcal{H}_p} \sqrt{d_{E,h} d_{C,h}} \right) \sqrt{T} \right)$$

*where* $d_{E,h} = \dim_E\left(\mathcal{F}_h, \frac{B}{T}\right)$ *and* $d_{C,h} = \log(\mathcal{N}(\mathcal{F}_h, 1/T, \|\cdot\|_\infty))$.

*Proof.* We can now combine Lemmas 7 and 11 to conclude that $\mathbb{M}_\star \in \mathcal{C}_{\mathcal{M}}(\mathcal{D}_t, \delta)$ for all $t$ with probability $1 - \delta/16$. We can now combine this observation with Lemmas 8 and 9 to observe that Assumption 3 is satisfied by POR-UCRL. By Theorem 4, the following holds with probability $1 - \delta$.

$$\text{Regret}(T) = \mathcal{O}\left( c_\delta Bp\sqrt{SAHT} + c_\delta BpHSA + \sum_{h \in \mathcal{H}_p} B\kappa_{2,h} d_{E,h} + \sum_{h \in \mathcal{H}_p} \kappa_{2,h}\sqrt{d_{E,h}\beta_{h,T}(\delta)T} \right)$$

where $c_\delta = 8\sqrt{S \log(2) + \log(HTSA/\delta)}$, $d_{E,h} = \dim_E\left(\mathcal{F}_h, \frac{B}{T}\right)$ is the Eluder dimension of $\mathcal{F}_h$ and $\beta_{h,T}(\delta) = \beta_{h,t}\left(\frac{\delta}{2t^2 H} = \widetilde{\mathcal{O}}(B^2\eta_h^2 d_{C,h})\right)$. This is because all the terms dependent on $p$ get absorbed by the first term in our expression below.

We further refine it by ignoring terms independent of $T$ and using the fact that $\beta_{h,T}(\delta) = \widetilde{\mathcal{O}}(d_{C,h})$ to get that

$$\text{Regret}(T) = \widetilde{\mathcal{O}}\left( pS\sqrt{AHT} + \sum_{h \in \mathcal{H}_p} \sqrt{d_{E,h} d_{C,h} T} \right)$$

$\qquad\square$

Analogously, we can provide a sample complexity result for POR-UCRL.

**Corollary 2** (POR-UCRL Sample complexity). *Let $\varepsilon > 0, \delta \in [0, 1]$. Ignoring polynomial terms independent of $\varepsilon$, we can bound the sample complexity $N(\varepsilon, \delta)$ of POR-UCRL as follows*

$$\widetilde{\mathcal{O}}\left(\frac{p^2 H S^2 A}{\varepsilon^2} + \frac{p^2 d_E d_C}{\varepsilon^2}\right)$$

*where $d_E := \max_{h \in \mathcal{H}_p} d_{E,h}$, and $d_C := \max_{h \in \mathcal{H}_p} d_{C,h}$.*

*Proof.* We invoke the regret-to-PAC conversion in Lemma 5 with confidence $\delta' = \delta/2$ and we plug the regret bound in Theorem 1 to write

$$\varepsilon = \widetilde{\mathcal{O}}\left(\left(BpS\sqrt{AH} + \sum_{h \in \mathcal{H}_p} \kappa_{2,h}\sqrt{d_{E,h} d_{C,h}} + Bp\sqrt{\log(1/\delta)}\right)\left(\frac{1}{\sqrt{T}}\right)\right)$$

from which we get the result by picking $N = T$ and the definition of $d_E, d_C$. $\qquad\square$

Also note the following theorem and corresponding corollary.

**Theorem 7** (POR-UCRL Regret if $\mathbb{P}_\star$ is Known). *If we know the transition matrix $\mathbb{P}_\star$ in POR-UCRL, then our regret is given by the following with probability $1 - \delta$, ignoring polynomial terms independent of $T$.*

$$\text{Regret}(T) = \widetilde{\mathcal{O}}\left(\left(Bp + \sum_{h \in \mathcal{H}_p} \sqrt{d_{E,h} d_{C,h}}\right)\sqrt{T}\right)$$

*Proof.* We can now use Lemmas 8 and the fact that $\mathcal{C}_{\mathcal{P}}^t(\delta)$ is always a singleton to observe that Assumption 3 is satisfied by this version of POR-UCRL as well. By Theorem 4, the following holds with probability $1 - \delta$.

$$\text{Regret}(T) = \widetilde{\mathcal{O}}\left(Bp\sqrt{T} + \sum_{h \in \mathcal{H}_p} B\kappa_{2,h} d_{E,h} + \sum_{h \in \mathcal{H}_p} \kappa_{2,h}\sqrt{d_{E,h}\beta_{h,T}(\delta)T}\right)$$

We further refine it by ignoring terms independent of $T$ and using the fact that $\beta_{h,T}(\delta) = \widetilde{\mathcal{O}}(d_{C,h})$ to get that

$$\text{Regret}(T) = \widetilde{\mathcal{O}}\left(\left(Bp + \sum_{h \in \mathcal{H}_p} \kappa_{2,h}\sqrt{d_{E,h}\beta_{h,T}(\delta)}\right)\sqrt{T}\right)$$

$$\square$$

**Corollary 3** (POR-UCRL sample complexity if $\mathbb{P}_\star$ is Known). *Let $\varepsilon > 0, \delta \in [0, 1]$. Ignoring polynomial terms independent of $\varepsilon$, we can bound the sample complexity $N(\varepsilon, \delta)$ of POR-UCRL when $\mathbb{P}_\star$ is known as follows*

$$\widetilde{\mathcal{O}}\left(\frac{p^2 d_E d_C}{\varepsilon^2}\right)$$

*where $d_E := \max_{h \in \mathcal{H}_p} d_{E,h}$, and $d_{C,h} := \max_{h \in \mathcal{H}_p} d_{C,h}$.*

*Proof.* The proof proceeds as in Corollary 2 by plugging Theorem 7 in Lemma 5. $\qquad\square$

# E  DETAILS AND PROOFS FOR CARDINAL POR-UCBVI

We now describe how we instantiate POR-UCBVI from a generic optimistic algorithm using bonuses. Note that again, this is **crucially different** from naively summarizing the history to define a modified state space, since we are separating the use of history summarization for getting bonuses for $f$ from using only the current state while getting bonuses for the Markovian transitions $\mathbb{P}$. Like with confidence sets, it is a priori unclear if we can use ideas from optimism to prove guarantees with a favorable (non-exponential) dependence on the complexity of transitions. In particular, we will note that showing that the bonuses are optimistic would naively need a union bound over the doubly exponential $(A^{(SA)^H})$ set of history-dependent policies, which is a non-trivial challenge to overcome.

Given a dataset of the first $t$ trajectory samples $\{\tau_i\}_{i=1}^t$ and an index $h \in [H]$, we consider the following:

**Estimates for POR-UCBVI:**

$$\widehat{f}_h^{t+1} = \arg\min_{f_h \in \mathcal{F}_h} \sum_{i=1}^t \left(\sigma(f_h(\tau_i[h])) - o_h^i\right)^2$$

We also use the MLE estimate for $\mathbb{P}$ after $t$ episodes to define $\hat{\mathbb{P}}^t(\cdot \mid s, a) := \frac{N_t(s,a,s')}{N_t(s,a)}$. Now for $\zeta(n, \delta) = 2\sqrt{\frac{S \log(2) + \log(n(n+1)SA/\delta)}{2n}}$, define $\mathcal{C}_{\mathbb{P}_t}(\delta)$ as below:

$$\left\{\mathbb{P} \Big| \|\mathbb{P}(\cdot \mid s, a) - \hat{\mathbb{P}}_t(\cdot \mid s, a)\|_1 \leq \zeta(N_t(s,a), \delta) \forall s, a\right\}$$

Recall the definition of our bonus below.

**Bonuses for POR-UCBVI.**  Recall that simple least squares guarantees imply the lemma below.

**Lemma 7** (Concentration for $\sigma \circ f_h$). *Define*

$$\mathrm{MSE}_{h,t}(f_h, f_h') := \sum_{i=1}^t \left(\sigma_h(f_h(\tau_i[h])) - \sigma_h(f_h'(\tau_i[h]))\right)^2$$

*Also define* $\bar{\beta}_{h,t}(\delta) = \eta_h^2 \log\left(\frac{N(\mathcal{F}_h, \frac{B}{T}, \|\cdot\|_\infty)}{\delta}\right) + \alpha_{h,t}$ *with* $\alpha_{h,t} := \frac{tB + t\eta_h \log\left(\frac{t}{\delta}\right)}{T}$. *Then* $f_h^\star$ *simultaneously satisfies* $\mathrm{MSE}_{h,t}(f_h^\star, \widehat{f}_h^{(t+1)}) \leq \bar{\beta}_{h,t}\left(\frac{\delta}{2t^2 H}\right)$ *for all* $h, t$ *with probability* $1 - \delta/32$.

We use the data from trajectories $\{\tau_i\}_{i=1}^t$ to build the confidence sets $\mathcal{C}_{\mathcal{F}}^{t+1}(\delta) = \prod_h \mathcal{C}_h^{t+1}(\delta)$ with $\mathcal{C}_h^{t+1}(\delta)$ defined below, where $\beta_{h,t}(\delta) := \bar{\beta}_{h,t}\left(\frac{\delta}{2t^2 H}\right)$.

$$\mathcal{C}_h^{t+1}(\delta) := \left\{f_h \in \mathcal{F}_h \Big| \mathrm{MSE}_{h,t}(f_h^\star, \widehat{f}_h^{(t+1)}) \leq \beta_{h,t}(\delta)\right\}$$

We first define a trajectory dependent bonus term below, with $\bar{\delta} := \frac{\delta}{HS^H A^H}$

$$\gamma_{h,t}(\tau[h], \delta) = \max_{f_h, f_h' \in \mathcal{C}_h^t(\bar{\delta})} f_h(\tau[h]) - f_h'(\tau[h])$$

Note that according to the definition of $\beta$, this does not create any exponential dependence in the confidence intervals used to define $\mathcal{C}_h^{t+1}$.

$$\beta_{h,t}\left(\frac{\delta}{16S^H A^H}\right) \leq 64\left(\log(N(\mathcal{F}_h, \alpha, \|\cdot\|_\infty)) + B + \eta_h \log(1/\delta) + \eta_h^2 H \log(THSA/\delta)\right)$$

$$= O(d_{C,h} + H)$$

It follows by a union bound over all trajectory segments and all timesteps $t$ that with probability at least $1 - \delta/16$ and for any trajectory $\tau$ and $t \geq 1, h \in \mathcal{H}_p$,

$$|f_h^\star(\tau[h]) - \widehat{f}_h^t(\tau[h])| \leq \gamma_{h,t}(\tau[h], \delta) \tag{3}$$

**Remark 5.** In the case of many popular function classes $\mathcal{F}$, like the linear class $\mathcal{F}_H = \{\tau \mapsto \phi(\tau)^\top \mathbf{w} \mid \|\mathbf{w}\| \leq W\}$, we can compute $\gamma_{h,t}(\tau[h], \delta)$ quite easily. In this case $\gamma_{H,t}$ is given by

$$\sup_{\mathbf{w}, \mathbf{w}' \in W_t} \phi(\tau)^\top (\mathbf{w} - \mathbf{w}') = \|\phi(\tau)\|_{V_t} \sup_{\mathbf{w}, \mathbf{w}' \in W_t} \|w - w'\|_{V_t^{-1}}$$

for a suitable quadratic form $V_t$.

$\gamma_{h,t}(\tau[h], \delta)$ induces a trajectory-dependent bonus, given by

$$b_{\mathcal{F}}^t(\tau, \delta) := \sum_{h \in \mathcal{H}_p} \gamma_{h,t}(\tau[h], \delta)$$

This in turn induces a policy-level bonus (which depends on the transition kernel), given by:

$$b_{\mathcal{F}}^t(\mathbb{P}, \pi, \delta) := \mathbb{E}_{\tau \sim \mathbb{P}^\pi}\left[b_{\mathcal{F}}^t(\tau, \delta)\right] = \mathbb{E}_{\tau \sim \mathbb{P}^\pi}\left[\sum_{h \in \mathcal{H}_p} \gamma_{h,t}(\tau[h], \delta)\right]$$

Let us define a term $\xi^t(s, a, \delta)$ that will be used to define the probability bonus.

$$\xi^t(s, a, \delta) := \min\left(2, 4\sqrt{\frac{H \log(6HSA) + S \log(8t^2 H^2) + \log(32t^2 N_t(s, a)/\delta)}{2N_t(s, a)}}\right)$$

This induces a trajectory-dependent bonus, given by

$$b_{\mathcal{P}}^t(\tau, \delta) := \sum_{h=1}^{H-1} \xi^t(s_h, a_h, \delta)$$

This induces a policy-level bonus (which depends on the transition kernel), given by:

$$b_{\mathcal{P}}^t(\mathbb{P}, \pi, \delta) := \min\left(4, \mathbb{E}_{\tau \sim \mathbb{P}^\pi}\left[b_{\mathcal{F}}^t(\tau, \delta)\right]\right) = \min\left(4, \mathbb{E}_{\tau \sim \mathbb{P}^\pi}\left[\sum_{h=1}^{H-1} \xi^t(s_h, a_h, \delta)\right]\right)$$

**Estimates and Bonuses in case $\mathbb{P}_\star$ is known.** If $\mathbb{P}_\star$ is instead known, keep $\hat{f}^t$ and $b_{\mathcal{F}}^t(\mathbb{P}, \pi, \delta)$ the same as above, but set $\hat{\mathbb{P}}_t := \mathbb{P}_\star$ and $b_{\mathcal{P}}^t(\mathbb{P}, \pi, \delta) := 0$ for all $t$.

For completeness we state POR-UCBVI here, which is an instantiation of Algorithm 3, the generic optimistic algorithm using bonuses.

---

**Algorithm 6** POR-UCBVI

---

1: **Input** Known family of reward functions $\{\mathcal{R}_h\}_{h=1}^H$, methods $\texttt{Est}(\mathcal{D})$ to estimate $\hat{\mathbb{P}}_t$ and $\hat{f}^t$ from dataset $\mathcal{D}$, confidence level $\delta$
2: **Initialize** $\mathcal{D}_1 \leftarrow \{\}$, initialize $\hat{f}^{\mathcal{D}_1}, \hat{\mathbb{P}}_{\mathcal{D}_1}$ arbitrarily.
3: **for** $t = 1, ..., T$ **do**
4:     **Compute** optimistic history dependent policy,

$$\pi_t = \arg\max_\pi V(\hat{\mathbb{P}}_t, \hat{f}^t, \pi) + b_{\mathcal{F}}^t(\hat{\mathbb{P}}_t, \pi, \delta) + z(Bp)(b_{\mathcal{P}}^t(\hat{\mathbb{P}}_t, \pi, \delta))$$

5:     **Observe** trajectory $\tau_t = \{(s_h^t, a_h^t)\}_{h=1}^H$ and feedback $\{o_h\}_{h \in \mathcal{H}_p}$.
6:     **Compute** new estimates $\hat{f}^{t+1}, \hat{\mathbb{P}}_{t+1} \leftarrow \texttt{Est}(\mathcal{D}_{t+1})$ and compute new bonus functions $b_{\mathcal{F}}^{t+1}(\hat{f}^{t+1}, \cdot, \delta), b_{\mathcal{P}}^{t+1}(\hat{\mathbb{P}}_{t+1}, \cdot, \delta)$.
7: **end for**

---

We will show the following regret bound.

**Theorem 8** (POR-UCBVI Regret). *Under Assumption 1, POR-UCBVI satisfies Assumption 4 and its regret* $\mathrm{Regret}(T)$ *is bounded by the following with probability at least* $1 - \delta$, *ignoring polynomial terms independent of* $T$.

$$\tilde{\mathcal{O}} \left( \left( pC(H, S, A) + \sum_{h \in \mathcal{H}_p} \sqrt{d_{E,h}(d_{C,h} + H)} \right) \sqrt{T} \right)$$

*where* $C(H, S, A) := H\sqrt{SA} + S\sqrt{HA}$

### E.1 SHOWING THAT ASSUMPTION 4 IS SATISFIED

#### E.1.1 BOUNDING EFFECT OF ERROR IN $\mathcal{F}$

**Lemma 13** (Bounding $\hat{f}^t$ Value Error). *Given any* $\mathbb{P}$, *with* $\hat{f}^t$ *computed using data from* $\{\tau_i\}_{i=1}^t \sim \mathbb{P}_\star^{\pi_i}$ *for any sequence of policies* $\pi_i$ *using least squares, the following holds with probability* $1 - \delta/16$ *uniformly over all* $\pi$.

$$|V(\mathbb{P}, \hat{f}^t, \pi) - V(\mathbb{P}, f^\star, \pi)| \leqslant b_{\mathcal{F}}^t(\mathbb{P}, \pi, \delta)$$

*Proof.* Recall that with probability at least $1 - \delta/16$, the following holds for any trajectory $\tau$ and any $t \geqslant 1, h \in \mathcal{H}_p$.

$$|f_h^\star(\tau[h]) - \hat{f}_h^t(\tau[h])| \leqslant \gamma_{h,t}(\tau[h], \delta) \tag{4}$$

Now note the following inequalities, where $(i)$ holds with probability $1 - \delta/16$ uniformly over all policies due to inequality 4 above.

$$V(\mathbb{P}, \hat{f}^t, \pi) - V(\mathbb{P}, f^\star, \pi) = \mathbb{E}_{\tau \sim \mathbb{P}^\pi} \left[ \sum_{h \in \mathcal{H}_p} \hat{f}_h^t(\tau[h]) - f_h^\star(\tau[h]) \right]$$

$$\overset{(i)}{\leqslant} \mathbb{E}_{\tau \sim \mathbb{P}^\pi} \left[ \sum_{h \in \mathcal{H}_p} \gamma_{h,t}(\tau[h], \delta) \right]$$

$$= b_{\mathcal{F}}^t(\mathbb{P}, \pi, \delta)$$

$\square$

**Lemma 14** (Bounding Sum of $\mathcal{F}$ Bonuses). *The following holds with probability* 1.

$$\sum_{t=1}^T b_{\mathcal{F}}^t(\mathbb{P}_\star, \pi_t, \delta) = \tilde{\mathcal{O}} \left( \sum_{h \in \mathcal{H}_p} B d_{E,h} + \sum_{h \in \mathcal{H}_p} \sqrt{d_{E,h}\beta_{h,T}(\bar{\delta})T} + Bp\sqrt{T \log(T/\delta)} \right)$$

*Proof.* First note the following inequality, which hold with probability $1 - \delta/16$ by the Azuma-Hoeffding inequality.

$$\sum_{t=1}^T b_{\mathcal{F}}^t(\mathbb{P}_\star, \pi_t, \delta) = \sum_{t=1}^T \mathbb{E}_{\tau \sim \mathbb{P}_\star^{\pi_t}} \left[ \sum_{h \in \mathcal{H}_p} \gamma_{t,h}(\tau[h], \delta) \right]$$

$$\leqslant \sum_{t=1}^T \sum_{h \in \mathcal{H}_p} \gamma_{t,h}(\tau[h], \delta) + \mathcal{O} \left( Bp\sqrt{T \log(T/\delta)} \right)$$

Now apply Lemma 3 in (Chan et al., 2021) for each $h$ separately, with the function class in the lemma set to $\{\sigma \circ f_h | f_h \in \mathcal{F}_h\}$, $P = 1$, $\mathbf{x}_{t,p} = \mathbf{x}_{t,1} = \tau_t[h]$ and misspecification $\varepsilon = 0$ (decoupled from the

Eluder dimension's $\varepsilon$). We also note that $o_h^t$ are $\eta$-subgaussian samples with mean $\sigma(f_h(\tau[h]))$. We obtain

$$\sum_{t=1}^{T} \max_{f_h, f_h' \in \mathcal{C}_h^t(\delta)} \sigma(f_h(\tau[h])) - \sigma(f_h'(\tau[h])) \leqslant \mathcal{O}\left(Bd_{E,h} + \sqrt{d_{E,h}\beta_{h,T}(\bar{\delta})T}\right)$$

where $d_{E,h} = \dim_E\left(\mathcal{F}_h, \frac{B}{T}\right)$ is the Eluder dimension of $\mathcal{F}_h$ and $\beta_{h,T}(\bar{\delta}) = \bar{\beta}\left(\frac{\bar{\delta}}{2t^2 H}\right)$. Since the Lipschitz constant of $\sigma^{-1}$ is $\kappa_2$, we have that the following holds with probability 1.

$$\sum_{t=1}^{T} \gamma_{t,h}(\tau[h], \delta) \leqslant \mathcal{O}\left(B\kappa_2 d_{E,h} + \kappa_2\sqrt{d_{E,h}\beta_{h,T}(\bar{\delta})T}\right)$$

This implies that the following holds with probability $1 - \delta/16$.

$$\sum_{t=1}^{T} b_{\mathcal{F}}^t(\mathbb{P}_\star, \pi_t, \delta) \leqslant \sum_{t=1}^{T}\sum_{h \in \mathcal{H}_p} \gamma_{t,h}(\tau[h], \delta) + \mathcal{O}\left(Bp\sqrt{T\log(T/\delta)}\right)$$

$$= \tilde{\mathcal{O}}\left(\sum_{h \in \mathcal{H}_p} B\kappa_2 d_{E,h} + \sum_{h \in \mathcal{H}_p} \kappa_2\sqrt{d_{E,h}\beta_{h,T}(\bar{\delta})T} + Bp\sqrt{T\log(T/\delta)}\right)$$

$\square$

### E.1.2 BOUNDING EFFECT OF ERROR IN $\mathcal{P}$

We now restate Lemma B.2 of Chatterji et al. (2021) in our notation.

**Lemma 15** (Change of Measure Inequality). *For any function $\mu$ of trajectories bounded by $D$, if $\hat{\mathbb{P}}_t$ is computed from data that includes trajectories $\{\tau_i \sim \mathbb{P}_\star^{\pi_i}\}_{i=1}^{t}$ for any sequence of policies $\pi_i$, then the following holds uniformly over all policies $\pi$ with probability $1 - \delta/16$.*

$$\mathbb{E}_{\tau \sim \mathbb{P}_\star^\pi}[\mu(\tau)] - \mathbb{E}_{\tau \sim \hat{\mathbb{P}}_t^\pi}[\mu(\tau)] \leqslant 2D\sqrt{\log(D)}b_{\mathcal{P}}^t(\hat{\mathbb{P}}_t, \pi, \delta)$$

*The same statement holds if we switch the roles of $\mathbb{P}$ and $\hat{\mathbb{P}}_t$ on both sides.*

*Proof.* For the order of $\mathbb{P}$ and $\hat{\mathbb{P}}_t$ in the statement, the following follows from Lemma B.2 of Chatterji et al. (2021) with $\eta = D$ and $\varepsilon = \frac{1}{t^2}$. We pull the additive $\log(D)$ in the square root outside to fit our assumption's phrasing.

$$\mathbb{E}_{\tau \sim \mathbb{P}_\star}[\mu(\tau)] - \mathbb{E}_{\tau \sim \hat{\mathbb{P}}_t}[\mu(\tau)] \leqslant D\sqrt{\log(D)}b_{\mathcal{P}}^t(\hat{\mathbb{P}}_t, \pi, \delta) + \frac{1}{t^2} \leqslant 2D\sqrt{\log(D)}b_{\mathcal{P}}^t(\hat{\mathbb{P}}_t, \pi, \delta)$$

The only subtlety is that more data than that from $\{\tau_i\}_{i=1}^{t}$ could have been used to compute $\hat{\mathbb{P}}_t$. The proof still follows since $c_{\mathcal{P}}(\hat{\mathbb{P}}_t, \pi, D)$ is decreasing in the counts $N_t(s, a)$.

Finally, if we switch $\mathbb{P}$ and $\hat{\mathbb{P}}_t$ on both sides, we can follow the proof of Lemma B.2 verbatim with $\mathbb{P}$ and $\hat{\mathbb{P}}_t$ switched everywhere, except for the martingale argument. There, instead of switching the two transition kernels, we negate the martingale to get our desired result. This exception is because we still need the expectation to be over the true transition kernel $\mathbb{P}_\star$ for the stochastic process defined to be a martingale. $\square$

**Lemma 16** (Bounding $\hat{\mathcal{P}}_t$ Value Error). *Consider any sequence of functions $f^t$ that induce value functions $V(\mathbb{P}, f^t, \pi)$. For any sequence of policies $\pi_t$, if the estimates $\hat{\mathbb{P}}_t$, bonuses $b_{\mathcal{P}}$ and costs $c_{\mathcal{P}}$ are generated using data including that of $\tau_i \sim \mathbb{P}_\star^{\pi_i}, i = 1 \rightarrow t$, then the following holds uniformly over $t$ and over all policies with probability $1 - \delta/16$.*

$$V(\hat{\mathbb{P}}_t, f^t, \pi) - V(\mathbb{P}_\star, f^t, \pi) = Bp\sqrt{\log(Bp)}(b_{\mathcal{P}}^t(\mathbb{P}_\star, \pi, \delta))$$

*The statement also holds if we switch $\hat{\mathbb{P}}_t$ and $\mathbb{P}_\star$.*

*Proof.* Note the following two inequalities that immediately follow from Lemma 15

$$V(\mathbb{P}_\star, f^t, \pi) - V(\hat{\mathbb{P}}_t, f^t, \pi) \leq Bp\sqrt{\log(Bp)}(b_\mathcal{P}^t(\hat{\mathbb{P}}_t, \pi, \delta))$$
$$V(\hat{\mathbb{P}}_t, f^t, \pi) - V(\mathbb{P}_\star, f^t, \pi) \leq Bp\sqrt{\log(Bp)}(b_\mathcal{P}^t(\mathbb{P}_\star, \pi, \delta))$$

Our result follows immediately. □

**Lemma 17** (Bounding Sum of $\mathcal{P}$ Bonuses). *The following holds with probability $1 - \delta/16$ whenever the data used to compute $b_\mathcal{P}^t(\mathbb{P}, \pi, \delta)$ includes the data of trajectories $\tau_i, t = 1 \to t$.*

$$\sum_{t=1}^T b_\mathcal{P}^t(\mathbb{P}_\star, \pi_t, \delta) = \widetilde{\mathcal{O}}\left(SA\bar{c}_\delta + \bar{c}_\delta\sqrt{HSAT}\right)$$

*where*

$$\bar{c}_\delta := 4\sqrt{\frac{H\log(6HSA) + S\log(8t^2H^2) + \log(32t^2N_T(s,a)/\delta)}{2}}$$

*This means that for any $s, a$, $\xi^t(s, a, \delta) = 2$ until $N_t(s, a) \geq \frac{\bar{c}_\delta}{2}$*

*Proof.* First note that by the definition of the bonus and the Azuma-Hoeffding inequality, we have the following.

$$\sum_{t=1}^T b_\mathcal{P}^t(\mathbb{P}_\star, \pi_t, \delta) \leq \sum_{t=1}^T \mathbb{E}_{\tau \sim \mathbb{P}^\pi} b_\mathcal{P}^t(\tau, \delta)$$

$$\leq \sum_{t=1}^T b_\mathcal{P}^t(\tau_t, \delta) + \mathcal{O}(4\sqrt{T\log(T/\delta)})$$

$$= \sum_{t=1}^T \sum_{h=1}^{H-1} \xi^t(s_h^t, a_h^t, \delta) + \mathcal{O}(4\sqrt{T\log(T/\delta)})$$

Now note that the first inequality holds even if more data beyond that of $\{\tau_i\}_{i=1}^t$ is used to compute $\xi^t(s, a, \delta)$, since $\xi^t(s, a, \delta)$ is decreasing in $N_t(s, a)$.

$$\sum_{t=1}^T \sum_{h=1}^{H-1} \xi^t(s_h^t, a_h^t, \delta) \leq SA\bar{c}_\delta + \sum_{t=1}^T \sum_{h=1}^{H-1} \frac{\bar{c}_\delta}{\sqrt{N_t(s_h^{(t)}, a_h^{(t)})}}$$

$$\leq SA\bar{c}_\delta + \bar{c}_\delta \sum_{(s,a)\in\mathcal{S}\times\mathcal{A}} \sum_{l=1}^{N_T(s,a)} \frac{1}{\sqrt{l}}$$

$$\leq SA\bar{c}_\delta + 2\bar{c}_\delta \sum_{(s,a)\in\mathcal{S}\times\mathcal{A}} \sqrt{N_T(s,a)}$$

$$\leq SA\bar{c}_\delta + 2\bar{c}_\delta \sqrt{SA \sum_{(s,a)\in\mathcal{S}\times\mathcal{A}} N_T(s,a)}$$

$$= \mathcal{O}\left(SA\bar{c}_\delta + \bar{c}_\delta\sqrt{SATH}\right)$$

This concludes our proof, since $1 = \widetilde{\mathcal{O}}(\sqrt{HSA})$ □

### E.1.3 PUTTING EVERYTHING TOGETHER

**Theorem 8** (POR-UCBVI Regret). *Under Assumption 1, POR-UCBVI satisfies Assumption 4 and its regret $\mathrm{Regret}(T)$ is bounded by the following with probability at least $1 - \delta$, ignoring polynomial terms independent of $T$.*

$$\widetilde{\mathcal{O}}\left(\left(pC(H, S, A) + \sum_{h\in\mathcal{H}_p} \sqrt{d_{E,h}(d_{C,h} + H)}\right)\sqrt{T}\right)$$

*where $C(H, S, A) := H\sqrt{SA} + S\sqrt{HA}$*

*Proof.* Note that by Lemmas 13, 14, 15, 16, 17, Assumption 4 is satisfied by POR-UCBVI. Using Theorem 5 and Lemmas 14 and 17, we have the following.

$$\text{Regret}(T) = \mathcal{O}\Big( \sum_{h \in \mathcal{H}_p} B\kappa_{2,h} d_{E,h} + BpHSA\bar{c}_\delta + \sum_{h \in \mathcal{H}_p} \kappa_{2,h}\sqrt{d_{E,h}\beta_{h,T}(\bar{\delta})T}$$
$$+ Bp\sqrt{T\log(T/\delta)} + \bar{c}_\delta BpH\sqrt{SAT}\Big)$$

We further refine it grouping terms and ignoring terms independent of $T$, and also noting that $\bar{c}_\delta = \widetilde{\mathcal{O}}(\sqrt{H} + \sqrt{S})$ as well as $\beta_{h,T}(\bar{\delta}) = \mathcal{O}(d_{C,h} + H)$

$$\text{Regret}(T) = \widetilde{\mathcal{O}}\left(\left(\sum_{h \in \mathcal{H}_p} \kappa_2\sqrt{d_{E,h}(d_{C,h} + H)} + Bp(H\sqrt{SA} + S\sqrt{HA})\right)\sqrt{T}\right)$$
$$= \mathcal{O}\left(\left(p(H\sqrt{SA} + S\sqrt{HA}) + \sum_{h \in \mathcal{H}_p} \sqrt{d_{E,h}(d_{C,h} + H)}\right)\sqrt{T}\right)$$

$\square$

From the latter we derive a sample complexity result as follows.

**Corollary 4** (POR-UCBVI Sample complexity). *Let $\varepsilon > 0, \delta \in [0,1]$. Ignoring polynomial terms independent of $\varepsilon$, we can bound the sample complexity $N(\varepsilon, \delta)$ of POR-UCBVI as follows*

$$\widetilde{\mathcal{O}}\left(\frac{p^2 HSA \max(H, S)}{\varepsilon^2} + \frac{p^2 d_E \max(d_C, H)\log(1/\delta)}{\varepsilon^2}\right)$$

*where $d_E := \max_{h \in \mathcal{H}_p} d_{E,h}$, and $d_C := \max_{h \in \mathcal{H}_p} d_{C,h}$.*

*Proof.* We invoke the regret-to-PAC conversion in Lemma 5 with confidence $\delta' = \delta/2$ and we plug the regret bound in Theorem 8 to write

$$\varepsilon = \widetilde{\mathcal{O}}\left(\left(Bp(H\sqrt{SA} + S\sqrt{HA}) + \sum_{h \in \mathcal{H}_p} \sqrt{d_{E,h}\beta_{h,T}(\bar{\delta})} + Bp\sqrt{\log(1/\delta)}\right)\left(\frac{1}{\sqrt{T}}\right)\right)$$

from which we get the result by noting $N = pT$ and the definition of $d_E, d_C$. $\square$

We also have the following theorem and corollary, in the same vein as Theorem 7.

**Theorem 9** (Regret for POR-UCBVI if $\mathbb{P}_\star$ is Known). *When $\mathbb{P}_\star$ is known, POR-UCBVI that sets $\hat{\mathbb{P}}_t := \mathbb{P}_\star$ and $b_\mathcal{P}^t(\mathbb{P}, \pi\delta) := 0$ for all $t \geqslant 1$ still satisfies Assumption 4 and its regret $\text{Regret}(T)$ is bounded by the following with probability at least $1 - \delta$, ignoring terms independent of $T$.*

$$\widetilde{\mathcal{O}}\left(\left(\sum_{h \in \mathcal{H}_p} \sqrt{d_{E,h}(d_{C,h} + H)}\right)\sqrt{T}\right)$$

*where $d_{E,h} = \dim_E\left(\mathcal{F}_h, \frac{B}{T}\right)$.*

*Proof.* Note that by Lemmas 13 and 14, Assumption 4 is satisfied by POR-UCBVI. Using Lemma 14, we have the following.

$$\text{Regret}(T) = \mathcal{O}\left(\sum_{h \in c\mathcal{H}_p} B\kappa_2 d_{E,h} + \sum_{h \in \mathcal{H}_p} \kappa_2\sqrt{d_{E,h}\beta_{h,T}(\delta)T} + Bp\sqrt{T\log(T/\delta)}\right)$$

We further refine it grouping terms and ignoring terms independent of $T$, and also noting that $\bar{c}_\delta = \widetilde{\mathcal{O}}(\sqrt{H} + \sqrt{S})$

$$
\begin{aligned}
\text{Regret}(T) &= \widetilde{\mathcal{O}}\left(\left(\sum_{h \in \mathcal{H}_p} \kappa_2 \sqrt{d_{E,h}\beta_{h,T}(\delta)} + Bp\right)\sqrt{T}\right) \\
&= \mathcal{O}\left(\left(Bp + \sum_{h \in \mathcal{H}_p} \sqrt{d_{E,h}\beta_{h,T}(\delta)}\right)\sqrt{T}\right) \\
&= \widetilde{\mathcal{O}}\left(\left(\sum_{h \in \mathcal{H}_p} \sqrt{d_{E,h}(d_{C,h} + H)}\right)\sqrt{T}\right)
\end{aligned}
$$

$\square$

**Corollary 5** (POR-UCBVI Sample complexity if $\mathbb{P}_\star$ is Known). *Let $\varepsilon > 0, \delta \in [0,1]$. Ignoring polynomial terms independent of $\varepsilon$, we can bound the sample complexity $N(\varepsilon, \delta)$ of POR-UCBVI when $\mathbb{P}_\star$ is known as follows*

$$
\widetilde{\mathcal{O}}\left(\frac{p^2 H d_E \max(d_C, H)}{\varepsilon^2}\right)
$$

*where $d_E := \max_{h \in \mathcal{H}_p} d_{E,h}$, $\beta := \max_{h \in \mathcal{H}_p} \beta_{T,h}(\delta)$, and $d_C := \max_{h \in \mathcal{H}_p} d_{C,h}$.*

*Proof.* The proof proceeds as in Corollary 4 by plugging Theorem 9 in Lemma 5. $\square$

## F    EXTENSION TO GENERAL FUNCTION APPROXIMATION FOR PROBABILITY TRANSITIONS

Note that for simplicity of exposition and proofs, the confidence sets for $\mathcal{P}$ in POR-UCRL and the bonuses for $\mathcal{P}$ in POR-UCBVI are given for tabular observed states and actions $s, a$. The function approximation using $\mathcal{G}$ and $\mathcal{F}$ is relegated to the effect of latent states $u$. Using the work of Liu et al. (2022b), it is quite straightforward to extend and modify the proofs here to general function approximation for $\mathcal{P}$.

Let us assume that we have a set of parameters $\Theta$ and a mapping from $\theta \in \Theta$ to probability transition functions $\mathbb{P}_\theta \in \mathcal{P}$ so that the image of $\Theta$ under this mapping is all of $\mathcal{P}$. These functions $\mathbb{P}_\theta$ mapping from $s, a$ to distributions over $s'$ act on a featurization $\phi(s, a)$ of $s, a$. This is the most general function approximation setting for probability transition functions. Linear MDPs, tabular MDPs, factored MDPs, kernel linear MDPs, and many other function approximation settings are special cases of this.

For convenience of notation, we will occasionally drop the subscript $\theta$. It is beneficial to remember that throughout this section, we are working under general function approximation nevertheless.

### F.1    POR-UCRL

For POR-UCRL, we redefine the confidence sets to be the ones in Algorithm 2 (reward-free OMLE) in Liu et al. (2022b). That is, we first recursively define $\mathcal{D}_{t+1} := \mathcal{D}_t \cup \{(\pi_t, \tau_t)\}$ instead of just appending $\tau_t$. We also recursively define $\mathcal{C}_\mathcal{P}(\mathcal{D}_{t+1}, \delta)$ below. Note that we merely rephrased the definition in Liu et al. (2022b) to use the negative log-likelihood instead of the log-likelihood.

$$\mathcal{C}_\mathcal{P}(\mathcal{D}_{t+1}, \delta) :=$$

$$\mathcal{C}_\mathcal{P}(\mathcal{D}_t, \delta) \cap \left\{ \theta \in \Theta \middle| \sum_{(\pi, \tau) \in \mathcal{D}_{t+1}} -\log(\mathbb{P}_\theta^\pi(\tau)) \leqslant \min_\theta \sum_{(\pi, \tau) \in \mathcal{D}_{t+1}} -\log(\mathbb{P}_\theta^\pi(\tau)) + \beta \right\}$$

We now simply need to replace Lemma 9 with a version involving function approximation. Recall the following.

$$V(\mathbb{P}, f^t, \pi_t) := \mathbb{E}_{\tau' \sim \mathbb{P}^{\pi_t}} \left[ \sum_{h=1}^H \widetilde{f}_h^t(\tau[h]) \right]$$

$$= \mathbb{E}_{\tau' \sim \mathbb{P}^{\pi_t}} \left[ \sum_{h \in \mathcal{H}_p} \widetilde{f}_h^t(\tau[h]) \right]$$

So, by a simple change of measure inequality and the fact that $|f_h^t| \leqslant B$ for all $h$, we have the following.

$$\left| V(\widetilde{\mathbb{P}}_t, f^t, \pi_t) - V(\mathbb{P}_\star, f^t, \pi_t) \right| \leqslant B p \, d_{TV}(\widetilde{\mathbb{P}}_t^{\pi_t}, \mathbb{P}_\star^{\pi_t}) \tag{5}$$

Where $d_{TV}$ represents the $TV$ distance taken over all trajectories of length $H$. This implies the following.

$$\left| \sum_{t=1}^T V(\widetilde{\mathbb{P}}_t, f^t, \pi_t) - V(\mathbb{P}_\star, f^t, \pi_t) \right| \leqslant B p \sum_{t=1}^T d_{TV}(\widetilde{\mathbb{P}}_t^{\pi_t}, \mathbb{P}_\star^{\pi_t})$$

Now, we have two options. We can either use the distributional eluder dimension directly, or we can use the strong SAIL condition of Liu et al. (2022b). The former is more flexible and allows us to address all function approximation scenarios. The latter approach has very crisp guarantees and is still satisfied by most function approximation models.

#### F.1.1    USE THE DISTRIBUTIONAL ELUDER DIMENSION

Note that by proposition B.2 and step 2 of the proof of Theorem 3.2 in Liu et al. (2022b), we have the following upon setting $\Pi_{\exp}(\pi) := \pi$ in their proofs. This way, we are not running any extra

"exploratory" policies at every step.

$$\sum_{k=1}^{t-1} d_{TV}^2(\widetilde{\mathbb{P}}_t^{\pi_k}, \mathbb{P}_\star^{\pi_k}) \leqslant \mathcal{O}(\beta) \tag{6}$$

Here, $\beta = c\log(\mathcal{N}_{br,\mathcal{P}}(1/T)) + c\log(T/\delta)$, where $\mathcal{N}_{br,\mathcal{P}}(\varepsilon)$ is the $\varepsilon$-bracketing number of $\mathcal{P} = \{\mathbb{P}_\theta\}_{\theta \in \Theta}$ as in definition 2.2 of Liu et al. (2022b), and $c$ is a universal constant.

Now, consider the functions $\phi_t$ defined on the space of policies by $\phi_t(\pi) := d_{TV}(\widetilde{\mathbb{P}}_t^\pi, \mathbb{P}_\star^\pi)$. Define $\mu_t$ to be the Dirac-delta measures on $\pi_t$. Let the class of Dirac-delta measures over policies be $\mathcal{D}_\Pi$, and let the class of possible $\phi_t$ functions be $\Phi$. Let $d_{E,\mathcal{P}}$ be the corresponding distributional Eluder dimension $\dim_{DE}(\Phi, \mathcal{D}_\Pi, \sqrt{1/T})$, as defined in Jin et al. (2021). Then by equation 6 and the properties of the distributional Eluder dimension (Lemma 41 of Jin et al. 2021), we have the following.

$$\sum_{t=1}^{T} d_{TV}(\widetilde{\mathbb{P}}_t^{\pi_t}, \mathbb{P}_\star^{\pi_t}) = \widetilde{\mathcal{O}}(\sqrt{d_{E,\mathcal{P}}\beta T})$$

We can further define $d_{C,\mathcal{P}} = \log(\mathcal{N}_{br,\mathcal{P}}(1/T))$ to be the bracketing dimension of $\mathcal{P}$. By equation 5, we have the following.

$$\left| V(\widetilde{\mathbb{P}}_t, f^t, \pi_t) - V(\mathbb{P}_\star, f^t, \pi_t) \right| \leqslant \widetilde{\mathcal{O}}(\sqrt{d_{E,\mathcal{P}}\beta T})$$

We can now follow the rest of the proof for the guarantee for POR-UCRL verbatim and establish the following bound on POR-UCRL regret.

**Theorem 10.** *Consider the functions $\phi_t$ defined on the space of policies by $\phi_t(\pi) := d_{TV}(\widetilde{\mathbb{P}}_t^\pi, \mathbb{P}_\star^\pi)$. Define $\mu_t$ to be the Dirac-delta measures on $\pi_t$. Let the class of Dirac-delta measures over policies be $\mathcal{D}_\Pi$, and let the class of possible $\phi_t$ functions be $\Phi$. Let $d_{E,\mathcal{P}}$ be the corresponding distributional Eluder dimension $\dim_{DE}(\Phi, \mathcal{D}_\Pi, \sqrt{1/T})$. Define $d_{C,\mathcal{P}} = \log(\mathcal{N}_{br,\mathcal{P}}(1/T))$ to be the bracketing dimension of $\mathcal{P}$. Then, Then, we have the following bound for the regret of our modified POR-UCRL algorithm, with probability $1 - \delta$.*

$$\text{Regret}(T) = \widetilde{\mathcal{O}}\left( \left( p\sqrt{d_{E,\mathcal{P}}d_{C,\mathcal{P}}} + \sum_{h \in \mathcal{H}_p} \sqrt{d_{E,h}d_{C,h}} \right) \sqrt{T} \right)$$

Future work can investigate the choice of the distribution class $\mathcal{D}_\Pi$ and make other technical tweaks to obtain crisper versions of the $d_{E,\mathcal{P}}$ defined here.

### F.1.2 USE THE STRONG SAIL CONDITION

We make a minor modification to the POR-UCRL algorithm in the spirit of Algorithm 2 (reward-free OMLE) in Liu et al. (2022b) to use this condition. Instead of merely running $\pi_t$, we run all policies in an exploratory set $\Pi_{\exp}(\pi_t)$ of size $\Pi_{\exp}$ every time we collect trajectories. Recall that the strong SAIL condition of Liu et al. (2022b) is satisfied by well conditioned PSRs, factored MDPs, kernel linear MDPs, sparse linear bandits, etc. These automatically subsume tabular MDPs and linear MDPs.

Assume that our model class $\mathcal{P}$ satisfies the $(\sqrt{d_{S,\mathcal{P}}}, \kappa, C)$ strong SAIL condition. That is, we define $d_{S,\mathcal{P}} := d^2$ for the $d$ in the original strong SAIL condition. Now by theorem 7.2 of Liu et al. (2022b), we can conclude that since $\widetilde{\mathbb{P}}_t \in \mathcal{C}_{\mathcal{P}}(\mathcal{D}_t, \delta)$, we have that

$$d_{TV}(\widetilde{\mathbb{P}}_t^{\pi_t}, \mathbb{P}_\star^{\pi_t}) \leqslant poly(H)\sqrt{d_{S,\mathcal{P}}} \left( \frac{C|\Pi_{\exp}|}{t} + \kappa\sqrt{\frac{\beta|\Pi_{\exp}|^2}{t}} \log^2(t/|\Pi_{\exp}|) \right)$$

Here, $\beta = c\log(\mathcal{N}_{br,\mathcal{P}}(1/T)) + c\log(T/\delta)$, where $\mathcal{N}_{br,\mathcal{P}}(\varepsilon)$ is the $\varepsilon$-bracketing number of $\mathcal{P} = \{\mathbb{P}_\theta\}_{\theta \in \Theta}$ and $c$ is a universal constant. Adding these up across $t$ and discarding logarithmic terms, we get the following.

$$Bp \sum_{t=1}^{T} d_{TV}(\widetilde{\mathbb{P}}_t^{\pi_t}, \mathbb{P}_\star^{\pi_t}) = \widetilde{\mathcal{O}}\left( poly(H)Bp\kappa\sqrt{d_{S,\mathcal{P}}\beta|\Pi_{\exp}|^2 T} \right)$$

This means that by equation 5, we have the following.

$$\left| \sum_{t=1}^{T} V(\widetilde{\mathbb{P}}_t, f^t, \pi_t) - V(\mathbb{P}_\star, f^t, \pi_t) \right| = \widetilde{\mathcal{O}} \left( poly(H) Bp\kappa \sqrt{d_{S,\mathcal{P}} \beta |\Pi_{\exp}|^2 T} \right)$$

Let us also define $d_{C,\mathcal{P}} = \log(\mathcal{N}_{br,\mathcal{P}}(1/T))$ to be the bracketing dimension of $\mathcal{P}$. We can now follow the rest of the proof for the guarantee for POR-UCRL verbatim and get the following bound on POR-UCRL regret, with the caveat that we are ignoring the contribution of the non-optimal exploratory policies. We have thus established the following theorem.

**Theorem 11.** *Assume that our model class $\mathcal{P}$ satisfies the $(\sqrt{d_{S,\mathcal{P}}}, \kappa, C)$ strong SAIL condition. Define $d_{C,\mathcal{P}} = \log(\mathcal{N}_{br,\mathcal{P}}(1/T))$ to be the bracketing dimension of $\mathcal{P}$. Then, we have the following bound for the regret of our modified POR-UCRL algorithm, with probability $1 - \delta$.*

$$\text{Regret}(T) = \widetilde{\mathcal{O}} \left( \left( poly(H) p\kappa \sqrt{d_{S,\mathcal{P}} d_{C,\mathcal{P}} |\Pi_{\exp}|^2} + \sum_{h \in \mathcal{H}_p} \sqrt{d_{E,h} d_{C,h}} \right) \sqrt{T} \right)$$

### F.2 POR-UCBVI

Again, we start with confidence sets that we will use to define the bonuses. That is, we define the confidence sets to be the ones in Algorithm 2 (reward-free OMLE) in Liu et al. (2022b). That is, we first recursively define $\mathcal{D}_{t+1} := \mathcal{D}_t \cup \{(\pi_t, \tau_t)\}$ instead of just appending $\tau_t$. We also recursively define $\mathcal{C}_\mathcal{P}(\mathcal{D}_{t+1}, \delta)$ below. Note that we merely rephrased the definition in Liu et al. (2022b) to use the negative log-likelihood instead of the log-likelihood.

$$\mathcal{C}_\mathcal{P}(\mathcal{D}_{t+1}, \delta) :=$$

$$\mathcal{C}_\mathcal{P}(\mathcal{D}_t, \delta) \cap \left\{ \theta \in \Theta \left| \sum_{(\pi, \tau) \in \mathcal{D}_{t+1}} -\log(\mathbb{P}_\theta^\pi(\tau)) \leq \min_\theta \sum_{(\pi, \tau) \in \mathcal{D}_{t+1}} -\log(\mathbb{P}_\theta^\pi(\tau)) + \beta \right. \right\}$$

Defining bonuses is trickier than defining confidence sets. Like the case of POR-UCRL above, we have two options again – we can use the distributional Eluder dimension or the strong SAIL condition. **While it is possible to work out the details for the former, they are much more involved and do not cover significantly more useful examples than those covered by the strong SAIL condition. So, we will focus on the latter.** This is especially true when considering sample complexity, since the use of exploratory policies under the strong SAIL condition creates no complications in giving sample complexity guarantees, unlike it does for regret guarantees.

#### F.2.1 USING THE STRONG SAIL CONDITION

Again, we make a minor modification to the POR-UCBVI algorithm in the spirit of Algorithm 2 (reward-free OMLE) in Liu et al. (2022b) to use this condition. Instead of merely running $\pi_t$, we run all policies in an exploratory set $\Pi_{\exp}(\pi_t)$ of size $\Pi_{\exp}$ every time we collect trajectories. Again, recall that the strong SAIL condition of Liu et al. (2022b) is satisfied by well conditioned PSRs, factored MDPs, kernel linear MDPs, sparse linear bandits, etc. These automatically subsume tabular MDPs and linear MDPs.

Let $\hat{\mathbb{P}}_t$ be the MLE model at time $t$. Assume that our model class $\mathcal{P}$ satisfies the $(\sqrt{d_{S,\mathcal{P}}}, \kappa, C)$ strong SAIL condition. That is, we define $d_{S,\mathcal{P}} := d^2$ for the $d$ in the original strong SAIL condition. Now by theorem 7.2 of Liu et al. (2022b), we can conclude that since $\mathbb{P}_t \in \mathcal{C}_\mathcal{P}(\mathcal{D}_t, \delta)$, we have that the following holds with probability $1 - \delta$.

$$\max_\pi d_{TV}(\hat{\mathbb{P}}_t^\pi, \mathbb{P}_\star^\pi) \leq poly(H) \sqrt{d_{S,\mathcal{P}}} \left( \frac{C|\Pi_{\exp}|}{t} + \kappa \sqrt{\frac{\beta |\Pi_{\exp}|^2}{t}} \log^2(t/|\Pi_{\exp}|) \right)$$

Here, $\beta = c \log(\mathcal{N}_{br,\mathcal{P}}(1/T)) + c \log(T/\delta)$, where $\mathcal{N}_{br,\mathcal{P}}(\varepsilon)$ is the $\varepsilon$-bracketing number of $\mathcal{P} = \{\mathbb{P}_\theta\}_{\theta \in \Theta}$ and $c$ is a universal constant. By a simple change of measure, for any policy $\pi$, we have the following.

$$\left| V(\hat{\mathbb{P}}_t, f^t, \pi) - V(\mathbb{P}_\star, f^t, \pi) \right| \leq Bp d_{TV}(\hat{\mathbb{P}}_t^\pi, \mathbb{P}_\star^\pi)$$

holds for any policy $\pi$ with probability 1, the following holds simultaneously for all policies with probability $1 - \delta$.

$$\left| V(\hat{\mathbb{P}}_t, f^t, \pi) - V(\mathbb{P}_\star, f^t, \pi) \right| \leqslant poly(H) B p \sqrt{d_{S,\mathcal{P}}} \left( \frac{C|\Pi_{\exp}|}{t} + \kappa \sqrt{\frac{\beta|\Pi_{\exp}|^2}{t}} \log^2(t/|\Pi_{\exp}|) \right)$$

Define the right hand side as the bonus $b_\mathcal{P}^t(\mathbb{P}, \pi, \delta)$. Adding these up across $t$ and discarding logarithmic terms, we get the following.

$$\sum_{t=1}^T b_\mathcal{P}^t(\mathbb{P}, \pi, \delta) = \tilde{\mathcal{O}}\left( poly(H) B p \kappa \sqrt{d_{S,\mathcal{P}} \beta |\Pi_{\exp}|^2 T} \right)$$

This means that by equation 5, we have the following.

$$\left| \sum_{t=1}^T V(\widetilde{\mathbb{P}}_t, f^t, \pi_t) - V(\mathbb{P}_\star, f^t, \pi_t) \right| = \tilde{\mathcal{O}}\left( poly(H) B p \kappa \sqrt{d_{S,\mathcal{P}} \beta |\Pi_{\exp}|^2 T} \right)$$

Let us also define $d_{C,\mathcal{P}} = \log(\mathcal{N}_{br,\mathcal{P}}(1/T))$ to be the bracketing dimension of $\mathcal{P}$. We can now follow the rest of the proof for the guarantee for POR-UCBVI verbatim and get a bound on POR-UCBVI regret, with the caveat that we are ignoring the contribution of the non-optimal exploratory policies. We have thus established the following theorem.

**Theorem 12.** *Assume that our model class $\mathcal{P}$ satisfies the $(\sqrt{d_{S,\mathcal{P}}}, \kappa, C)$ strong SAIL condition. Define $d_{C,\mathcal{P}} = \log(\mathcal{N}_{br,\mathcal{P}}(1/T))$ to be the bracketing dimension of $\mathcal{P}$. Then, we have the following bound for the regret of our modified POR-UCBVI algorithm, with probability $1 - \delta$.*

$$\text{Regret}(T) = \tilde{\mathcal{O}}\left( \left( poly(H) p \kappa \sqrt{d_{S,\mathcal{P}} d_{C,\mathcal{P}} |\Pi_{\exp}|^2} + \sum_{h \in \mathcal{H}_p} \sqrt{d_{E,h} d_{C,h}} \right) \sqrt{T} \right)$$

### F.3 DUELING PORRL

We have the following immediate corollary of Theorem 3, Theorem 10 and Lemma 2.

**Corollary 6** (Dueling Regret using modified POR-UCRL Confidence Sets)**.** *Consider the functions $\phi_t$ defined on the space of policies by $\phi_t(\pi) := d_{TV}(\widetilde{\mathbb{P}}_t^\pi, \mathbb{P}_\star^\pi)$. Define $\mu_t$ to be the Dirac-delta measures on $\pi_t$. Let the class of Dirac-delta measures over policies be $\mathcal{D}_\Pi$, and let the class of possible $\phi_t$ functions be $\Phi$. Let $d_{E,\mathcal{P}}$ be the corresponding distributional Eluder dimension $\dim_{DE}(\Phi, \mathcal{D}_\Pi, \sqrt{1/T})$. Define $d_{C,\mathcal{P}} = \log(\mathcal{N}_{br,\mathcal{P}}(1/T))$ to be the bracketing dimension of $\mathcal{P}$. Then, the modified confidence sets for POR-UCRL described in section F.1.1 satisfy Assumption 2 and using them in Algorithm 1 leads to the following dueling regret bound with probability $1 - \delta$.*

$$\text{Regret}_D(T) = \tilde{\mathcal{O}}\left( \left( p \sqrt{d_{E,\mathcal{P}} d_{C,\mathcal{P}}} + \sum_{h \in \mathcal{H}_p} \sqrt{d_{E,h} d_{C,h}} \right) \sqrt{T} \right)$$

Similar corollaries can be immediately produced for the strong SAIL method, whenever $|\Pi_{\exp}| = 1$. The strong SAIL method doesn't quite work for dueling regret guarantees when $|\Pi_{\exp}| > 1$, so we must stick to the distributional eluder dimension in that case.

# G  DETAILS AND PROOFS FOR PORRL WITH GOLF

For completeness and establishing notation, we recall GOLF here.

---

**Algorithm 7** GOLF

---

1: **Input** Known class of Bellman consistent Q-functions $\mathcal{Q}$, confidence level $\delta$.
2: **Initialize** dataset $\mathcal{D}_1 \leftarrow \{\}$ and $\mathcal{C}_{\mathcal{Q}}(\mathcal{D}_1, \delta) \leftarrow \mathcal{Q}$.
3: **for** $t = 1, ..., T$ **do**
4:     $\tau[0] \leftarrow ()$
5:     **for** $h = 1, \dots H$ **do**
6:         **Compute** $a_h^t, Q_h^t \leftarrow \arg\max_{a, Q \in \mathcal{C}_{\mathcal{Q}}(\mathcal{D}_t, \delta)} Q(\tau[h], a)$
7:         **Play** $a_h^t$ and observe feedback $o_h^t$
8:     **end for**
9:     **Update** $\mathcal{D}_{t+1} \leftarrow \mathcal{D}_t \cup \{\tau, (o_1^t, \dots o_H^t\}$
10:    **Compute**

$$\mathcal{C}_{\mathcal{Q}}(\mathcal{D}_{t+1}, \delta) \leftarrow \left\{ \mathcal{L}_{\mathcal{D}_t}(Q_h, Q_{h+1}) \leqslant \inf_{g \in \mathcal{G}_h} \mathcal{L}_{\mathcal{D}_t}(g, Q_{h+1}) + \beta \right\}$$

11: **end for**

---

**Theorem 2** (Modified GOLF Regret). *Let Assumption 1 hold, let $\mathcal{Q}$ be Bellman complete, let $d_{\text{HABE}} = \dim_{\text{HABE}}(\mathcal{Q}, \alpha, \min(\alpha, \sqrt{1/T}))$ and let $d_{C,\mathcal{Q}} := \log(\mathcal{N}(\mathcal{Q} \cup \mathcal{G}, 1/T, \|\cdot\|_\infty))$. Choose hyperparameter $\beta = c(\log(HT) + d_{C,\mathcal{Q}})$ for some universal constant $c$ and the auxiliary function class $\mathcal{G}$ used in GOLF, and define . Then, GOLF satisfies $\text{Regret}(T) = \mathcal{O}\left(pH\sqrt{d_{\text{HABE}}d_{C,\mathcal{Q}}T}\right)$.*

*Proof.* The meat of the theorem is in proving Lemma 18. We $\beta = c\log(HT\mathcal{N}(\mathcal{Q} \cup \mathcal{G}, 1/T, \|\cdot\|_\infty))$ for some suitably large universal constant $c$, and use Theorem 6 and Lemma 18 to get that

$$\text{Regret}(T) = \sum_{h=1}^{H} \left( T\omega + Bp\left(\frac{B^2p^2\beta}{\alpha^2} + 1\right)\left(\sum_{l=1}^{h-1} d_l(\alpha)\right) + \min(d_h(\omega), T)Bp + 2Bp\sqrt{\beta d_h(\omega)T} \right)$$

where $d_h(\varepsilon) := \dim_{DE}(\Phi_h, \mathcal{D}_h, \varepsilon)$. Now set $\omega = \frac{Bp}{T}$ and use the fact that $d_h(\varepsilon)$ increases with decreasing $\varepsilon$ to get that

$$\text{Regret}(T) = \tilde{\mathcal{O}}\left(pH\sqrt{d_{\text{HABE}}\beta T}\right)$$
$$= \tilde{\mathcal{O}}\left(pH\sqrt{d_{\text{HABE}}d_{C,\mathcal{Q}}T}\right)$$

since $d_{\text{HABE}} := \dim_{\text{HABE}}(\mathcal{Q}, \min(\alpha, Bp/T)) := \max_h d_h(\min(\alpha, Bp/T))$. $\square$

**Corollary 7** (GOLF Sample complexity). *Let $\varepsilon > 0, \delta \in [0, 1]$. Ignoring polynomial terms independent of $\varepsilon$, we can bound the sample complexity $N(\varepsilon, \delta)$ of GOLF as follows*

$$\tilde{\mathcal{O}}\left(\frac{p^2H^2d_{\text{HABE}}d_{c,\mathcal{Q}}}{\varepsilon^2}\right).$$

*Proof.* Again, we use Lemma 5 and a quick computation shows our result. $\square$

## G.1  COMPARING $\dim_{\text{HABE}}$ AND $\dim_{\text{BE}}$

It is easy to see that since the function class $\Phi_h$ is a subset of the class $\Psi_h$ of all Bellman errors, $\dim_{\text{HABE}} \leqslant \max_h \dim_{DE}(\Psi_h, \mathcal{D}_{h,\mathcal{Q}}, \varepsilon)$. Recall that the Bellman eluder dimension is a minimum over the RHS and another term that uses Dirac-$\delta$ distributions, but typically, the RHS is smaller. So, in many cases, $\dim_{\text{HABE}} \leqslant \dim_{\text{BE}}$. However, we don't have a universal inequality in either direction.

### G.2 COMPUTING DIMENSIONS FOR THE COMBINATION LOCK

**Proposition 1** (Dimensions for the Combination Lock). *Consider the combination lock problem with model class $\mathcal{M} = \mathcal{P} \times \mathcal{F}$ and induced Q-function class $\mathcal{Q}$.*

- *Under dense intermediate feedback with $\mathcal{H}_p = [H]$, $\dim_{\mathrm{HABE}}(\mathcal{Q}, \alpha) = A$ for all $\alpha < q$, while its BE dimension is at least $A^H - 2$. The eluder dimension for reward functions $\dim_E(\mathcal{F}_h, \frac{B}{T})$ is at least $A^h$ for any $h \leqslant H$.*

- *For sparse intermediate feedback with $\mathcal{H}_p = \{H\}$ and any $\alpha > 0$, the $\alpha$-HABE dimension, the BE dimension and the eluder dimension of $\mathcal{F}_H$ are all at least $A^H - 2$. Moreover, any algorithm in this setting will have regret $\Omega(\sqrt{A^H T})$.*

*Proof.* We separately resolve the cases of sparse and dense intermediate feedback.

#### G.2.1 DENSE INTERMEDIATE FEEDBACK, $\mathcal{H}_p = [H]$

Notice that we get a reward $Ber(q)$ at every step as long as we are on the correct sequence of actions $a_1^\star, \dots a_H^\star$, and as soon as we take a wrong action, we always get a reward of 0 subsequently. It is then easy to see that the induced function classes $\mathcal{Q}$ then are given by $\mathcal{Q} = \{(Q_1, \dots Q_H) \mid \exists a_1, \dots a_H \in \mathcal{A} \text{ s.t. } Q_h = (H - h + 1)q\mathbb{1}_{a_1, \dots a_h}\}$.

**$\alpha$-HABE dimension:** It suffices to show the upper bound using $\mathcal{D}_{h, \mathcal{Q}(\alpha, h-1)}$, since the $\alpha$-HABE dimension takes the minimum of distirbutional eluder dimensions over two distributions. For any $\alpha < q$, consider the function class

$$\mathcal{Q}(\alpha, h - 1) = \left\{ Q \in \mathcal{Q}, \left| \mathbb{E}_{\mu_l(Q)}[Q_l - \mathcal{T}_l Q_{l+1}] \right| \leqslant \alpha, \ \forall 1 \leqslant l \leqslant h - 1 \right\}$$

Now note that $\mathbb{E}_{\mu_l(Q)}[Q_l - \mathcal{T}_l Q_{l+1}] = q\mathbb{1}_{a_1, \dots a_l} - q\mathbb{1}_{a_1^\star, \dots a_l^\star}$. If this is smaller than $\alpha$, then this is smaller than $q$ and thus must be 0. So, $(a_1, \dots a_{h-1}) = (a_1^\star, \dots a_{h-1}^\star)$ for any $Q \in \mathcal{Q}(\alpha, h - 1)$. This also means that any $\phi_h \in \Phi_h$, there is a $Q \in \mathcal{Q}(\alpha, h - 1)$ so that

$$\phi_h = Q_h - \mathcal{T}_h Q_{h+1} = q\mathbb{1}_{a_1^\star, \dots a_{h-1}^\star, a_h} - q\mathbb{1}_{a_1^\star, \dots a_{h-1}^\star, a_h^\star}$$

Thus, the size of $\Phi_h$ is just $A$. More importantly, the set $\mathcal{D}_{h, \mathcal{Q}(\alpha, h-1)}$ of distributions $\mu_h(Q)$ induced by $Q \in \mathcal{Q}(\alpha, h - 1)$ only include indicators of the form $\mathbb{1}_{a_1^\star, \dots a_{h-1}^\star, a}$ for actions $a$. Thus, the set of distributions $\mathcal{D}_{\mathcal{Q}(\alpha, h-1)}$ has size $A$. We know that the distributional eluder dimension $d = \dim_{DE}(\Phi_h, \mathcal{D}_{\mathcal{D}(\alpha, h-1)}, \min(\alpha, Bp/T))$ is bounded by the number of possible distributions $\left| \mathcal{D}_{\mathcal{Q}(\alpha, h-1)} \right|$. So, $d \leqslant A$.

**BE dimension:** The Bellman differences, from above, are $q\mathbb{1}_{a_1, \dots a_h} - q\mathbb{1}_{a_1^\star, \dots a_h^\star}$. This is an affine transformation of a family of $A^H$ indicator functions. The distributions $\mu_l(Q)$ over trajectories induced by $Q$ include indicators $\mathbb{1}_{a_1', \dots a_l'}$ of all trajectories of length $l$. Now for any sequence $\mu_1, \dots \mu_n, \mu_{n+1}$ of different indicator distributions not including $a_1^\star, \dots a_l^\star$, we consider the Bellman difference $g_{n+1} = q\mathbb{1}_{a_1, \dots a_h} - q\mathbb{1}_{a_1^\star, \dots a_h^\star}$ with action sequence given by $\mu_{n+1}$. Note that $\mathbb{E}_{\mu_i} g_{n+1} = 0$ for all $i \leqslant n$ but $\mathbb{E}_{\mu_{n+1}} g_{n+1} = q$. This means that the longest possible sequence in the definition of the distributional eluder dimension has length $A^H - 2$. So, the BE dimension is at least $A^H - 2$.

**Eluder dimension:** The reward function class $\mathcal{F}_h$ is given by all functions of the form $q\mathbb{1}_{a_1, \dots a_h}$. This is a scaled version of a class of $A^h$ indicator functions. Since it contains $A^h$ indicator functions, its eluder dimension is at least $A^h$.

#### G.2.2 SPARSE INTERMEDIATE FEEDBACK, $\mathcal{H}_p = [H]$

Notice that we get a reward $Ber(q)$ at the *last* step if we took correct sequence of actions $a_1^\star, \dots a_H^\star$, and reward 0 otherwise. It is then easy to see that now, the induced function classes $\mathcal{Q}$ then are given by $\mathcal{Q} = \{(Q_1, \dots Q_H) \mid \exists a_1, \dots a_H \in \mathcal{A} \text{ s.t. } Q_h = q\mathbb{1}_{a_1, \dots a_h}\}$.

**$\alpha$-HABE dimension:** This time, note that $\mathbb{E}_{\mu_h(Q)}[Q_l - \mathcal{T}_h Q_{h+1}] = 0$ for all $h \leqslant H - 1$. So, the function class $\Phi_h = \{0\}$ for all $h \leqslant H - 1$. Only for $h = H$ do we have that $\mathbb{E}_{\mu_H(Q)}[Q_H - \mathcal{T}_H Q_{H+1}] = q\mathbb{1}_{a_1, \dots a_H} - q\mathbb{1}_{a_1^\star, \dots a_H^\star}$. Also note that $\mathcal{Q}(\alpha, H - 1) = \mathcal{Q}$ for all $\alpha$ since $\mathbb{E}_{\mu_h(Q)}[Q_l - \mathcal{T}_h Q_{h+1}] = 0$ for all $h \leqslant H - 1$. So, this is merely the BE dimension of the problem. Now, the Bellman differences at timestep $H$ are identical to those for the sparse feedback problem, and the

distributions $\mathcal{D}_{\mathcal{Q}(\alpha, H-1)} = \mathcal{D}_{\mathcal{Q}}$ since we have established that $\mathcal{Q}(\alpha, H-1) = \mathcal{Q}$. This means that by the argument for BE dimension in the dense feedback case, we have that the distributional eluder dimension of $\Phi_H$ is at least $A^H - 2$, which is then also the $\alpha$-HABE dimension of this problem.

**BE dimension:** From the argument for the $\alpha$-HABE dimension in the sparse case, the BE dimension and the $\alpha$-HABE dimension match in this case, and are both at least $A^H - 2$.

**Eluder dimension:** Again, the reward function class $\mathcal{F}_H$ is given by all functions of the form $q\mathbb{1}_{a_1, \dots a_H}$. This is a scaled version of a class of $A^H$ indicator functions. Since it contains $A^H$ indicator functions, its eluder dimension is at least $A^H$.

Also note that this example also produces a universal lower bound of $\sqrt{(A^H T)}$ under any algorithm. That is, no algorithm can improve over our bounds under sparse feedback. This lower bound is an immediate consequence of regret lower bounds for bandit algorithms by treating each sequence of actions taken as a different arm.

$\square$

### G.3 PROOFS OF LEMMAS

Recall that $\mathcal{Q}(\alpha, h) = \{Q \in \mathcal{Q} \mid |\mathbb{E}_{\mu_l(Q)}[Q_l - \mathcal{T}_l Q_{l+1}]| \leqslant \alpha, \forall 1 \leqslant l \leqslant h\}$, that $\mu_h(Q)$ is the distribution induced on $\tau[h-1], a_h$ by $\pi_Q$ and $\mathcal{D}_{h, \mathcal{Q}} := \{\mu_h(Q) \mid Q \in \mathcal{Q}\}$.

**Lemma 18.** *Let $d_h(\varepsilon) := \dim_{DE}(\Phi_h, \mathcal{D}_{h, \mathcal{Q}(\alpha, h-1)}, \varepsilon)$ with*

$$\Phi_h := \left\{ Q_h - \mathcal{T}_h Q_{h+1} \Big| Q \in \mathcal{Q}(\alpha, h-1) \right\}$$

*Then, we have that for $\beta = c \log(HT\mathcal{N}(\mathcal{Q} \cup \mathcal{G}, 1/T, \|\cdot\|_\infty))$, $\sum_{j=1}^t |\mathbb{E}_{\mu_h(Q^j)}[Q_h^j - \mathcal{T}_h Q_{h+1}^j]|$ is bounded by*

$$t\omega + Bp\left(\frac{B^2 p^2 \beta}{\alpha^2} + 1\right)\left(\sum_{l=1}^{h-1} d_l(\alpha)\right) + \min(d_h(\omega), t)Bp + 2Bp\sqrt{\beta d_h(\omega)t}$$

*Proof.* We modify the proof of Lemma 41 in Jin et al. (2021). Pick arbitrary $h$ and $t$ and let $\Psi_h$ be the function class given by

$$
\begin{aligned}
\Phi_h &:= \left\{ Q_h - \mathcal{T}_h Q_{h+1} \Big| Q \in \mathcal{Q}(\alpha, h) \right\} \\
&= \left\{ Q_h - \mathcal{T}_h Q_{h+1} \Big| (Q_1, \dots Q_H) \in \mathcal{Q}, |\mathbb{E}_{\mu_l(Q)}[Q_l - \mathcal{T}_l Q_{l+1}]| \leqslant \alpha, \forall 1 \leqslant l \leqslant h-1 \right\}
\end{aligned}
$$

Also note that we have the function class $\Phi_h$ of timestep $h$ Bellman errors induced by "historically $\alpha$-accurate" functions - functions whose expected Bellman errors in previous timesteps are smaller than $\alpha$. The distribution used for computing the expected Bellman errors for previous timesteps is $\mu_l(Q)$.

Now abbreviating $\psi_l^j := Q_l^j - \mathcal{T}_l Q_{l+1}^j$ gives a sequence $\psi_l^1, \dots \psi_l^t$ of functions in $\Psi_l$ for every $1 \leqslant l \leqslant h$. This must have a subsequence $\phi_l^1, \dots \phi_l^{r_l}$ consisting of all the functions in the sequence that lie in $\Phi_l$, for every $1 \leqslant l \leqslant h$. Also let $d_h(\varepsilon) = \dim_{DE}(\Phi_h, \mathcal{D}_h, \varepsilon)$ for any $\varepsilon$. Now note that

$$\sum_{j=1}^t |\mathbb{E}_{\mu_h(Q^j)}[Q_h^j - \mathcal{T}_h Q_{h+1}^j]|$$

$$= \sum_{j=1}^t |\mathbb{E}_{\mu_h(Q^j)}[\psi_h^j]|$$

$$\overset{(i)}{=} \sum_{j=1}^t \left|\mathbb{E}_{\mu_h(Q^j)}[\psi_h^j]\right| \mathbb{1}\left(|\mathbb{E}_{\mu_h(Q^j)}[\psi_h^j]| \leqslant \omega\right)$$

$$+ \sum_{l=1}^{h-1} \sum_{j=1}^t \left|\mathbb{E}_{\mu_h(Q^j)}[\psi_h^j]\right| \mathbb{1}\left(|\mathbb{E}_{\mu_h(Q^j)}[\psi_h^j]| > \omega, Q \in \mathcal{Q}(\alpha, l-1)\backslash\mathcal{Q}(\alpha, l)\right)$$

$$+ \sum_{j=1}^{t} \left| \mathbb{E}_{\mu_h(Q^j)}[\psi_h^j] \right| \mathbb{1}\left( |\mathbb{E}_{\mu_h(Q^j)}[\psi_h^j]| > \omega, Q \in \mathcal{Q}(\alpha, h-1) \right)$$

$$\leqslant \sum_{j=1}^{t} \left| \mathbb{E}_{\mu_h(Q^j)}[\psi_h^j] \right| \mathbb{1}\left( |\mathbb{E}_{\mu_h(Q^j)}[\psi_h^j]| \leqslant \omega \right)$$

$$+ \sum_{l=1}^{h-1} \sum_{j=1}^{t} \left| \mathbb{E}_{\mu_h(Q^j)}[\psi_h^j] \right| \mathbb{1}\left( Q \in \mathcal{Q}(\alpha, l-1) \backslash \mathcal{Q}(\alpha, l) \right)$$

$$+ \sum_{j=1}^{t} \left| \mathbb{E}_{\mu_h(Q^j)}[\psi_h^j] \right| \mathbb{1}\left( |\mathbb{E}_{\mu_h(Q^j)}[\psi_h^j]| > \omega, Q \in \mathcal{Q}(\alpha, h-1) \right)$$

$$\leqslant t\omega + \sum_{l=1}^{h-1} \sum_{j=1}^{t} \left| \mathbb{E}_{\mu_h(Q^j)}[\psi_h^j] \right| \mathbb{1}\left( |\mathbb{E}_{\mu_l(Q^j)}\psi_l^j| > \alpha, Q \in \mathcal{Q}(\alpha, l-1) \right)$$

$$\sum_{j=1}^{t} \left| \mathbb{E}_{\mu_h(Q^j)}[\psi_h^j] \right| \mathbb{1}\left( |\mathbb{E}_{\mu_h(Q^j)}[\psi_h^j]| > \omega, Q \in \mathcal{Q}(\alpha, h-1) \right)$$

$$\overset{(ii)}{\leqslant} t\omega + \sum_{l=1}^{h-1} \sum_{j=1}^{r_l} \left| \mathbb{E}_{\mu_h(Q^j)}[\phi_h^j] \right| \mathbb{1}\left( |\mathbb{E}_{\mu_l(Q^j)}[\phi_l^j]| > \alpha \right) + \sum_{j=1}^{r_h} \left| \mathbb{E}_{\mu_h(Q^j)}[\phi_h^j] \right| \mathbb{1}\left( |\mathbb{E}_{\mu_h(Q^j)}[\phi_h^j]| > \omega \right)$$

$$\overset{(ii)}{\leqslant} t\omega + Bp\left( \frac{B^2 p^2 \beta}{\alpha^2} + 1 \right) \left( \sum_{l=1}^{h-1} d_l(\alpha) \right) + \sum_{j=1}^{r_h} \left| \mathbb{E}_{\mu_h(Q^j)}[\phi_h^j] \right| \mathbb{1}\left( |\mathbb{E}_{\mu_h(Q^j)}[\phi_h^j]| > \omega \right)$$

Here, $(i)$ holds since one of three possibilities holds – either $\left| \mathbb{E}_{\mu_h(Q^j)}[\psi_h^j] \right| \leqslant \omega$, or $|\mathbb{E}_{\mu_h(Q^j)}\psi_l^j| > \omega$ and there is a least $l \leqslant h-1$ so that $Q \in \mathcal{Q}(\alpha, l-1)$ but $Q \notin \mathcal{Q}(\alpha, h-1)$, or $Q \in \mathcal{Q}(\alpha, h-1)$. $(ii)$ holds since if $|\mathbb{E}_{\mu_k(Q)}\psi_k^j| \leqslant \alpha$ for all $k \leqslant l-1$, then $\psi_l^j = \phi_l^i$ for some $i$. Finally, $(iii)$ holds by Proposition 43 of Jin et al. (2021) since $\sum_{j=1}^{s-1} \mathbb{E}_{\mu_l(Q^j)}[(\phi_l^j)^2] \leqslant \beta$ by Lemma 39(a) of Jin et al. (2021). While our rewards are stochastic and theirs are not, we can repeat their arguments verbatim after noting that the martingale defined in the beginning of their proof continues to be a martingale even for stochastic rewards that have second moments.

Now arrange the sequence $\left| \mathbb{E}_{\mu_h(Q^j)}\phi_s \right|$ in order to get $e_1, \ldots e_{r_h}$. We can then write

$$\sum_{j=1}^{t} \left| \mathbb{E}_{\mu_h(Q^j)}[Q_h^j - \mathcal{T}_h Q_{h+1}^j] \right| \leqslant t\omega + Bp\left( \frac{B^2 p^2 \beta}{\alpha^2} + 1 \right) \left( \sum_{l=1}^{h-1} d_l(\alpha) \right) + \sum_{j=1}^{r_h} e_j \mathbb{1}(e_j > \omega)$$

For any $e_j > \omega$, consider arbitrary $\gamma$ such that $e_j > \gamma > \omega$. This means that by Proposition 43 of Jin et al. (2021) again,

$$j \leqslant \sum_{i=1}^{r_h} \mathbb{1}(e_i > \gamma) \leqslant \left( \frac{B^2 p^2 \beta}{\gamma^2} + 1 \right) d_h(\omega)$$

This means that $\gamma \leqslant Bp\sqrt{\frac{\beta d_h(\omega)}{j - d_h(\omega)}}$ for any such $\gamma$. Since $e_j \leqslant Bp$, we get that $e_j \leqslant \min\left( Bp, Bp\sqrt{\frac{\beta d_h(\omega)}{j - d_h(\omega)}} \right)$. Finally, this means that

$$\sum_{j=1}^{t} \left| \mathbb{E}_{\mu_h(Q^j)}[Q_h^j - \mathcal{T}_h Q_{h+1}^j] \right|$$

$$\leqslant t\omega + Bp\left( \frac{B^2 p^2 \beta}{\alpha^2} + 1 \right) \left( \sum_{l=1}^{h-1} d_l(\alpha) \right) + \sum_{j=1}^{r_h} e_j \mathbb{1}(e_j > \omega)$$

$$\leqslant t\omega + Bp\left( \frac{B^2 p^2 \beta}{\alpha^2} + 1 \right) \left( \sum_{l=1}^{h-1} d_l(\alpha) \right) + \sum_{j=1}^{r_h} \min\left( Bp, Bp\sqrt{\frac{\beta d_h(\omega)}{j - d_h(\omega)}} \right)$$

$$\leqslant t\omega + Bp\left(\frac{B^2p^2\beta}{\alpha^2} + 1\right)\left(\sum_{l=1}^{h-1} d_l(\alpha)\right) + \min(d_h, r_h)Bp + \sum_{j=1}^{r_h} Bp\sqrt{\frac{\beta d_h(\omega)}{j}}$$

$$\leqslant t\omega + Bp\left(\frac{B^2p^2\beta}{\alpha^2} + 1\right)\left(\sum_{l=1}^{h-1} d_l(\alpha)\right) + \min(d_h(\omega), r_h)Bp + 2Bp\sqrt{\beta d_h(\omega)r_h}$$

$$\leqslant t\omega + Bp\left(\frac{B^2p^2\beta}{\alpha^2} + 1\right)\left(\sum_{l=1}^{h-1} d_l(\alpha)\right) + \min(d_h(\omega), t)Bp + 2Bp\sqrt{\beta d_h(\omega)t}$$

as desired. $\qquad\square$

# H  PROOFS FOR DUELING FEEDBACK

## H.1  PROOF FOR REDUCTION TO CONFIDENCE-SET OPTIMISM

**Theorem 3** (Reduction from Dueling to Confidence-Set-Based Optimism). *If the confidence sets* $\mathcal{C}_{\mathcal{M}}(\mathcal{D}_t, \delta)$ *satisfy Assumption 2, then the dueling regret* $\text{Regret}_D(T)$ *of Algorithm 1 is given by*

$$\text{Regret}_D(T) = \widetilde{\mathcal{O}}(C_P(\mathcal{M}, T, \delta) + C_F(\overline{\mathcal{M}}, T, \delta))$$

**Remark 6.** While the theorem states that we need Assumption 2 from the main paper, we actually use its slightly more refined version – Assumption 3. The less refined version was added to the main paper for brevity.

*Proof.* For ease of notation, let us use the sets $\mathcal{C}_{\mathcal{M}}(\mathcal{D}_t, \delta)$ given by the pre-image of $\mathcal{C}_{\overline{\mathcal{M}}}(\mathcal{D}_t, \delta)$ under the map $\mathbb{M} \mapsto \overline{\mathbb{M}}$ from Section 4. We first recall that $\mathbb{M}_\star \in \mathcal{C}_{\mathcal{M}}(\mathcal{D}_t, \delta)$ and so $\pi_\star \in \Pi_t$ for all $t$ with probability $1 - \delta/16$. Recall that the value of a duel $(\pi, \pi')$ under model $\overline{\mathbb{M}} \leftrightarrow$ is denoted by

$$V_D(\overline{\mathbb{M}}, \pi, \pi') := V(\mathbb{M}, \pi) - V(\mathbb{M}, \pi') = V(\mathbb{P}, f, \pi) - V(\mathbb{P}, g, \pi')$$

We overload notation and use the natural maps $(\mathbb{P}, f) \leftrightarrow \mathbb{M} \mapsto \overline{\mathbb{M}}$ to define

$$V_D(\mathbb{M}, \pi, \pi') := V_D(\overline{\mathbb{M}}, \pi, \pi')$$

For ease of notation, we will then work with $\mathcal{C}_{\mathcal{M}}(\mathcal{D}_t, \delta)$ in this proof until we can. Since $\pi_{i,t} \in \Pi_t$ for $i = 1, 2$, there exists some $\mathbb{M}_{i,t} \in \mathcal{C}_{\mathcal{M}}(\mathcal{D}_t, \delta)$ for $i = 1, 2$ so that $V_D(\mathbb{M}_{i,t}, \pi, \pi_{1,t}) \leqslant 0$ for all $\pi$. Note that dueling regret is given below. Inequality $(i)$ is by definition of $\mathbb{M}_{i,t}$, since $V_D(\mathbb{M}_{i,t}, \pi_\star, \pi_{i,t}) \leqslant 0$ for $i = 1, 2$. Inequality $(ii)$ holds by definition of $\pi_{1,t}, \pi_{2,t}$.

$$\begin{aligned}
\text{Regret}_D(T) &= \sum_{t=1}^{T} \sum_{i=1}^{2} V_D(\mathbb{M}_\star, \pi_\star, \pi_{i,t}) \\
&= \sum_{t=1}^{T} \Big[ \sum_{i=1}^{2} V_D(\mathbb{M}_\star, \pi_\star, \pi_{i,t}) - V_D(\mathbb{M}_{i,t}, \pi_\star, \pi_{i,t}) + \sum_{i=1}^{2} V_D(\mathbb{M}_{i,t}, \pi_\star, \pi_{i,t}) \Big] \\
&\overset{(i)}{\leqslant} \sum_{i=1}^{2} \sum_{t=1}^{T} [V_D(\mathbb{M}_\star, \pi_\star, \pi_{i,t}) - V_D(\mathbb{M}_{i,t}, \pi_\star, \pi_{i,t})] \\
&\overset{(ii)}{\leqslant} \sum_{t=1}^{T} 2 \max_{\mathbb{M}, \mathbb{M}' \in \mathcal{C}_{\mathcal{M}}(\mathcal{D}_t, \delta)} \big[ V_D(\mathbb{M}, \pi_{1,t}, \pi_{2,t}) - V_D(\mathbb{M}', \pi_{1,t}, \pi_{2,t}) \big]
\end{aligned}$$

Continuing, we have

$$\begin{aligned}
\text{Regret}_D(T) &\leqslant \sum_{t=1}^{T} 2 \max_{\mathbb{M}, \mathbb{M}' \in \mathcal{C}_{\mathcal{M}}(\mathcal{D}_t, \delta)} \big[ V_D(\mathbb{M}, \pi_{1,t}, \pi_{2,t}) - V_D(\mathbb{M}', \pi_{1,t}, \pi_{2,t}) \big] \\
&= 2 \sum_{t=1}^{T} \max_{\mathbb{M}, \mathbb{M}' \in \mathcal{C}_{\mathcal{M}}(\mathcal{D}_t, \delta)} \Big[ V_D(\mathbb{M}, \pi_{1,t}, \pi_{2,t}) - V_D(\mathbb{M}_\star, \pi_{1,t}, \pi_{2,t}) + V_D(\mathbb{M}_\star, \pi_{1,t}, \pi_{2,t}) \\
&\qquad\qquad\qquad\qquad - V_D(\mathbb{M}', \pi_{1,t}, \pi_{2,t}) \Big] \\
&\leqslant 2 \sum_{t=1}^{T} \max_{\mathbb{M}, \mathbb{M}' \in \mathcal{C}_{\mathcal{M}}(\mathcal{D}_t, \delta)} \big[ V_D(\mathbb{M}, \pi_{1,t}, \pi_{2,t}) - V_D(\mathbb{M}_\star, \pi_{1,t}, \pi_{2,t}) \big] + \\
&\qquad \max_{\mathbb{M}, \mathbb{M}' \in \mathcal{C}_{\mathcal{M}}(\mathcal{D}_t, \delta)} \big[ V_D(\mathbb{M}_\star, \pi_{1,t}, \pi_{2,t}) - V_D(\mathbb{M}', \pi_{1,t}, \pi_{2,t}) \big] \\
&= 2 \sum_{t=1}^{T} \big[ V_D(\widetilde{\mathbb{M}}_t, \pi_{1,t}, \pi_{2,t}) - V_D(\mathbb{M}_\star, \pi_{1,t}, \pi_{2,t}) \big] + \big[ V_D(\mathbb{M}_\star, \pi_{1,t}, \pi_{2,t}) - V_D(\widetilde{\mathbb{M}}'_t, \pi_{1,t}, \pi_{2,t}) \big]
\end{aligned}$$

where $\widetilde{\mathbb{M}}_t$ and $\widetilde{\mathbb{M}}'_t$ are the respective maximisers. It suffices to analyse only one of the terms, as a consequence of the symmetry of Assumption 3.

We can now use the fact that $\mathbb{M}$ is described by $(\mathbb{P}, f)$ to analyse the first term, letting $\widetilde{\mathbb{M}}_t \leftrightarrow (\widetilde{\mathbb{P}}_t, \widetilde{f}^t)$.

$$\sum_{t=1}^{T} \left[ V_D(\widetilde{\mathbb{M}}_t, \pi_{1,t}, \pi_{2,t}) - V_D(\mathbb{M}_\star, \pi_{1,t}, \pi_{2,t}) \right]$$

$$= 2 \sum_{t=1}^{T} \left[ V_D(\widetilde{\mathbb{P}}_t, \widetilde{f}^t, \pi_{1,t}, \pi_{2,t}) - V_D(\mathbb{P}_\star, f^\star, \pi_{1,t}, \pi_{2,t}) \right]$$

$$\leqslant 2 \sum_{t=1}^{T} \left[ V_D(\widetilde{\mathbb{P}}_t, \widetilde{f}^t, \pi_{1,t}, \pi_{2,t}) - V_D(\mathbb{P}_\star, \widetilde{f}^t, \pi_{1,t}, \pi_{2,t}) \right] + \left[ V_D(\mathbb{P}_\star, \widetilde{f}^t, \pi_{1,t}, \pi_{2,t}) - V_D(\mathbb{P}_\star, f^\star, \pi_{1,t}, \pi_{2,t}) \right]$$

$$\leqslant 2 \sum_{t=1}^{T} \left[ V_D(\widetilde{\mathbb{P}}_t, \widetilde{f}^t, \pi_{1,t}, \pi_{2,t}) - V_D(\mathbb{P}_\star, \widetilde{f}^t, \pi_{1,t}, \pi_{2,t}) \right] + \left[ V_D(\mathbb{P}_\star, \widetilde{f}^t, \pi_{1,t}, \pi_{2,t}) - V_D(\mathbb{P}_\star, f^\star, \pi_{1,t}, \pi_{2,t}) \right]$$

$$\overset{(i)}{=} 2 \sum_{t=1}^{T} \left[ V(\widetilde{\mathbb{P}}_t, \widetilde{f}^t, \pi_{1,t}) - V(\mathbb{P}_\star, \widetilde{f}^t, \pi_{1,t}) \right] - \left[ V(\widetilde{\mathbb{P}}_t, \widetilde{f}^t, \pi_{2,t}) - V(\mathbb{P}_\star, \widetilde{f}^t, \pi_{2,t}) \right]$$

$$+ \left[ V_D(\mathbb{P}_\star, \widetilde{f}^t, \pi_{1,t}, \pi_{2,t}) - V_D(\mathbb{P}_\star, f^\star, \pi_{1,t}, \pi_{2,t}) \right]$$

$$\overset{(ii)}{=} 2 \sum_{t=1}^{T} \left[ V(\widetilde{\mathbb{P}}_t, \widetilde{f}^t, \pi_{1,t}) - V(\mathbb{P}_\star, \widetilde{f}^t, \pi_{1,t}) \right] - \left[ V(\widetilde{\mathbb{P}}_t, \widetilde{f}^t, \pi_{2,t}) - V(\mathbb{P}_\star, \widetilde{f}^t, \pi_{2,t}) \right]$$

$$+ \left[ V(\mathbb{P}_\star \otimes \mathbb{P}_\star, \overline{f}^t, (\pi_{1,t}, \pi_{2,t})) - V(\mathbb{P}_\star \otimes \mathbb{P}_\star, \overline{f}^\star, (\pi_{1,t}, \pi_{2,t})) \right]$$

Where $(i)$ holds by the definition of $V_D$ and $V$, and $(ii)$ holds in the product MDP $\overline{\mathbb{M}}_\star$ once we define $\overline{f}^t_h((\tau_1, \tau_2)[h]) := \widetilde{f}^t_h(\tau_1[h]) - \widetilde{f}^t_h(\tau_2[h])$ and recall that $\overline{\mathbb{P}}_\star = \mathbb{P}_\star \otimes \mathbb{P}_\star$. Now, we can immediately apply Assumption 3 to the last line in two different ways. For the first two terms, we apply the first point in the assumption to each under cardinal feedback for MDP $\mathbb{M}_\star$, noting that the datasets $\mathcal{D}_t$ contain trajectories from $\pi_{1,t}$ as well as $\pi_{2,t}$. For the last term, we apply the second point in the assumption under cardinal feedback for the MDP $(\overline{\mathbb{P}}_\star, \overline{f}^\star)$.

This gives us that with probability $1 - \delta$,

$$\mathrm{Regret}(T) = \widetilde{\mathcal{O}}(C_P(\mathcal{M}, T, \delta) + C_F(\overline{\mathcal{M}}, T, \delta))$$

$\square$

We have the following lemma, which is an immediate consequence of

**Lemma 2** (Relating $\mathcal{F}$ and $\overline{\mathcal{F}}$). *For any function class $\mathcal{F}$, $\dim_E(\overline{\mathcal{F}}, \varepsilon) \leqslant 9 \dim_E(\mathcal{F}, \varepsilon/2)$.*

*Proof.* Let $\overline{d}_h = \dim_E(\overline{\mathcal{F}}_h, \varepsilon)$. Pick the $\varepsilon'$ so that there is a sequence of $\overline{d}_h$ pairs $\overline{\tau}_j, j = 1 \to \overline{d}_h$ of length $h$ trajectories, where each one is $\varepsilon'$-independent of its predecessors. Note that $\overline{\tau}_j = (\tau_{1,j}, \tau_{2,j})$. We now inductively build a sequence $i_j$ so that each $\tau_{i_j,j}$ is $\varepsilon'/2$-independent of its predecessors.

Pick the first $i_1$ arbitrarily. Now assume that we have built the sequence until index $j = k$. Also, by definition of this sequence, there exist $\overline{f}_j, \overline{f}'_j$, we have $\sqrt{\sum_{j=1}^{k} (\overline{f}_j(\overline{\tau}_j) - \overline{f}'_j(\overline{\tau}_j))^2} \leqslant \varepsilon'$ but $|\overline{f}_{k+1}(\overline{\tau}_j) - \overline{f}'_{k+1}(\overline{\tau}_j)| \geqslant \varepsilon'$. Since $a^2 + b^2 \leqslant 2(a+b)^2$, we have that

$$\sqrt{\sum_{j=1}^{k} (f_j(\tau_{i_j,j}) - f'_j(\tau_{i_j,j}))^2} \leqslant \sqrt{\sum_{j=1}^{k} (f_j(\tau_{i_j,j}) - f'_j(\tau_{i_j,j}))^2 + (f_j(\tau_{3-i_j,j}) - f'_j(\tau_{3-i_j,j}))^2}$$

$$\leqslant \sqrt{\sum_{j=1}^{k} 2(\overline{f}_j(\overline{\tau}_j) - \overline{f}'_j(\overline{\tau}_j))^2} \leqslant \sqrt{2}\varepsilon'$$

Additionally, since

$$|f_{k+1}(\tau_{1,k+1}) - f'_{k+1}(\tau_{1,k+1})| + |f_{k+1}(\tau_{2,k+1}) - f'_{k+1}(\tau_{2,k+1})| \geqslant |\overline{f}_{k+1}(\overline{\tau}_j) - \overline{f}'_{k+1}(\overline{\tau}_j)| \geqslant \varepsilon'$$

. So, there is an $i_{k+1}$ so that

$$|f_{k+1}(\tau_{i_{k+1},k+1}) - f'_{k+1}(\tau_{i_{k+1},k+1})| \geqslant \varepsilon'/2$$

So, we have a sequence $x_j := \tau_{i_j,j}$ and a sequence of pairs of functions $f_j, f'_j$ so that for any $1 \leqslant k \leqslant \overline{d}_h$, $\sum_{j=1}^k (f_j(x_j) - f'_j(x_j))^2 \leqslant 2(\varepsilon')^2$ but $|f_{k+1}(x_{k+1}) - f'_{k+1}(x_{k+1})| \geqslant \varepsilon'/2$. This implies the following. Inequality $(i)$ holds by Proposition 43 of Jin et al. (2021) upon setting $\beta = 2(\varepsilon')^2$ and setting the proposition's $\varepsilon$ to $\varepsilon'/2$. Inequality $(ii)$ holds since $\varepsilon'/2 \geqslant \varepsilon/2$.

$$\begin{aligned}
\overline{d}_h &= \sum_{j=1}^{\overline{d}_h} \mathbb{1}(|f_j(x_j) - f'_j(x_j)| \geqslant \varepsilon'/2) \\
&\overset{(i)}{\leqslant} \left( \frac{2(\varepsilon')^2}{(\varepsilon'/2)^2} + 1 \right) \dim_E(\mathcal{F}_h, \varepsilon/2) \\
&= 9 \dim_E(\mathcal{F}_h, \varepsilon'/2) \\
&\leqslant 9 \dim_E(\mathcal{F}_h, \varepsilon/2)
\end{aligned}$$

This establishes our claim. $\qquad\square$

We have the following immediate corollary of Theorem 3, Theorem 7 and Lemma 2.

**Corollary 1** (Dueling Regret using POR-UCRL Confidence Sets). *The confidence sets from POR-UCRL satisfy Assumption 2 and using them in Algorithm 1 leads to the following regret bound* $\text{Regret}_D(T) = \tilde{\mathcal{O}}\left( \left( pS\sqrt{HA} + \sum_{h \in \mathcal{H}_p} \sqrt{d_{E,h} d_{C,h}} \right) \sqrt{T} \right).$

## H.2 REDUCTION TO BONUS-BASED OPTIMISM

We define the reduction using the algorithm below.

---

**Algorithm 8** Reduction from Dueling to Cardinal Bonus-Based Optimism

---

1: **Input** Known reward function $\{r_h\}_{h=1}^H$, method $\mathtt{Est}(\mathcal{D})$ to estimate $\hat{\mathbb{P}}_{\mathcal{D}}$ and $\overline{f}_{\mathcal{D}}$ from dataset $\mathcal{D}$, bonus functions $b_{\overline{\mathcal{F}}}^{\mathcal{D}}(\mathbb{P}, \pi, \delta)$ and $b_{\mathcal{P}}^{\mathcal{D}}(\mathbb{P}, \pi, \delta)$, confidence level $\delta$.
2: Initialize dataset $\mathcal{D}_1 \leftarrow \{\}$
3: **for** $t = 1, ..., T$ **do**
4:     Compute good set $\Pi_t$                                       {Valid $\pi_\star$ candidates}

$$\Pi_t := \Big\{ \pi \in \Pi \Big| V_D((\hat{\mathbb{P}}_{\mathcal{D}_t}, \overline{f}_{\mathcal{D}_t}), \pi, \pi_1) + b_{\overline{\mathcal{F}}}(\hat{\mathbb{P}}_{\mathcal{D}_t}, (\pi, \pi_1), \delta)$$
$$+ z(Bp)b_{\mathcal{P}}(\hat{\mathbb{P}}_{\mathcal{D}_t}, \pi, \delta) + z(Bp)b_{\mathcal{P}}(\hat{\mathbb{P}}_{\mathcal{D}_t}, \pi_1, \delta) \geqslant 0, \ \forall \pi_1 \in \Pi \Big\}$$

5:     Pick $(\pi_{1,t}, \pi_{2,t})$ given by                          {Most uncertain duel}

$$\arg\max_{\pi, \pi' \in \Pi_t} b_{\overline{\mathcal{F}}}(\hat{\mathbb{P}}_{\mathcal{D}_t}, (\pi, \pi'), \delta) + z(Bp)b_{\mathcal{P}}(\hat{\mathbb{P}}_{\mathcal{D}_t}, \pi, \delta) + z(Bp)b_{\mathcal{P}}(\hat{\mathbb{P}}_{\mathcal{D}_t}, \pi_1, \delta)$$

6:     Collect trajectories $\tau_{t,i} = \Big\{ (s_{h,i}^t, a_{h,i}^t)) \Big\}_{h=1}^H$ along with feedback $\{o_h\}_{h \in \mathcal{H}_p}$ by sampling from $\mathbb{P}_\star^{\pi_{i,t}}$ for $i = 1, 2$.
7:     Append the data to $\mathcal{D}_t$ to get $\mathcal{D}_{t+1}$, update estimates and bonuses.
8: **end for**

---

**Theorem 13** (Reduction from Dueling to Bonus-Based Optimism). *If the bonuses and estimates used in Algorithm 8 satisfy Assumption 3, then with probability $1 - \delta$, the dueling regret $\mathrm{Regret}_D(T)$ of Algorithm 8 is given by*

$$\mathrm{Regret}_D(T) = \widetilde{\mathcal{O}}(C_P(\mathcal{M}, T, \delta) + C_F(\overline{\mathcal{M}}, T, \delta))$$

*Proof.* Recall that the value of a duel $(\pi, \pi')$ under model $\overline{\mathbb{M}} \leftrightarrow \mathbb{M} \leftrightarrow (\mathbb{P}, f)$ is denoted by

$$V_D(\overline{\mathbb{M}}, \pi, \pi') := V(\mathbb{M}, \pi) - V(\mathbb{M}, \pi') = V(\mathbb{P}, f, \pi) - V(\mathbb{P}, g, \pi')$$

We overload notation and use the natural bijection $\mathcal{M} \leftrightarrow \overline{\mathcal{M}}$ to define

$$V_D(\mathbb{M}, \pi, \pi') := V_D(\overline{\mathbb{M}}, \pi, \pi')$$

For ease of notation in the proof, we often work with an arbitrary pre-image $\hat{f}_{\mathcal{D}}$ of $\overline{f}_{\mathcal{D}}$ under the map $f \mapsto \overline{f}$. A careful read will confirm that this does not affect the correctness of any of the statements. First note that $\pi_\star \in \Pi_t$ for all $T$ with probability $1 - \delta/16$ since the following hold uniformly over all $\pi_1 \in \Pi$

$$-V_D((\hat{\mathbb{P}}_{\mathcal{D}_t}, \hat{f}_{\mathcal{D}_t}), \pi_\star, \pi_1) = V(\hat{\mathbb{P}}_{\mathcal{D}_t}, \hat{f}_{\mathcal{D}_t}, \pi_1) - V(\hat{\mathbb{P}}_{\mathcal{D}_t}, \hat{f}_{\mathcal{D}_t}, \pi_\star)$$
$$= \Big[ V(\hat{\mathbb{P}}_{\mathcal{D}_t}, \hat{f}_{\mathcal{D}_t}, \pi_1) - V(\mathbb{P}_\star, \hat{f}_{\mathcal{D}_t}, \pi_1) \Big] - \Big[ V(\mathbb{P}_\star, \hat{f}_{\mathcal{D}_t}, \pi_1) - V(\hat{\mathbb{P}}_{\mathcal{D}_t}, \hat{f}_{\mathcal{D}_t}, \pi_1) \Big]$$
$$+ V(\mathbb{P}_\star, f^\star, \pi_1) - V(\mathbb{P}_\star, f^\star, \pi_\star)$$
$$+ V_D((\mathbb{P}_\star, \overline{f}_{\mathcal{D}_t}), \pi_1, \pi_\star) - V_D((\mathbb{P}_\star, f^\star), \pi_1, \pi_\star)$$
$$\leqslant z(Bp)b_{\mathcal{P}}(\hat{\mathbb{P}}_{\mathcal{D}_t}, \pi_\star, \delta) + z(Bp)b_{\mathcal{P}}(\hat{\mathbb{P}}_{\mathcal{D}_t}, \pi_1, \delta)$$
$$+ 0$$
$$+ b_{\overline{\mathcal{F}}}(\hat{\mathbb{P}}_{\mathcal{D}_t}, (\pi_\star, \pi_1), \delta)+$$
$$= b_{\overline{\mathcal{F}}}(\hat{\mathbb{P}}_{\mathcal{D}_t}, (\pi_\star, \pi_1), \delta) + z(Bp)b_{\mathcal{P}}(\hat{\mathbb{P}}_{\mathcal{D}_t}, \pi_\star, \delta) + z(Bp)b_{\mathcal{P}}(\hat{\mathbb{P}}_{\mathcal{D}_t}, \pi_1, \delta)$$

where the inequality holds by Assumption 4 and the optimality of $\pi_\star$ in the true model. Note let $\hat{\mathbb{M}}_t$ be the model given by $\hat{\mathbb{P}}_{\mathcal{D}_t}, \hat{f}_{\mathcal{D}_t}$ and let $\overline{\mathbb{M}}_t$ be the corresponding model in $\overline{\mathcal{M}}$. We make the following

abbreviation:

$$b_{\overline{\mathcal{M}}}(\overline{\mathbb{M}}_t, (\pi, \pi'), \delta) := b_{\overline{\mathcal{F}}}(\hat{\mathbb{P}}_{\mathcal{D}_t}, (\pi, \pi'), \delta) + z(Bp)b_{\mathcal{P}}(\hat{\mathbb{P}}_{\mathcal{D}_t}, \pi, \delta) + z(Bp)b_{\mathcal{P}}(\hat{\mathbb{P}}_{\mathcal{D}_t}, \pi', \delta)$$

$$\text{Regret}_D(T) = \sum_{t=1}^{T}\sum_{i=1}^{2} V_D(\mathbb{M}_\star, \pi_\star, \pi_{i,t})$$

$$= \sum_{t=1}^{T}\Big[\sum_{i=1}^{2} V_D(\mathbb{M}_\star, \pi_\star, \pi_{i,t}) - V_D(\hat{\mathbb{M}}_t, \pi_\star, \pi_{i,t}) + b_{\overline{\mathcal{M}}}(\overline{\mathbb{M}}_t, (\pi_\star, \pi_{i,t}), \delta)$$

$$+ \sum_{i=1}^{2} V_D(\hat{\mathbb{M}}_t, \pi_\star, \pi_{i,t}) - b_{\overline{\mathcal{M}}}(\overline{\mathbb{M}}_t, (\pi_\star, \pi_{i,t}), \delta)\Big]$$

$$\overset{(i)}{\leqslant} \sum_{i=1}^{2}\sum_{t=1}^{T} \Big[V_D(\mathbb{M}_\star, \pi_\star, \pi_{i,t}) - V_D(\hat{\mathbb{M}}_t, \pi_\star, \pi_{i,t}) + b_{\overline{\mathcal{M}}}(\overline{\mathbb{M}}_t, (\pi_\star, \pi_{i,t}), \delta)\Big]$$

Inequality $(i)$ holds since $V_D(\hat{\mathbb{M}}_t, \pi_\star, \pi_{i,t}) = -V_D(\hat{\mathbb{M}}_t, \pi_{i,t}, \pi_\star)$, and $\pi_{i,t} \in \Pi_t$ for $i = 1, 2$ implies that

$$V_D(\hat{\mathbb{M}}_t, \pi_{i,t}, \pi_\star) + b_{\overline{\mathcal{F}}}(\hat{\mathbb{P}}_{\mathcal{D}_t}, (\pi_\star, \pi_1), \delta) + z(Bp)b_{\mathcal{P}}(\hat{\mathbb{P}}_{\mathcal{D}_t}, \pi_\star, \delta) + z(Bp)b_{\mathcal{P}}(\hat{\mathbb{P}}_{\mathcal{D}_t}, \pi_1, \delta) \geqslant 0$$

Now note that the following holds uniformly over all timesteps $t$ with probability $1 - 3\delta/8$ for $i = 1, 2$ simultaneously using Assumption 4 multiple times and applying a union bound.

$$V_D(\mathbb{M}_\star, \pi_\star, \pi_{i,t}) - V_D(\hat{\mathbb{M}}_t, \pi_\star, \pi_{i,t}) = V_D((\mathbb{P}_\star, \overline{f}^\star), \pi_\star, \pi_{i,t}) - V_D((\hat{\mathbb{P}}_{\mathcal{D}_t}, \overline{f}_{\mathcal{D}_t}), \pi_\star, \pi_{i,t})$$

$$= V_D((\mathbb{P}_\star, \overline{f}^\star), \pi_\star, \pi_{i,t}) - V_D((\hat{\mathbb{P}}_{\mathcal{D}_t}, \overline{f}^\star), \pi_\star, \pi_{i,t})$$

$$+ V_D((\hat{\mathbb{P}}_{\mathcal{D}_t}, \overline{f}^\star), \pi_\star, \pi_{i,t}) - V_D((\hat{\mathbb{P}}_{\mathcal{D}_t}, \overline{f}_{\mathcal{D}_t}), \pi_\star, \pi_{i,t})$$

$$= V(\mathbb{P}_\star, \overline{f}^\star, \pi_\star) - V(\hat{\mathbb{P}}_{\mathcal{D}_t}, \overline{f}^\star, \pi_\star) + V(\hat{\mathbb{P}}_{\mathcal{D}_t}, \overline{f}^\star, \pi_{i,t}) - V(\mathbb{P}_\star, \overline{f}^\star, \pi_{i,t})$$

$$+ V_D((\hat{\mathbb{P}}_{\mathcal{D}_t}, \overline{f}^\star), \pi_\star, \pi_{i,t}) - V_D((\hat{\mathbb{P}}_{\mathcal{D}_t}, \overline{f}_{\mathcal{D}_t}), \pi_\star, \pi_{i,t})$$

$$= z(Bp)b_{\mathcal{P}}(\hat{\mathbb{P}}_{\mathcal{D}_t}, \pi_\star, \delta) + z(Bp)b_{\mathcal{P}}(\hat{\mathbb{P}}_{\mathcal{D}_t}, \pi_{i,t}, \delta)$$

$$+ b_{\overline{\mathcal{F}}}(\hat{\mathbb{P}}_{\mathcal{D}_t}, (\pi_\star, \pi_{i,t}), \delta)$$

$$= b_{\overline{\mathcal{M}}}(\overline{\mathbb{M}}_t, (\pi_\star, \pi_{i,t}), \delta)$$

So, with probability $1 - 3\delta/16$, we have that

$$\text{Regret}_D(T) \leqslant \sum_{t=1}^{T}\sum_{i=1}^{2} b_{\overline{\mathcal{M}}}(\overline{\mathbb{M}}_t, (\pi_\star, \pi_{i,t}), \delta)$$

$$\overset{(i)}{\leqslant} 2\sum_{t=1}^{T} b_{\overline{\mathcal{M}}}(\overline{\mathbb{M}}_t, (\pi_{1,t}, \pi_{2,t}), \delta)$$

$$= 2\sum_{t=1}^{T} z(Bp)b_{\mathcal{P}}(\hat{\mathbb{P}}_{\mathcal{D}_t}, \pi_{1,t}, \delta) + z(Bp)b_{\mathcal{P}}(\hat{\mathbb{P}}_{\mathcal{D}_t}, \pi_{2,t}, \delta) + b_{\overline{\mathcal{F}}}(\hat{\mathbb{P}}_{\mathcal{D}_t}, (\pi_{1,t}, \pi_{2,t}), \delta)$$

$$\overset{(ii)}{\leqslant} \tilde{\mathcal{O}}\left(\sum_{t=1}^{T}(z_1(Bp)b_{\mathcal{P}}(\mathbb{P}_\star, \pi_{1,t}, \delta) + z_1(Bp)b_{\mathcal{P}}(\mathbb{P}_\star, \pi_{2,t}, \delta) + b_{\overline{\mathcal{F}}}(\mathbb{P}_\star, (\pi_{1,t}, \pi_{2,t}), \delta)\right)$$

where inequality $(i)$ holds since $(\pi_{1,t}, \pi_{2,t}) = \arg\max_{\pi, \pi' \in \Pi_t} b_{\overline{\mathcal{M}}}(\overline{\mathbb{M}}_t, (\pi, \pi'), \delta)$ and inequality $(ii)$ holds with probability $1 - 3\delta/8$ by 6 applications of the change of measure inequality in Assumption 4.

Now, we can use the fact that Assumption 4 is satisfied again to conclude that with probability $1 - \delta/32$.

$$\sum_{t=1}^{T}(z_1(Bp)b_{\mathcal{P}}(\mathbb{P}_\star, \pi_{1,t}, \delta) + z_1(Bp)b_{\mathcal{P}}(\mathbb{P}_\star, \pi_{2,t}, \delta) + b_{\overline{\mathcal{F}}}(\mathbb{P}_\star, (\pi_{1,t}, \pi_{2,t}), \delta) = \tilde{\mathcal{O}}(C_P(\mathcal{M}, T, \delta) + C_F(\overline{\mathcal{M}}, T, \delta))$$

Taking a union bound over all inequalities stated so far, we have the following with probability $1 - \delta$

$$\text{Regret}_D(T) = \widetilde{\mathcal{O}}(C_P(\mathcal{M}, T, \delta) + C_F(\overline{\mathcal{M}}, T, \delta))$$

as desired. $\qquad \square$

Again, the following corollary is immediate from Theorem 5, Theorem 9 and Lemma 2.

**Corollary 8.** *By using POR-UCBVI as the algorithm in the dueling reduction in Algorithm 8, we can get a bound on the dueling regret given by*

$$\text{Regret}_D(T) = \widetilde{\mathcal{O}}\left(\left(pC(H, S, A) + \sum_{h \in \mathcal{H}_p} \sqrt{\bar{d}_{E,h}(d_{C,h} + H)}\right)\sqrt{T}\right)$$

*where $\bar{d}_{E,h} = \dim_E\left(\mathcal{F}_h, \frac{B}{2T}\right)$.*

