# OpenReview forum: "A Theoretical Framework for Partially-Observed Reward States in RLHF"
_ICLR.cc/2025/Conference — ICLR 2025 Poster_

### Official Review · Reviewer_ExHp · 2024-10-30

**Soundness:** 3
**Presentation:** 2
**Contribution:** 2
**Rating:** 5
**Confidence:** 3

**Summary:**

This paper studies the problem of reinforcement learning with partially observed reward-states (PORRL). The authors accommodate two kinds of feedback – cardinal and dueling feedback. The authors first demonstrate that PORRL subsumes a wide class of RL problems, including traditional RL, RLHF, and reward machines. For cardinal feedback, the authors present two model-based methods (POR-UCRL, POR-UCBVI). The authors give both cardinal regret and sample complexity guarantees for the methods, showing that they improve over naive history-summarization. The authors then discuss the benefits of a model-free method like GOLF with naive history-summarization in settings with recursive internal states and dense intermediate feedback. For this purpose, the authors define a new history aware version of the Bellman-eluder dimension and give a new guarantee for GOLF in the setting of this paper, which can be exponentially sharper in illustrative examples. For dueling feedback, the authors show that a naive reduction to cardinal feedback fails to achieve sublinear dueling regret. The authors then present the first explicit reduction that converts guarantees for cardinal regret to dueling regret. In both feedback settings, the authors show that the proposed models and guarantees generalize and extend existing ones.

**Strengths:**

1.	The authors introduce PORRL, which generalizes current RLHF models to incorporate “internal states” and intermediate feedback.
2.	The authors design model-based optimistic algorithms that, POR-UCRL and POR-UCBVI, and provide regret guarantees under naive history-summarization. The authors further show that their guarantees subsume and improve over past results.
3.	The authors study the model-free algorithm GOLF, applied using history-summarization. The authors establish a regret guarantee for GOLF. The authors also show that when internal states have a recursive structure, their guarantee can be exponentially smaller than existing guarantees and guarantees for their model-based methods.
4.	The authors show that a naive blackbox reduction from dueling to cardinal PORRL always fails. They design a whitebox reduction from dueling PORRL to a large class of optimistic algorithms for cardinal PORRL.

**Weaknesses:**

1.	The writing of this paper needs improvement. This paper is too dense, and hard to follow.
2.	There is no experiment provided. It is hard to validate the effectiveness and implementability of the proposed algorithm, and the provided theoretical results.
3.	The authors should elaborative more on the motivation of combining POMDPs and RLHF, e.g., give 1-2 concrete application scenarios.
4.	The authors should discuss more on the technical novelty in combining POMDPs and RLHF. It looks like a combination of existing POMDP works and preference feedback.

**Questions:**

Please see the weaknesses above.

---

> ### Author Response · Authors · 2024-11-20
> **Response to Reviewer ExHP**
>
> We are happy that you appreciate many aspects of our paper:
>
> 1. Our generalization of current RLHF models to incorporate internal states and intermediate feedback.
> 2. The novelty of our algorithms and the improvement our guarantees provide over existing guarantees.
> 3. Our new regret guarantee for GOLF and the exponential improvement that we demonstrate over existing guarantees and guarantees for model based methods under a recursive structure on internal states.
> 4. Our demonstration that a naive blackbox reduction fails and our whitebox reduction from cardinal regret guarantees to dueling regret guarantees.
>
> We would like to address your qualms below. **If you feel that our responses adequately address your qualms, we implore you to consider increasing your score.**
>
> ## Weaknesses
>
> 1. **Writing:** We agree with your prioritization of good writing. Despite the wealth of content in the paper, we have tried our best to fit in all relevant information in the main paper in a structured manner, while also being readable and welcoming. We try to complement every mathematically heavy definition or result with an intuitive explanation either just before or just after it. We feel that we have managed to strike a good balance between content and writing, but we are very open to feedback. Are there specific paragraphs that you found hard to follow? We are happy to address any specific concerns that you have.
>
> 2. **Experiments and practicality:** We would like to address your concern in three ways:
>
>     a. **Practical implications:** Despite the theoretical nature of our work, we provide concrete practical takeaways in lines 522-536 in section 5.
>
>     b. **Theoretical focus:** As the reviewer must have noticed, this is primarily a theoretical paper providing insights for future practical work. So, we believe that discussing the implications of our theory already completes the arc of the paper’s story. Adding a practical algorithm would be both beyond the scope of the paper and would also take too much space in an already crowded paper. While we agree that ML research should eventually lead to some real-life contributions, we believe that a theoretical paper that opens the door for future practical work is an important first step towards such a goal.
>
>     c. **Ongoing follow-up work:** However, we would like to reassure the reviewer that we have ongoing practical work extending online iterative RLHF under DPO and PPO to intermediate feedback. In the intermediate feedback setting, it turns out that PPO learns the raw rewards and serves as a model-based algorithm and DPO implicitly goes through the Q function and serves as a model-free algorithm.
>
> 3. **Motivation, examples and necessity are discussed in the paper:** We would like to make three points:
>
>     a. **We are not combining POMDPs and RLHF:** We would like to first emphasize that the paper is not combining POMDPs and RLHF at all. We work with partially-observed rewards, but as we discuss in lines 303-310, our setting is emphatically different from POMDPs and cannot be addressed using POMDP frameworks and algorithms. We discuss this further in point 4 too.
>
>     b. **Motivation and examples for PORRL are given in the paper:** Figure 1 as well as lines 140-157 and lines 178-196 in section 2 are all examples of the PORRL problem, making a total of 7 fairly general examples as well as one specific example of the combination lock. The strongest example of partially-observed rewards (addressed in figure 1 and lines 140-157) is that of a human interacting with an LLM, where the internal state is the human’s emotions about the LLM’s responses or the human’s level of confidence in the correctness of the response. Feedback is received in a preferential or good/bad fashion. We also mention in lines 178-196 that the PORRL framework subsumes traditional RL, other models of once-per-episode feedback, learning reward machines, etc.
>
>     c. **Need for new algorithms discussed in the paper:** We discuss in the paper why existing work cannot apply to the PORRL setting in lines 301-321. Traditional RL, POMDPs, naive history summarization, all either do not apply or are statistically inefficient. Further there does not exist an understanding of general methods that can satisfactorily utilize additional structure to internal states, like we show for model-based methods.

---

> > ### Author Response · Authors · 2024-11-20
> > **Response to Reviewer ExHP continued**
> >
> > 4. **Technical novelty:** We would like to make four points:
> >
> >     a. **We are not combining POMDPs and RLHF:** As we mentioned in the previous point, we are in fact not combining POMDPs and RLHF at all, as we discuss in lines 303-310. We deal with PORMDPs, which are not a subcase of POMDPs since $s, u, a \to s’, u’$ is not a Markovian transition. Even if we did consider the subcase of PORMDPs where this assumption holds, they would form an overcomplete POMDP. Literature on overcomplete POMDPs is scarce and existing work on them does not apply to our setting at all.
> >
> >     b. **Technical novelty in POR-UCRL and POR-UCBVI:** Our work seeks to handle non-Markovian reward models along with a Markovian transition structure in a statistically efficient fashion. There are many challenges with this, and we discuss three of the biggest challenges with giving guarantees for POR-UCRL and POR-UCBVI in lines 358-363. We also discuss how we overcome these challenges in our response to Question 1 from reviewer 4dew. These challenges are unique to our delicate balance of non-Markovian rewards and Markovian transitions.
> >
> >     c. **Technical novelty in guarantees for GOLF:** We have to provide an entirely new regret guarantee for GOLF to show the exponential statistical speed-up obtained when internal states have a recursive structure. As you have seen and stated in your review, this also involves designing a new history-aware notion of dimension that is much smaller than the usual Bellman-eluder dimension. These challenges are completely unrelated to challenges faced in work on POMDPs.
> >
> >     d. **Technical novelty in dueling regret reduction:** We provide the first explicit reduction from cardinal regret guarantees to dueling regret guarantees. The challenges in coming up with such a general whitebox reduction involve extracting and formalizing intuition provided in previous work on dueling RL (which is also unrelated to POMDPs).
> >
> > ## Questions
> >
> > Same as weaknesses.

---

> ### Comment · Reviewer_ExHp · 2024-11-25
> **Thank you for your response**
>
> Thank you for your response.
>
> I understand this paper is a theoretical work. The proposed algorithms in this paper are mainly based on algorithms in prior theoretical RL works, e.g., GOLF, which are not very implemental. Given that the motivating and application scenario of this paper is LLMs, even for a theoretical work, it is also helpful and important to provide simulations to validate their algorithms or ideas. The practical takeaways provided in this paper still look like some conjectures. It is unclear how the proposed algorithms and theoretical results can be applied to or provide insights and guidence for practice.
>
> I tend to maintain my score currently, and will participate in the discussion with other reviewers and AC.

---

> ### Author Response · Authors · 2024-11-25
> **Thank you for engaging with our response**
>
> Thank you for engaging with our response, we are grateful for your reply. We are glad that you seem satisfied with the motivation behind our framework and the technical novelty of our work after our response. **It seems like your main remaining concern is that this is theoretical work.** We would like to emphasize a few additional points, and hope that you have productive discussion with other reviewers and the AC:
>
> 1. **Other purely theoretical published work on RLHF:** There exists other purely theoretical work (e.g. [1,2,3]) in RLHF and preference-based reinforcement learning, published at ICLR and NeurIPS previously. The work also has no experiments, and unlike us, they do not even discuss practical implications of their work. Nevertheless, the value of their theoretical contributions has been considered enough to make them worthy of publication.
> 2. **This is an already crowded paper, cannot add experiments:** As you pointed out in your own review, the paper is currently packed with a lot of content, theory and insights. It would be near impossible to add and discuss experiments, given the page constraints of ML conference papers.
> 3. **Application of this paper to current experimental work and ongoing follow-up work:** As we mentioned in our response above, we have ongoing experimental follow-up work that already demonstrates the potential of this paper to impact future practical work. While adding that experimental work to this paper would be near impossible due to space constraints, we noted in our comment that using PPO with a reward model that updates iteratively would behave like a model-based method. The experimental papers [4,5] learn such a reward model from intermediate feedback in an offline setting. We are trying to demonstrate that in the realistic online iterative setting, it is statistically (exponentially) more efficient to use DPO, which behaves like a model-free method by implicitly learning a Q-function. This insight is derived from our theoretical work in this paper, and should help us understand future experimental observations in RLHF under intermediate feedback. Such insights from our paper will help researchers make careful algorithmic choices in practice.
>
> ### References
>
> 1. Is RLHF More Difficult than Standard RL? A Theoretical Perspective. Wang et al, NeurIPS 2023.
> 2. Provable Reward-Agnostic Preference Based Reinforcement Learning. Zhan et al, ICLR 2024.
> 3. Provable Offline Preference-Based Reinforcement Learning. Zhan et al, ICLR 2024.
> 4. Enhancing Multi-Step Reasoning via Process-Supervised Reinforcement Learning from Human Feedback. Anonymous Authors, 2024 (ACL Rolling Review submission).
> 5. STEP-RLHF: Step-wise Reinforcement Learning from Human Feedback. Anonymous Authors, 2024 (ACL Rolling Review submission).

---

### Official Review · Reviewer_4dew · 2024-11-04

**Soundness:** 3
**Presentation:** 3
**Contribution:** 3
**Rating:** 6
**Confidence:** 3

**Summary:**

This paper provides a general formulation for RLHF by introducing humans' internal states. This new framework allows reward and policy to be non-Markovian but still tractable. The authors propose algorithms for both cardinal and dueling feedback and obtain improved regret compared with previous methods. Interestingly, to design model-free algorithms for cardinal feedback, the authors introduce a modified Bellman-eluder dimension, which can be exponentially smaller than the standard Bellman-eluder dimension for the combination lock example. In the end, the authors also propose the first explicit reduction that converts guarantees for cardinal regret to dueling regret.

**Strengths:**

1. This paper provides a general RLHF framework covering many important instances.
2. The proposed algorithms lead to improved regret compared with existing results,
3. The authors propose the first explicit reduction that converts guarantees for cardinal regret to dueling regret.
4. A new complexity measure called HABE is proposed, which can be exponentially smaller than the standard Bellman-eluder dimension for the combination lock example.

**Weaknesses:**

1. The proposed algorithms follow existing frameworks, which are generally computationally inefficient.

2. This paper assumes the internal state generation function $g$ is deterministic. Although this seems reasonable for rigorous math reasoning, it may not capture the complexity of human feedback for more ambiguous casual conversations.

**Questions:**

1. On Page 7, three main technical challenges for model-based algorithms are mentioned. Could the authors give brief explanations of how the new design tackles these challenges?

2. The model-free algorithms follow GOLF, which requires Bellman completeness (BC). Does the adaptation here also need BC? If so, I think it is better to explicitly mention it in the main text.

3. For cardinal feedback, the paper assumes it is emitted according to a Bernoulli distribution. Is this assumption also used in previous papers for cardinal feedback? It seems the KTO paper utilizes a more complex feedback model?

---

> ### Author Response · Authors · 2024-11-20
> **Response to Reviewer 4dew**
>
> We are happy that you appreciate many aspects of our paper:
>
> 1. The generality of our framework.
> 2. The improvement that we provide over existing regret bounds.
> 3. The fact that we propose the first explicit reduction converting guarantees from cardinal to dueling regret.
> 4. The novelty of our proposed complexity measure HABE, and its exponential improvement over existing complexity measures.
>
> We would like to address your qualms below. **If you feel that our responses adequately address your qualms, we implore you to consider increasing your score.**
>
> ## Weaknesses
>
> 1. **Computational efficiency:** We appreciate your concern about practicality, and we would like to address your concerns in three ways:
>
>     a. **Practical implications:** Despite the theoretical nature of our work, we provide concrete practical takeaways in lines 522-536 in section 5.
>
>     b. **Theoretical focus:** As the reviewer must have noticed, this is primarily a theoretical paper providing insights for future practical work. So, we believe that discussing the implications of our theory already completes the arc of the paper’s story. Adding a practical algorithm would be both beyond the scope of the paper and would also take too much space in an already crowded paper. While we agree that ML research should eventually lead to some real-life contributions, we believe that a theoretical paper that opens the door for future practical work is an important first step towards such a goal.
>
>     c. **Ongoing follow-up work:** However, we would like to reassure the reviewer that we have ongoing practical work extending online iterative RLHF under DPO and PPO to intermediate feedback. In the intermediate feedback setting, it turns out that PPO learns the raw rewards and serves as a model-based algorithm and DPO implicitly goes through the Q function and serves as a model-free algorithm.
>
> 2. **The g functions can be stochastic:** We thank the reviewer for pointing this out. While the $g$ function was assumed to be deterministic for simplicity of exposition, all the theory follows verbatim if we assume that $f(\tau[h]) - E[f(\tau[h])$ is $\eta_h$ subgaussian given $\tau[h]$, even if $g$ is stochastic. We have added a footnote to this effect in line 322.
>
> ## Questions
>
> 1. We are happy to answer how we tackle each challenge:
>
>     a. **Non markovian rewards and markovian transitions:** In Appendix C, we decompose regret for generic model-based algorithms in a way that allows us to separate the error arising from probability transitions and that from reward models.
>
>     b. **Uniform bonus across all policies:** This involves a telescoping sum trick borrowed from Lemma B.2 of [1].
>
>     c. **General function approximation for f:** We adapt ideas from existing proofs involving the eluder dimension to our non-Markovian reward setting, noticing that the core notion of the eluder dimension still applies to the function classes $\mathcal{F}_h$ for $h = 1 \to H$.
>
> 2. **Adding the Bellman completeness assumption:** We thank the reviewer for pointing this out. We do indeed need Bellman completeness and we have modified the statement of Theorem 2 to explicitly mention this in lines 415-416.
>
> 3. **Our cardinal feedback model is much more general than Bernoulli:** Our cardinal feedback model is not just restricted to Bernoulli feedback. As you can see in Definition 1, our feedback is $\eta_h$ subgaussian with mean $\sigma_h(r_h)$. This includes Bernoulli 0/1 feedback, Gaussian numerical feedback, bounded stochastic feedback, and most reasonable feedback models. This subsumes feedback models in all previous papers that mention an explicit cardinal feedback model in RLHF. Our model is a direct generalization of such a model in [1, 2]. The KTO paper [3] doesn’t explicitly mention a generative model for cardinal feedback - it mentions a generative model only for preferential feedback, saying that it is emitted according to the Bernoulli distribution. The paper then defines a loss function for cardinal feedback.
>
> ### References
>
> 1. Chatterji et al, 2021. On the theory of reinforcement learning with once-per-episode feedback.
> 2. Efroni et al, 2021. Reinforcement learning with trajectory feedback.
> 3. Ethayarajh et al, 2024. KTO: Model Alignment as Prospect Theoretic Optimization.

---

> > ### Comment · Reviewer_4dew · 2024-11-25
> >
> > I thank the authors for the detailed response. I have no further questions and will keep my positive score.

---

### Official Review · Reviewer_Rhuq · 2024-11-06

**Soundness:** 4
**Presentation:** 3
**Contribution:** 4
**Rating:** 8
**Confidence:** 4

**Summary:**

The paper introduces PORRL a framework that adds internal states and intermediate feedback into Reinforcement Learning from Human Feedback, which can speedup alignment. The authors provide two model-based algorithms POR-UCRL, POR-UCBVI for learning with cardinal feedback, and provide regret bounds for GOLF (a prior model-free algorithm) based on a new history aware bellman eluder dimension “HABE”. They also provide a reduction from duelling to cardinal PORRL.

**Strengths:**

- The PORRL framework is novel, and their modelling is highly general with non-Markovian transitions. The paper is technically solid and a good contribution in the space of theoretical guarantees for learning from Human feedback.
- The authors analyse both model-based and model-free algorithms with regret guarantees under cardinal and duelling feedback.

**Weaknesses:**

The practical takeaways in the conclusion are mainly phrased as conjectures, and the authors do not provide a concrete algorithm within the PORRL framework. Given the strong theoretical contributions, a practical algorithm would have strengthened the work further.

**Questions:**

Intermediate feedback at each step requires substantially more human queries than single preferential feedback at the end of an episode. Can you compare your framework's query complexity compared to that in [1, 2]?

[1] How to Query Human Feedback Efficiently in RL?

[2] Reinforcement Learning from Human Feedback with Active Queries

---

> ### Author Response · Authors · 2024-11-20
> **Response to Reviewer Rhuq**
>
> Thank you for your review! We are grateful to you for your confidence in our work, and we are glad that you appreciate:
>
> 1. The generality and novelty of our framework and modeling.
> 2. The technical strength of our paper.
> 3. The quality of our contribution to theoretical guarantees for learning from human feedback.
> 4. The comprehensive nature of our study of model-free and model-based approaches under both cardinal and dueling feedback.
>
> We would like to address the two qualms raised in your review:
>
> ## Weaknesses
>
> 1. **Practical algorithms:** We appreciate that the reviewer noticed the practical takeaways in our paper. We would like to address their concerns in two ways:
>
>     a. **Theoretical focus:** As the reviewer has noticed, this is primarily a theoretical paper providing insights for future practical work. So, we believe that discussing the implications of our theory already completes the arc of the paper’s story. Adding a practical algorithm would be both beyond the scope of the paper and would also take too much space in an already crowded paper.
>
>     b. **Ongoing follow-up work:** However, we would like to reassure the reviewer that we have ongoing practical work extending online iterative RLHF under DPO and PPO to intermediate feedback. In the intermediate feedback setting, it turns out that PPO learns the raw rewards and serves as a model-based algorithm and DPO implicitly goes through the Q function and serves as a model-free algorithm.
>
> ## Questions
>
> 1. **Query complexity:** We have three parts to our response:
>
>     a. **We improve over the query complexity guarantees of [1]:** In their setting, due to the known featurization $\phi$ determining rewards $r_h(s,a)$, we can show that $d_{HABE} = d_E = d$ and all covering dimensions are $d$ up to logarithmic terms. In that case, our model-free guarantees are $H^2d^2/\epsilon^2$, while theirs are $H^4d^2/\epsilon^2$. So, our guarantees are an improvement over them. On the other hand, our model-based guarantees have an extra $HS^2A/\epsilon^2$ term.
>
>     b. **We match the query complexity guarantees of [2]:** Crucially, the rewards in [2] are a linear function of a featurization of the prompt $x$ and the full response $y$, without any dependence on the horizon or on the intermediate states and actions that come up while generating response. The dimension $d$ of the reward parameter in their setting would thus be much larger than $d_{E}$ or $d_{HABE}$ in our setting. If we were to restrict to their linear model and work in their once-per-episode feedback setting, then we can argue that $d = Hd_{HABE}$ and $d = Hd_E$ are the appropriate relations, in which case we recover the same query complexity guarantees as them using our analysis of GOLF. For our model-based methods, we have an extra $HS^2A/\epsilon^2$ term coming from model-learning.
>
>     c. **General query complexity:** Outside of our comparison with algorithms in [1,2], the query complexity for getting feedback at $p$ points in a trajectory for $1<p\leq H$ is merely $p$ times the sample complexity of our algorithms. This is often much less than the query complexity of a once-per-trajectory feedback algorithm. For example, in the combination lock, the analysis of the model-free approach shows that we can reduce the query complexity from $H^3A^H/\epsilon^2$ for P-OMLE to $AH^6/\epsilon^2$, ignoring logarithmic terms.
>
> ### References
>
> 1. Zhan et al, 2023. How to Query Human Feedback Efficiently in RL?
> 2. Ji et al, 2024. Reinforcement Learning from Human Feedback with Active Queries.

---

> > ### Comment · Reviewer_Rhuq · 2024-11-25
> >
> > Thanks for addressing my questions, I will retain my positive rating for the paper.

---

### Author Response · Authors · 2024-11-21
**Responses to Reviews Added, Paper Updated on OpenReview**

We are grateful to all the reviewers for taking out time to review the paper. We have responded to all the reviews and we want to highlight that while we explicitly mention this in one review, we have made minor updates the paper on OpenReview. **All line numbers mentioned in our responses to reviews are with respect to the new version of the paper on OpenReview.**

If the reviewers feel that our responses adequately address their qualms, we implore them to consider increasing their scores.

---

### Meta-Review · Area_Chair_wt8k · 2024-12-21

**Metareview:**

This paper studies a formulation for RLHF by considering reinforcement learning with partially observed reward-states. Under cardinal and dueling feedback, the authors propose new algorithms and obtain improved guarantees compared with previous methods. These theoretical contributions could be of interest to the RL theory community.

The main weakness of the paper is the lack of empirical evaluations.

Given that this paper is technically solid with strong theoretical contributions, I do not think the lack of empirical evaluations is significant enough to prevent this paper being accepted. Therefore, I would recommend acceptance.

**Additional Comments On Reviewer Discussion:**

The reviewers raised concerns regarding the computational efficiency of the proposed algorithm, as well as assumptions about the internal state generation function. The authors provided detailed responses which addressed some of those concerns.

The lack of empirical evaluations is another concern raised by the reviewers. Given that this paper is technically solid with strong theoretical contributions, I do not think the lack of empirical evaluations is significant enough to prevent this paper being accepted

---

### Decision · Program_Chairs · 2025-01-22

Accept (Poster)